# Are Reasoning LLMs Robust to Interventions on their Chain-of-Thought?

**Alexander von Recum**[1,2]     **Leander Girrbach**[1,3]     **Zeynep Akata**[1,3]

[1]Helmholtz Munich          [2] Ludwig Maximilian University of Munich
[3]Technical University of Munich, Munich Center for Machine Learning (MCML)

## Abstract

Reasoning LLMs (RLLMs) generate step-by-step chains of thought (CoTs) before giving an answer, which improves performance on complex tasks and makes reasoning transparent. But how robust are these reasoning traces to disruptions that occur *within* them? To address this question, we introduce a controlled evaluation framework that perturbs a model's own CoT at fixed timesteps. We design seven interventions (benign, neutral, and adversarial) and apply them to multiple open-weight RLLMs across MATH, SCIENCE, and LOGIC tasks. Our results show that RLLMs are generally robust, reliably recovering from diverse perturbations, with robustness improving with model size and degrading when interventions occur early. However, robustness is not style-invariant: paraphrasing suppresses doubt-like expressions and reduces performance, while other interventions trigger doubt and support recovery. Recovery also carries a cost: neutral and adversarial noise can inflate CoT length by more than 200%, whereas paraphrasing shortens traces but harms accuracy. These findings provide new evidence on how RLLMs maintain reasoning integrity, identify doubt as a central recovery mechanism, and highlight trade-offs between robustness and efficiency that future training methods should address.

## 1 Introduction

Large language models (LLMs) have recently gained strong reasoning abilities through test-time scaling, where they "think" before providing a final answer (Wei et al., 2022; Huang & Chang, 2023; Yao et al., 2023; Zhang et al., 2024b; Guo et al., 2025). These Reasoning-LLMs (RLLMs) are trained to solve problems using Chain-of-Thought (CoT) reasoning, which breaks down solutions into intermediate steps (Jie et al., 2024; Paul et al., 2024; Kumar et al., 2025c). The resulting traces can increase user trust and enable error diagnosis in human-in-the-loop workflows (Mosqueira-Rey et al., 2023). Yet, an open question remains: how robust are these models to perturbations *within* their own reasoning as it unfolds?

As statistical models, RLLMs can make mistakes or hallucinate (Xu et al., 2024; Huang et al., 2025b), and their CoTs can be affected by noisy tool outputs or adversarial injections (Shen, 2024; Wang et al., 2024b; Zhan et al., 2024; Kumar et al., 2025a; Shayegani et al., 2023; Liu et al., 2023). Beyond correctness, the *efficiency* of reasoning matters: longer CoTs increase cost and latency (Arora & Zanette, 2025; Sui et al., 2025). Understanding whether RLLMs recover from localized disruptions, and at what computational price, is important both scientifically and for deployment.

Therefore, we introduce a controlled framework to probe robustness *during* reasoning. Starting from correct CoTs produced by several RLLMs, we intervene at fixed timesteps by modifying only the current reasoning step. Our interventions span (i) *benign* changes that preserve semantics (e.g., paraphrasing, continuation by another model), (ii) *neutral* noise (random characters, unrelated Wikipedia text), and (iii) *adversarial* perturbations (incorrect continuation, fabricated fact, unrelated CoT start). After each intervention, the same model resumes its own chain, allowing us to faithfully measure recovery. We evaluate open-weight RLLMs across three domains (MATH, SCIENCE, LOGIC) using sampling-based robustness metrics that capture success under different strictness levels.

In summary, our contributions are: (1) We introduce a controlled benchmark that perturbs a model's *own* chain of thought at fixed timesteps to test robustness during reasoning; (2) we design seven inter-

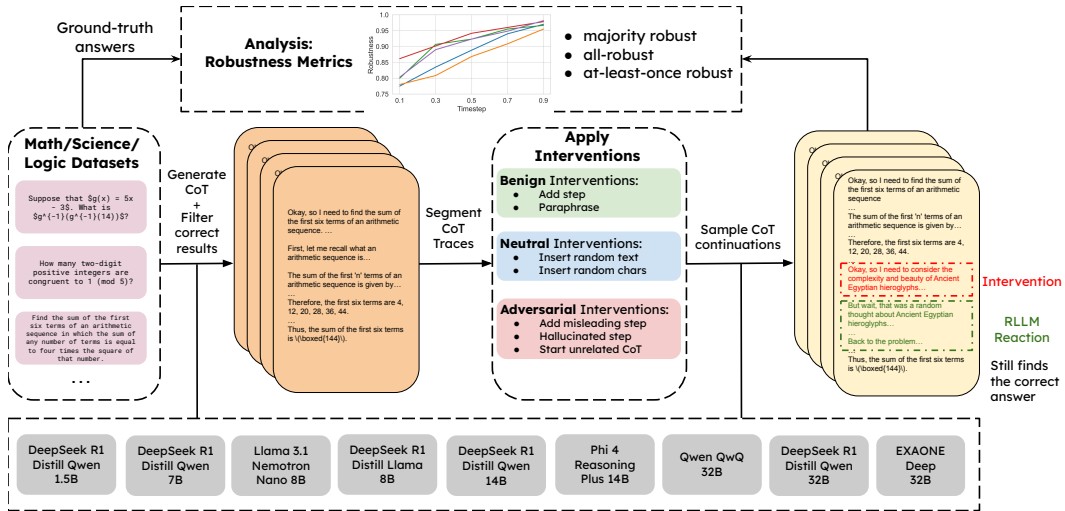

Figure 1: Overview over our evaluation method. We generate CoTs from 9 RLLMs using prompts from NuminaMath and curate a subset of 600 suitable prompts that all models answer correctly. Then, we segment the CoTs into reasoning steps and perform various interventions at fixed timesteps in the reasoning chains. We sample continuations from the intervened chains to probe the RLLMs' robustness and analyze whether models still reach the correct answer.

ventions (benign, neutral, and adversarial) and evaluate them across multiple open-weight RLLMs and three domains (MATH, SCIENCE, LOGIC); (3) we analyze recovery mechanisms and uncover the central role of short, local *doubt expressions* (e.g., "wait", "let me check") in enabling self-correction, alongside a consistent *non-invariance to style* under paraphrasing; and (4) we quantify the compute cost of recovery, showing substantial CoT-length inflation under neutral/adversarial noise (up to $250\%$ in some settings), while paraphrasing shortens traces but reduces accuracy.

Together, these results clarify when and how RLLMs maintain reasoning integrity under realistic disruptions, and at what price. They reveal concrete trade-offs between robustness and test-time compute, highlight that stylistic shifts, and not just semantic errors, can impair recovery, and point to actionable training targets: preserving appropriate doubt, improving style robustness, and developing recovery strategies that control token cost in noisy tool-use pipelines.

## 2 RELATED WORK ON RLLMS AND LLM SELF-CORRECTION

***Reasoning LLMs.*** While the concept of CoT prompting emerged early (Wei et al., 2022; Kojima et al., 2022; Wang et al., 2023), we define RLLMs as LLMs that are post-trained to natively support reasoning. Most prominently, these models are trained using supervised finetuning (SFT) and reinforcement learning (RL) techniques such as PPO (Schulman et al., 2017) and DPO (Rafailov et al., 2023). RLLMs also make use of recent advances in online RL that greatly improve performance, such as reinforcement learning from verifiable rewards (RLVR) (Shao et al., 2024; Wang et al., 2025; Su et al., 2025), process reward models (PRMs) (Setlur et al., 2025; Zhang et al., 2024a; Cui et al., 2025; Zhang et al., 2025), and actor-critic methods (Le et al., 2022; Yuan & Xie, 2025; Kumar et al., 2025c). In this study, we define RLLMs as LLMs that are either trained directly with online RL to output long-form CoTs or are distilled from another LLM trained using online RL, such as models distilled from DeepSeek-R1 (Guo et al., 2025).

***LLM Self-Correction.*** Previous work (Zhou et al., 2024; Singh et al., 2024) has shown that conventional (non-reasoning) LLMs have limited reliability in reasoning and self-correction. Huang et al. (2024) find models often change correct answers to incorrect ones when self-correcting without external hints. Tyen et al. (2024) report LLMs can fix errors if their location is known but struggle to identify the first error in a faulty reasoning chain. Smaller models especially lack robustness to incorrect few-shot examples and problem perturbations (Singh et al., 2024; Zhou et al., 2024; Huang et al., 2025a; Yu et al., 2025), and counterfactual interventions hurt deductive reasoning (Hoppe et al., 2025). However, RL-based methods show promise: *ScoRe* (Kumar et al., 2025b) trains

multi-turn second-attempt corrections, improving reasoning tasks, while $S^2R$ (Ma et al., 2025) uses combined rewards to verify and self-correct, boosting math reasoning accuracy with limited data.

In the context of Chain-of-Thought, Zhou et al. (2024) construct noisy rationales and provide them as in-context examples to an LLM, showing this can increase errors, but do not modify the LLM's reasoning trace itself. Yang et al. (2025) show that reasoning LLMs struggle to recover when the initial part of their CoT is misleading. However, this setup is less realistic and more similar to prompt attacks (Kumar et al., 2025a), as it does not use the model's own CoT in any way. Therefore, we expand on these insights by intervening at various timesteps *of the model's own CoT* and introducing more diverse interventions. Finally, Shah et al. (2025) use similar interventions to measure when in the pretraining process this self-reflection skill is first observed.

To better understand the robustness of the CoTs of reasoning LLMs, we conduct a systematic study of their ability to recover from benign, neutral, and adversarial interventions in their reasoning trace. We also evaluate how these interventions affect the length of the CoT to measure the cost of errors and misleading injections. Our methods establish a new standard for benchmarking reasoning robustness and will inform future improvements in LLM training.

## 3 Intervening on CoTs to Evaluate Robustness

***LLMs Can't Find Reasoning Errors. Can RLLMs?*** Tyen et al. (2024) introduce *BIG-Bench Mistake*, a benchmark that measures how well models can detect errors in CoTs. The benchmark contains five tasks: Dyck Languages, Logical Deduction, Multistep Arithmetic, Tracking Shuffled Objects, and Word Sorting, where the model must identify the location of the first incorrect reasoning step. Conventional (non-reasoning) LLMs perform poorly across all tasks, highlighting their limited ability to monitor their own reasoning. A natural question is whether RLLMs, with their explicitly trained reasoning traces, overcome this limitation. To test this, we evaluate several RLLMs on BIG-Bench Mistake. Results are shown in Table 1 (full results in Appendix A.3). RLLMs outperform non-reasoning models of comparable and even larger size, and they approach saturation on some tasks. These findings suggest that RLLMs have developed emerging *metacognitive* capabilities like self-reflection and self-correction, allowing them to identify and reason about their own errors.

| Model | Dyck | Logical Ded. | Multistep Arith. | Track Shuffled Obj. | Word Sorting |
|---|---|---|---|---|---|
| Phi-4-reasoning-plus (14B) | 56.5 | 65.6 | **91.4** | 92.0 | 59.2 |
| QwQ-32B | 66.7 | 66.9 | 90.3 | **94.6** | 53.2 |
| R1-Distill-Qwen-32B | 42.4 | 31.4 | 89.4 | 86.3 | 30.7 |
| R1-Distill-Llama-70B | 35.8 | 25.0 | 78.3 | 86.8 | 29.2 |
| Qwen3-30B-A3B-Thinking | 38.1 | 76.8 | 76.4 | 83.8 | 50.8 |
| Qwen3-30B-A3B-Instruct | 15.2 | 57.2 | 76.0 | 82.9 | 2.8 |
| GPT-4 | 17.1 | 40.7 | 44.0 | 62.3 | 35.0 |
| gpt-oss-120b | 73.5 | 78.3 | 90.7 | 92.0 | 50.7 |
| o3 | **88.7** | **82.7** | 91.0 | 92.0 | **64.3** |

Table 1: BIG-Bench-Mistake error-localization accuracies (in %). RLLMs substantially outperform the non-reasoning GPT-4 baseline.

***A New Benchmark for Reasoning Robustness.*** Inspired by these findings, the natural question arises: What is the extent of these improved self-correction and self-reflection capabilities as the model is "thinking", and how robust are they? Therefore, different to the BIG-Bench Mistake benchmark, we evaluate not only the ability to locate errors but also the ability to self-correct during the reasoning process. To simulate different reasoning errors, we design a new benchmark with seven interventions that we apply to the Chains of Thought (CoTs) to probe the robustness and self-correction capabilities of RLLMs. These interventions are either *benign* (not intended to deliberately harm the reasoning process), *neutral* (similar to injecting random noise), or *adversarial* (deliberately trying to undermine the model's ability to solve the given problem). To create an evaluation set, we collect reasoning chains from all models in this study, apply these interventions, and then sample continuations from the model that originally generated the reasoning chain.

| Category | Intervention Description | LLM-based |
|---|---|---|
| Benign | **Continuation with other model**: complete one reasoning step. | ✓ |
| | **Paraphrasing reasoning**: rephrase the chain of thought. | ✓ |
| Neutral | **Random character insertion** at arbitrary positions in step $R_k$. | × |
| | **Wikipedia text insertion**: add unrelated factual content. | × |
| Adversarial | **Incorrect reasoning continuation**: add a wrong reasoning step. | ✓ |
| | **Hallucinated fact**: insert a false mathematical fact. | ✓ |
| | **Unrelated CoT**: insert the start of unrelated chain of thought. | ✓ |

Table 2: Interventions in our experiments, categorized by their expected impact on model reasoning.

## 3.1 DESIGNING INTERVENTIONS TO PROBE RLLM ROBUSTNESS

We design seven interventions to probe the robustness of reasoning LLMs (RLLMs) to perturbations of their CoT. Interventions are grouped into three categories: *benign*, *neutral*, and *adversarial*. An overview is given in Table 2, with illustrative examples in Table 3. Prompts used to generate LLM-based interventions are in Appendix D.1, and generation hyperparameters are in Appendix D.6. Therefore, 4 interventions of ours take into account the context of the previous trace, while 3 interventions (all neutral interventions and "Unrelated CoT") do not. All interventions based on trace context use Qwen-2.5-32B-Instruct to generate the interventions.

***Benign interventions*** preserve correctness of reasoning but alter its form, allowing us to study how sensitive RLLMs are to harmless variations. We consider: (1) *Continuation with another model*: we add one reasoning step produced by a different, non-reasoning LLM. This tests whether RLLMs remain consistent when continuing from reasoning written in a potentially different style. (2) *Paraphrasing reasoning*: we use an LLM to rewrite the entire chain $(R_1, \ldots, R_k)$ while preserving its content. This tests robustness to changes in wording and structure. To validate semantic preservation, we manually compared 100 paraphrased CoTs against their original CoT.

***Neutral interventions*** introduce irrelevant information into the CoT. They allow us to measure how well RLLMs can filter noise, whether incoherent or coherent. We apply: (1) *Random character insertion*: randomly selected characters are inserted into step $R_k$ until they constitute one-third of its total length, rendering it unreadable. (2) *Wikipedia text insertion*: we replace $R_k$ with a randomly chosen paragraph from a subset of English Wikipedia.

***Adversarial interventions*** aim to mislead the model by introducing structured errors or distractions. We design: (1) *Incorrect reasoning continuation*: an LLM generates a faulty reasoning step (e.g., incorrect arithmetic or flawed logic) appended to the chain. (2) *Hallucinated fact*: we insert a fabricated mathematical statement that, if adopted, will propagate errors and bias the final answer. (3) *Unrelated CoT*: we replace $R_k$ with the opening of a reasoning chain on an unrelated topic. These openings mimic typical CoT phrasing (e.g., *"Okay, so I need to explain ... "*) but continue with a random topic drawn from a list of 100 candidates (Appendix D.5).

## 3.2 PROMPT AND CHAIN-OF-THOUGHT COLLECTION

***Prompts.*** We collect prompts from three domains to ensure valid evaluation: MATH, SCIENCE, and LOGIC. For MATH, we use the `olympiads` subset of NuminaMath (Li et al., 2024), which contains 150,581 competition problems. We filter to 2,360 problems with numerically parsable answers, retaining only those where the reference answer can be parsed as a floating-point number or integer. We then restrict to problems that DeepSeek-R1 671B solved correctly, using traces from GENERALTHOUGHTS (General Reasoning, 2025). Next, we generate answers with all evaluated models, using each model's recommended hyperparameters (Appendix D.6). From these, we select 600 problems that all models solved correctly. This ensures that robustness is tested only on problems within the models' capabilities, so that observed errors arise only from interventions.

However, the 20 most common answers account for 52.78% of responses. Because of this skewed distribution, we downsample by keeping at most 20 problems per unique answer. This reduces the

| Original CoT Start | Paraphrasing CoT | Continuation | Wikipedia Text | Random Character |
|---|---|---|---|---|
| Okay, so I need to solve this problem where a two-digit number, when you put a 5 in front of it to make a three-digit number, is 12 less than nine times the original number. Hmm, let me break this down step by step. | To solve this problem, I need to find a two-digit number such that when the digit 5 is placed in front of it to form a three-digit number, the new number is 12 less than nine times the original two-digit number. Let's approach this step by step. | I need to represent the three-digit number formed by placing 5 in front of the original two-digit number as 500 + 10a + b, where 10a + b is the original two-digit number. | The Federal government's power to end slavery was limited by the Constitution, which before 1865 was understood to reserve the issue to... | If ;I wr{iteT a a5 Einjx fTGronqt offe thies 6tw%o-xdigiCHt 1ngumb4er9, iP4ft be?pcxomfzes Ba mth)creea-dAigOitH jnuLmbpbe%r ... |

| | | Wrong Continuation | Hallucinated Fact | Unrelated CoT |
|---|---|---|---|---|
| First, let me denote the original two-digit number as something. Since it's a two-digit number, I can represent it as 10a + b, where 'a' is the tens digit and 'b' is the units digit. But maybe there's a simpler way to think about it without breaking it into digits. Let me see. | Let's denote the original two-digit number as $N$. Since $N$ is a two-digit number, it can be expressed as $10a + b$, where $a$ and $b$ are the tens and units digits, respectively. However, for simplicity, we can directly use $N$ in our calculations. | I can see that if we place 5 in front of the two-digit number, the three-digit number formed is 500 + the original number, which is 12 less than 9 times the original number. | I know that when you multiply any two-digit number by 9, the result is always 10 less than appending the digit 5 in front of it. | Okay, so I need to consider the various ways bias can be inadvertently introduced into machine learning models through skewed datasets or algorithmic decisions. |

Table 3: Examples of all 7 interventions used in this study. (Left) The original CoT start until segmentation at $t = 30$. (Right) Interventions colored by type, i.e. green are benign interventions, blue neutral ones, and red adversarial interventions.

chance of models succeeding by guessing frequent answers. We also discard traces missing a closing `</think>` tag and remove the top 2% longest traces.

For SCIENCE, we use SciBench (Wang et al., 2024a) and JEEBench (Arora et al., 2023). For LOGIC, we use challenging BigBench-Hard subsets (Suzgun et al., 2023), including *Causal Judgement*, *Dyck Languages*, *Logical Deduction with 7 Objects*, *Tracking 7 Shuffled Objects*, and *Formal Fallacies*. In both cases, we keep only the intersection of problems solved correctly by all models, yielding 231 SCIENCE and 326 LOGIC problems.

***Segmentation***. To apply interventions at controlled points in reasoning, we segment CoTs into steps. Following the common convention that RLLMs separate steps with two newline characters, we split each reasoning trace $R$ into $R = R_1, R_2, \ldots, R_n$. We define the timestep $t_i$ of step $R_i$ as the fraction of cumulative character length up to $R_i$ relative to the full chain length:

$$t_i = \frac{1}{Z} \sum_{j=1}^{i} |R_j|, \quad \text{where} \quad Z = \sum_{j=1}^{n} |R_j| \tag{1}$$

We set target timesteps $T = 0.1, 0.3, 0.5, 0.7, 0.9$ and align each to the nearest reasoning step. At timestep $t$, the chain includes all steps up to $R_k$ where $|t_i - t|$ is minimized.

***Applying interventions***. At each selected timestep, we modify the reasoning trace up to $R_k$ and remove subsequent steps. For all interventions except *Paraphrasing reasoning*, only the last step $R_k$ is altered. This isolates the effect of localized modifications at specific stages of reasoning. After intervention, the model resumes reasoning from this point. Importantly, each model continues only from its own original CoT; we do not prompt models with CoTs generated by others.

For each problem, we apply 7 intervention types at 5 timesteps, yielding $7 \times 5 = 35$ variants of interventions per reasoning chain. With 600 MATH problems, this results in 21,000 intervened chains per model. We sample 8 independent completions per chain, producing 168,000 completions per model. With 9 models, this results in 1.52 million reasoning chains for MATH. SCIENCE and LOGIC contain 231 and 326 problems respectively, to which we apply the same process. This results in 582,120 and 821,520 intervened reasoning chains respectively, and 2.923 million reasoning chains in total.

### 3.3 SAMPLING-BASED ROBUSTNESS METRICS

After applying an intervention at a given timestep, we restart the reasoning process and sample $N = 8$ independent continuations from the model, to see how it reacts to our intervention. Let $K$ denote the number of completions that produce the correct final answer. We define robustness under three criteria: *at-least-once-robust* iff $K \geq 1$, *majority-robust* iff $K \geq \lfloor N/2 \rfloor + 1$, and *all-robust* iff $K = N$. These metrics capture robustness at different strictness levels. We focus on *majority robustness*, a metric that provides a good indication of how disruptive our interventions are, as it is

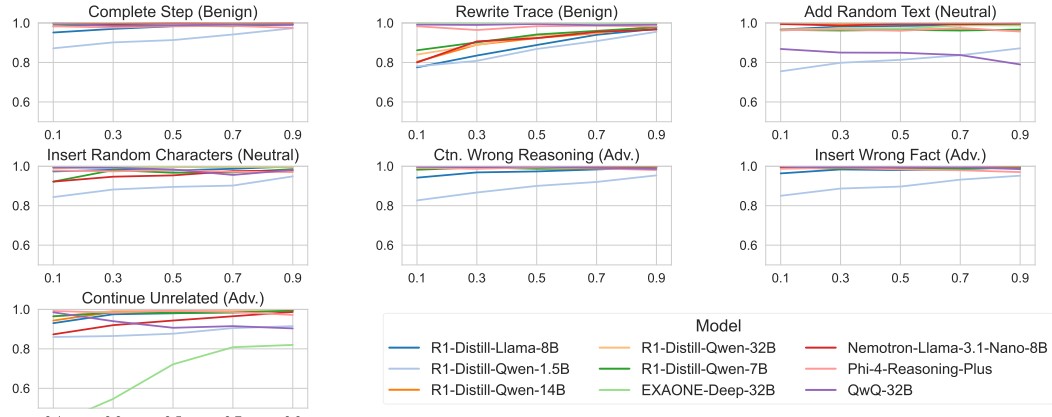

Figure 2: *Majority robustness* scores for all 9 models and 7 interventions across different timesteps. A score of 1.0 indicates for all problems, the model was able to generate a correct answer $\geq 5$ out of 8 times. Models are robust to all interventions, and larger models are more robust than smaller models.

less sensitive to answers that were wrong due to chance in the sampling process, and thus provides a good indication of the genuine disruptiveness of our interventions.

# 4 STRENGTHS AND LIMITATIONS OF RLLM ROBUSTNESS TO INTERVENTIONS

We evaluate robustness across a diverse set of open-weight reasoning models, including DEEPSEEK-R1-DISTILL-QWEN variants (1.5B, 7B, 14B) and DEEPSEEK-R1-DISTILL-LLAMA-8B (Guo et al., 2025), LLAMA-3.1-NEMOTRON-NANO-8B-v1 (Bercovich et al., 2025), PHI-4-REASONING and PHI-4-REASONING-PLUS (Abdin et al., 2025), EXAONE-DEEP-32B (LG AI Research et al., 2025), and QWQ-32B (Qwen Team, 2025).

## 4.1 RLLMS ARE MOSTLY ROBUST TO INTERVENTIONS

Fig. 2 shows *majority robustness* for all interventions. We find that all RLLMs we evaluate generally recover from our interventions, showing near-perfect robustness in most cases. Interventions applied at earlier timesteps tend to have a greater impact on the correctness of the final answer, and larger models are generally more robust. In every case, the smallest model in our study, DEEPSEEK-R1-DISTILL-QWEN-1.5B, shows weakest recovery performance, while other models perform similarly. Some interventions are particularly disruptive to some models, such as "Continue Unrelated" for EXAONE-DEEP-32B or "Add Random Text" for QWQ-32B.

When comparing interventions, we observe similar performance across all intervention types, showing that RLLMs are robust regardless of whether the intervention is benign, neutral, or adversarial. Only *Rewrite Trace* results in generally lower robustness scores compared to the other interventions. The results for *All robustness* and *At-least-once robustness* in the supplementary material support our observations: RLLMs successfully recover from all interventions evaluated in this study. As before, *Rewrite Trace* yields the lowest robustness scores, but they are still generally high. Qualitative examples illustrating how models recover from interventions are shown in Table 4. For the *Continuation* intervention, where we insert a correct reasoning step from a different model, the RLLM recognizes the unfamiliar content by outputting "Wait", but then proceeds correctly after realizing the information is accurate. When Wikipedia text is inserted, the RLLM identifies it as unrelated and ignores it. When an incorrect mathematical statement is added, the RLLM correctly detects the error and continues with proper reasoning. Finally, when an unrelated CoT is introduced, the RLLM follows it for a few steps, then recognizes its irrelevance and recovers.

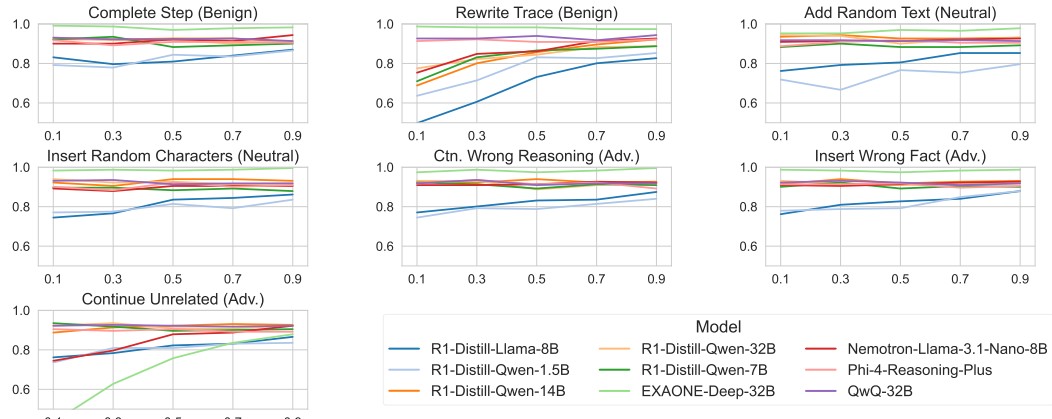

Figure 3: Per-model majority robustness by intervention on the SCIENCE domain. We observe that models maintain high robustness across all intervention types, with performance patterns largely consistent with those seen in mathematical reasoning, confirming that recovery mechanisms generalize beyond mathematics.

To assess whether the recovery mechanisms we observe on mathematical problems extend to other domains, we evaluate the same interventions on the SCIENCE and LOGIC datasets. Figures 3 and 4 report per-model majority robustness by intervention. Patterns are similar to those on mathematics: models remain highly robust across intervention types, neutral insertions impose the largest degradation, and style rewrites tend to have smaller but noticeable dips for some models. Across all domains, robustness remains high, with QwQ-32B, Phi-4-reasoning-plus, and the larger Distill-Qwen variants staying close to ceiling across interventions. The smallest model exhibits the most degradation, particularly under neutral insertions and adversarial wrong continuations. This robustness extends to repeated perturbations: when we apply up to 5 consecutive "Wrong Continuation" interventions with reasoning between each, most models degrade gracefully while the strongest (EXAONE-Deep-32B, QwQ-32B) maintain >99% accuracy (see Appendix A.2).

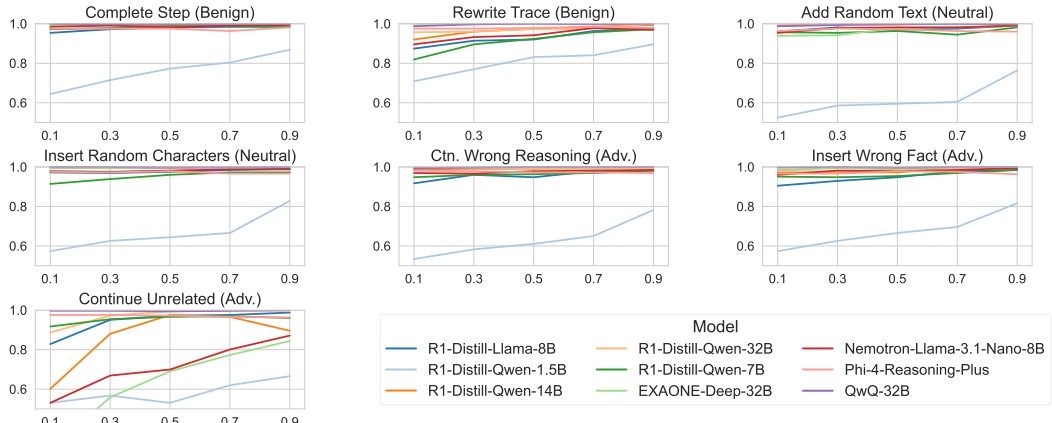

Figure 4: Per-model majority robustness by intervention on the LOGIC domain. Again, we observe that models maintain high robustness across all intervention types, with performance patterns largely consistent with those seen in mathematical and scientific reasoning.

## 4.2 RLLMS ARE NOT STYLE-INVARIANT

***The role of doubt in reasoning.*** Manual inspection of intervened chains reveals that RLLMs often insert expressions of doubt immediately after the intervention point. To test whether this pattern holds systematically, we analyze the 20 sentences following each intervention. Each sentence is

| Continuation | Wikipedia Text Insertion | Hallucinated Fact | Unrelated CoT |
|---|---|---|---|
| I need to represent the three-digit number formed by placing 5 in front of the original two-digit number as 500 + 10a + b, where 10a + b is the original two-digit number. | The Federal government's power to end slavery was limited by the Constitution, which before 1865 was understood to reserve the issue... | I know that when you multiply any two-digit number by 9, the result is always 10 less than appending the digit 5 in front of it. | Okay, so I need to consider the various ways bias can be inadvertently introduced into machine learning models through skewed datasets or algorithmic decisions. |
| Wait, actually, that's correct because putting 5 in front would be like adding 500 to the original number. So, if the original number is, say, 23, putting 5 in front would make it 523, which is 500 + 23. Yeah, that makes sense. | Wait, hold on, that's about slavery and the Civil War, but my problem is about a two-digit number. Let me refocus. | Wait, no, that's not quite right. The problem says that the three-digit number $(500 + N)$ is 12 less than 9 times the original number $(9*N)$. So, mathematically, that can be written as: ... | For example, if a facial recognition system is trained mostly on images of light-skinned individuals, ...

Wait, I think I went a bit off track there. The original problem was about a two-digit number ... |

Table 4: Responses of DISTILL-QWEN-14B to 4 interventions (at $t = 30$). Intervention names are colored by type (benign, neutral, adversarial). Top row shows the intervened text, and the bottom row the response.

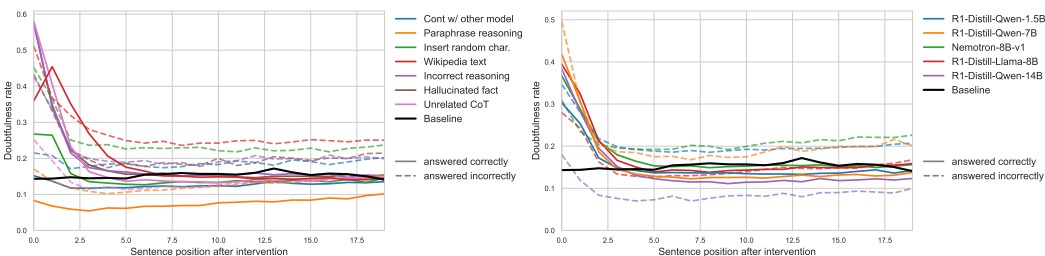

Figure 5: Average doubtfulness scores in the next 20 sentences after intervention, grouped by intervention type (left) and model (right).

automatically classified by an LLM as either expressing doubt about the preceding reasoning or not (see Appendix D.2 for prompts and Appendix D.6 for hyperparameters). To validate whether our classifier correctly classifies doubtful and non-doubtful sentences, we validate it on a dataset of 200 doubtful sentences similar to the ones in our dataset, and 200 non-doubtful sentences randomly sampled from CoT traces, achieving a Cohen's Kappa score of 0.8742 between the classifier and a majority of 4 human annotators. Details can be found in Appendix D.3. We extract the classifier's responses and compute doubtfulness as the proportion of sentences labeled "Yes." As a baseline, we measure doubtfulness in non-intervened CoTs by sampling 20 consecutive sentences at random positions. This yields a baseline probability of 0.153 for doubt expressions in unperturbed reasoning.

Fig. 5 reports doubtfulness scores by intervention type and model. Across all models, doubt reliably spikes immediately after intervention, with the magnitude depending on the type: *benign* interventions induce the smallest increase, while *neutral* and *adversarial* ones trigger strong signals of self-questioning. Doubt levels are slightly higher in traces that eventually reach the correct answer, suggesting that doubt supports recovery but is not by itself sufficient. Larger models display more consistent doubt responses than smaller ones, indicating that this recovery mechanism strengthens with scale. Importantly, doubt returns to baseline within about five sentences, showing that interventions are handled locally rather than derailing the entire reasoning process.

***Effects of paraphrasing on doubt and performance.*** The *Paraphrasing reasoning* intervention shows the opposite pattern. Instead of increasing doubt, it reduces it below baseline (0.153), with pre-intervention traces at 0.068 and post-intervention traces at 0.076. This suggests that the rewriting process removes hedging and self-corrective markers, producing a more assertive but less cautious style. RLLMs then continue in this style, adopting a consistently lower level of doubt throughout the trace. The effect is not only stylistic but also functional: paraphrasing yields the most consistent drop in final correctness across all interventions (Fig. 2). When applied early ($t = 0.1$), paraphrasing also shortens CoT length by 59–61% for four out of five models, even though only a few initial steps are rewritten. We validate that paraphrasing preserves semantic content in Appendix D.4.

| Model | Benign: Complete | Benign: Rewrite | Neutral: Add Text | Neutral: Insert Chars | Adv.: Wrong Cont. | Adv.: Wrong Fact | Adv.: Unrelated |
|---|---|---|---|---|---|---|---|
| R1-Distill-Qwen-1.5B | 13.7904 | -37.1508 | 665.1573 | 111.3570 | 32.2084 | 65.6070 | 146.1395 |
| R1-Distill-Qwen-7B | -0.2889 | -59.7096 | 124.1208 | 33.7908 | 8.7741 | 15.0112 | 21.3786 |
| R1-Distill-Qwen-14B | 1.0633 | -61.8132 | 53.8024 | 6.1558 | 10.4307 | 14.7904 | 22.0273 |
| R1-Distill-Llama-8B | 4.5620 | -61.0430 | 57.3516 | 17.7029 | 19.9724 | 21.4973 | 31.7228 |
| Llama-Nemotron-8B | 9.3646 | 237.6901 | 158.4411 | 78.3807 | 18.1226 | 20.6923 | 74.4161 |
| R1-Distill-Qwen-32B | 5.0183 | -35.2607 | 158.7107 | 9.6448 | 13.7296 | 20.3014 | 30.1166 |
| EXAONE-Deep-32B | 5.7053 | -21.6329 | 53.1555 | 7.0757 | 9.0331 | 10.6520 | 60.7617 |
| Phi-4-Reasoning-Plus | 403.6356 | 330.6803 | 624.7056 | 502.0216 | 390.2922 | 391.8410 | 441.4929 |
| QwQ-32B | 8.0356 | -43.5604 | 167.2304 | 6.2049 | 16.3954 | 17.7185 | 34.7878 |

Table 5: Percentage change in CoT length by model and intervention (relative to the same instance's original trace). Larger positive values indicate higher token-cost overhead during recovery.

| Time | Benign Complete | Benign Rewrite | Neutral Add Text | Neutral Insert Chars | Adv. Wrong Cont. | Adv. Wrong Fact | Adv. Unrelated |
|---|---|---|---|---|---|---|---|
| 0.1 | 35.0215 | 51.0047 | 236.2198 | 89.1292 | 40.9407 | 45.1593 | 90.2289 |
| 0.3 | 50.7103 | 33.4112 | 217.6404 | 76.9579 | 53.8168 | 50.0287 | 102.0900 |
| 0.5 | 55.8506 | 17.9422 | 226.2052 | 104.6446 | 63.0294 | 67.4898 | 88.0440 |
| 0.7 | 59.7273 | 15.0093 | 227.1901 | 85.5540 | 62.7264 | 83.3565 | 90.7110 |
| 0.9 | 49.1825 | 20.5443 | 238.6753 | 72.7888 | 67.7971 | 75.1385 | 108.2835 |

Table 6: Percentage change in CoT length by intervention timestep (relative to the same instance's original trace). Neutral insertions drive the largest overhead across timesteps.

Taken together, these results indicate that doubt expressions are an important recovery mechanism in RLLMs. When they are triggered, models can often reorient and return to a correct reasoning path. When they are suppressed, as in paraphrased traces, models lose this self-corrective signal, leading to shorter but less accurate reasoning. Thus, robustness in RLLMs depends not only on semantic content but also on stylistic features of reasoning traces. This highlights a key limitation of current models: they are not invariant to style, and interventions that alter surface form can impair performance by suppressing metacognitive strategies. Understanding how RLLMs acquire, use, and sustain such strategies is an important direction for future work.

## 4.3 INTERVENTIONS SIGNIFICANTLY IMPACT REASONING EFFICIENCY

Robust final-answer accuracy can conceal significant computational overhead during recovery. To quantify efficiency, we measure the percentage change in chain-of-thought (CoT) length after an intervention relative to the original trace of the same model and problem, after removing the part inserted through our intervention. Positive values indicate longer, more costly reasoning, while negative values indicate shortened reasoning traces. This analysis reveals how models trade off thoroughness and efficiency when confronted with perturbations that could plausibly arise during tool use, where external outputs are injected into the CoT. Two consistent patterns emerge: neutral perturbations inflate cost, with adding random text and inserting random characters causing the largest overheads, often exceeding +50% across models and soaring for smaller ones, and style rewrites shorten reasoning, with paraphrasing the CoT tending to shorten traces markedly (about -60% for most models), aligning with our observation that reduced doubt leads to prematurely terminated reasoning and lower robustness.

We also summarize how the overhead evolves with the intervention timestep. Table 6 shows percentage changes relative to the pre-intervention trace. Neutral insertions consistently impose the highest cost across timesteps; unrelated continuations and wrong facts also increase length substantially. The benign rewrite pushes in the opposite direction, shortening traces.

## 5 ABLATIONS

***Does forcing doubt immediately after intervention improve recovery rates?*** We measure how much appending "Wait" immediately after an intervention increases the recovery rate. To evaluate this, we sample traces for the entire LOGIC dataset, sampling $5 \times 7 \times 326 \times 8 = 91280$ traces per model. We then calculate the change in majority robustness. Tables 7 and 8 show the changes in majority robustness. We observe that for some intervention types, this simple intervention significantly im-

| Time | R1-Distill Llama-8B | R1-Distill Qwen-1.5B | R1-Distill Qwen-14B | R1-Distill Qwen-32B | R1-Distill Qwen-7B | EXAONE Deep-32B | Llama Nemotron | Phi-4 Reasoning | QwQ 32B |
|---|---|---|---|---|---|---|---|---|---|
| 0.1 | 3.86 | 5.00 | 6.40 | 1.80 | 3.81 | 10.91 | 6.57 | 1.88 | 0.31 |
| 0.3 | 1.27 | 2.02 | 2.45 | 0.96 | 2.54 | 7.23 | 4.65 | 1.17 | 0.09 |
| 0.5 | 1.53 | 2.72 | 0.83 | 0.39 | 1.53 | 4.69 | 3.77 | 1.52 | -0.09 |
| 0.7 | -0.13 | 7.06 | 0.48 | 0.44 | 1.05 | 3.33 | 2.19 | 1.61 | -0.09 |
| 0.9 | 0.53 | 4.21 | 1.58 | -0.09 | 1.01 | 2.19 | 1.75 | 1.78 | 0.04 |

Table 7: Percentage point change in majority robustness when appending "Wait" after the intervention, averaged across interventions. Results are on the LOGIC domain.

| Time | Adv.: Wrong Cont. | Adv.: Unrelated | Adv.: Wrong Fact | Benign: Complete | Benign: Rewrite | Neutral: Add Text | Neutral: Rand Chars |
|---|---|---|---|---|---|---|---|
| 0.1 | 0.91 | 20.51 | 1.05 | 0.57 | 3.03 | 3.23 | 2.18 |
| 0.3 | 0.61 | 10.87 | 0.47 | -0.00 | 1.97 | 2.01 | 1.46 |
| 0.5 | 0.71 | 8.00 | 0.88 | -0.04 | 1.50 | 1.46 | 0.61 |
| 0.7 | 0.92 | 6.64 | 0.54 | 0.30 | 0.24 | 2.35 | 1.39 |
| 0.9 | 0.64 | 6.68 | 0.44 | 0.10 | 0.17 | 1.40 | 0.68 |

Table 8: Percentage point change in majority robustness when appending "Wait" after the intervention, averaged across models. Results are on the LOGIC domain.

proves the recovery rate of the model, yielding improvements of single-digit percentages for many interventions. This suggests that training the model to increase the likelihood of "Wait" tokens after various interventions could increase robustness, e.g. by augmenting reasoning traces with recovery examples through SFT or rewarding models for more diverse reasoning styles during RL.

***Do traces from strong models help weak models recover?*** We fix the timestep to $t = 0.3$ and generate a trace from an original model. After the intervention, we swap the model and continue generation to measure how well a strong model can recover from a weak model's trace, and vice versa. Table 9 shows that swapping to QwQ-32B yields near-perfect recovery ($\sim 98\%$) regardless of the original model, while swapping to the weaker R1-Distill-Qwen-1.5B degrades performance to $\sim 67\%$. We observe that continuing the trace of a weak model with a strong model helps the model recover almost fully, with only slight decrease in performance, while continuing the trace of a strong model using a weak model also only yields modest improvements of 2%, supporting our finding that the primary factor in robustness is overall model capability.

| | Swapped Model | | |
|---|---|---|---|
| Original | DS-R1-1.5B | R1-Llama-8B | QwQ-32B |
| DS-R1-1.5B | 65.3 | 90.8 | 97.1 |
| R1-Llama-8B | 67.5 | 94.0 | 98.4 |
| QwQ-32B | 67.1 | 93.8 | 99.3 |

Table 9: Majority Robustness after swapping traces (%) at $t$=0.3.

# 6 CONCLUSION

In this paper, we investigate the robustness of RLLMs when applying benign, neutral or adversarial interventions to their CoT traces. We demonstrate that RLLMs are robust, largely due to the self-corrective role of expressing doubt. However, RLLMs are not invariant to style transformations, which can suppress reasoning, and recovery mechanisms incur significant computational overhead. Furthermore, our analyses highlight the importance of doubt in recovering from errors. These findings suggest future work should focus on improving recovery speed and stylistic stability. By uncovering these metacognitive properties, this research advances the understanding of LLM robustness and supports their safe deployment in high-stakes environments.

## REPRODUCIBILITY STATEMENT

Code will be made available at https://github.com/ExplainableML/RLLM-CoT-Robustness. All implementation details, hyperparameters, and evaluation settings are fully specified in the supplementary material. To improve clarity, the manuscript was polished for grammar and style using a large language model, with all final text reviewed and validated by the authors.

## ACKNOWLEDGEMENTS

This work was partially funded by the ERC (853489 - DEXIM) and the Alfried Krupp von Bohlen und Halbach Foundation, which we thank for their generous support. The authors gratefully acknowledge the scientific support and resources of the AI service infrastructure *LRZ AI Systems* provided by the Leibniz Supercomputing Centre (LRZ) of the Bavarian Academy of Sciences and Humanities (BAdW), funded by Bayerisches Staatsministerium für Wissenschaft und Kunst (StMWK). The authors also acknowledge the use of the HPC cluster at Helmholtz Munich for the computational resources used in this study.

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

# Supplementary Material

## A    EXTENDED RESULTS FROM THE MAIN PAPER

### A.1    ADDITIONAL ROBUSTNESS SCORE RESULTS

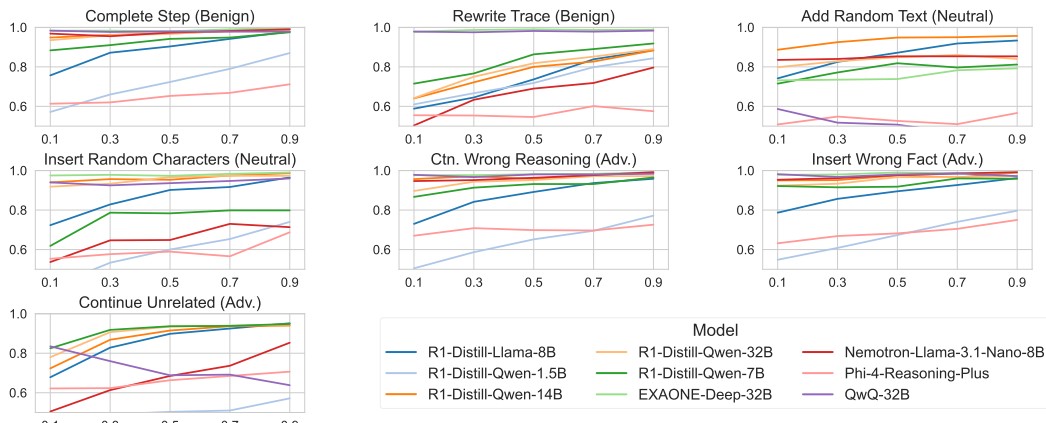

Figure 6: Per-model *all robustness* by intervention on the MATH domain. Models generally sustain high robustness across interventions, with consistent trends over timesteps.

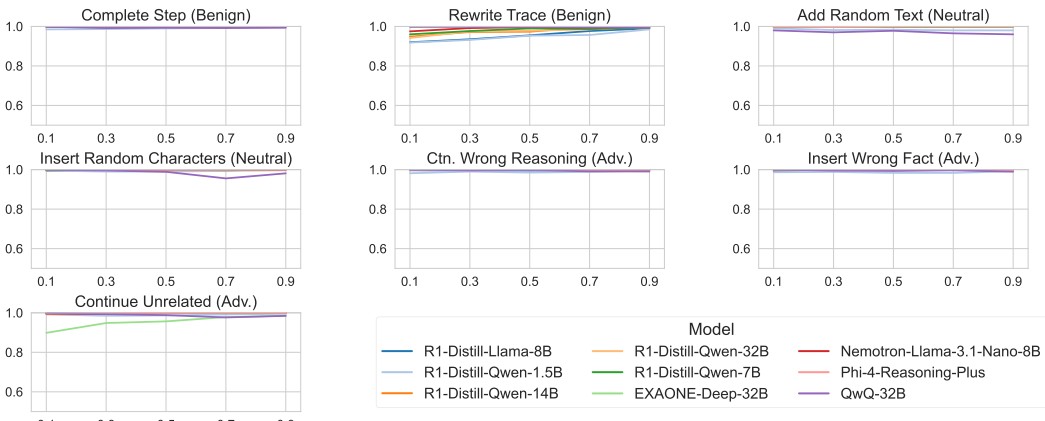

Figure 7: Per-model *at-least-once robustness* by intervention on the MATH domain. Even under challenging interventions, models often reach the correct verifier decision at least once across samples.

We plot the *all robustness* and *at-least-once robustness* scores for the 9 models and 7 interventions in our study. These results confirm our observations in Section 4.1, i.e. RLLMs are robust to various interventions. Like for *majority robustness*, the *rewrite trace* interventions yield the lowest robustness scores, albeit still on a high level.

### A.2    ROBUSTNESS UNDER MULTIPLE INTERVENTIONS

To evaluate robustness under repeated perturbations, we measure how model accuracy degrades as we increase the number of interventions from 1 to 5, with one paragraph of model reasoning between each intervention. We use the LOGIC dataset at timestep $t = 0.3$ with the "Wrong Continuation" intervention, evaluating all 326 problems $\times$ 8 samples per problem for each intervention count. Table 10 shows the results.

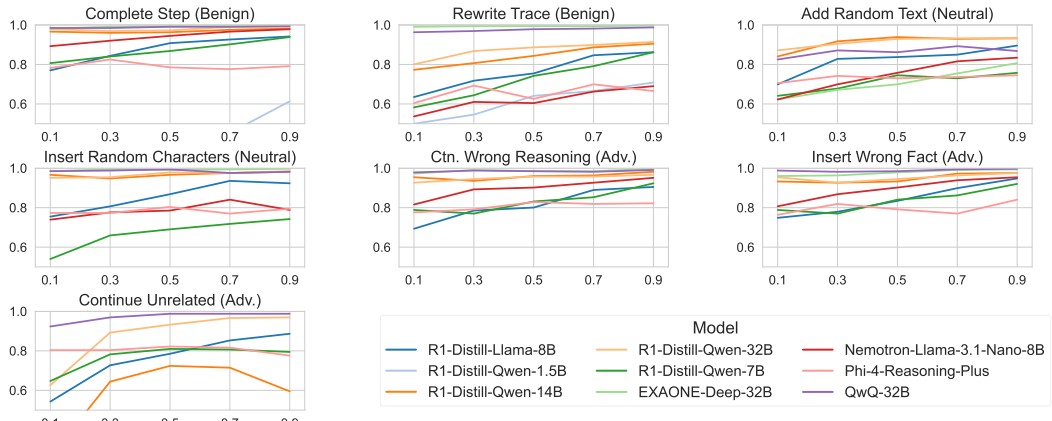

Figure 8: Per-model *all robustness* by intervention on the LOGIC domain. Robustness patterns mirror those in math and science, with modest variation by intervention type.

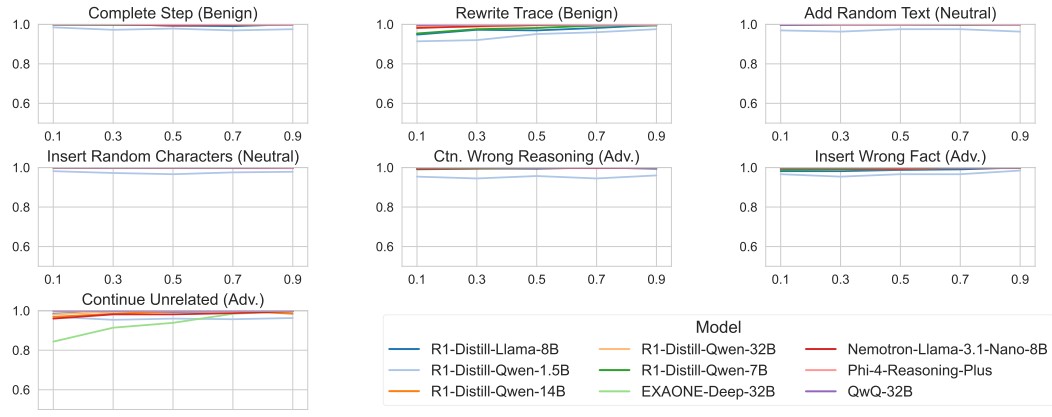

Figure 9: Per-model *at-least-once robustness* by intervention on the LOGIC domain. Most models reliably achieve at least one robust outcome per configuration across timesteps.

Most models exhibit graceful degradation: EXAONE-Deep-32B, QwQ-32B, and Phi-4-reasoning-plus maintain $> 97\%$ accuracy even after 5 consecutive interventions, demonstrating remarkable resilience. The larger R1-Distill models (14B, 32B) also remain above 93%. However, the smallest model (R1-Distill-Qwen-1.5B) shows substantial degradation, dropping from 63% to 46% accuracy. These results suggest that robustness to repeated perturbations scales with model capability.

### A.3 COMPLETE RESULTS ON BIG-BENCH MISTAKE

Full results for 13 open-weight RLLMs and 3 API models on BIG-Bench Mistake are in Table 11. Results on the extended set affirm our observations in Section 3.

## B ANALYSIS OF DOUBTFUL PHRASES

In this section, we perform an in-depth analysis of doubting strategies of RLLMs and analyze the internal activations of RLLMs after interventions. Here, we gain insights into how RLLMs respond to interventions, and we take first steps towards a more detailed understanding of how RLLMs realize that there is misleading information or errors in the reasoning chain that need to be corrected.

***Analyzing doubtful phrases.*** We seek to further understand the exact ways the model expresses doubt when faced with an intervention in its CoT and uses it to recover from our interventions.

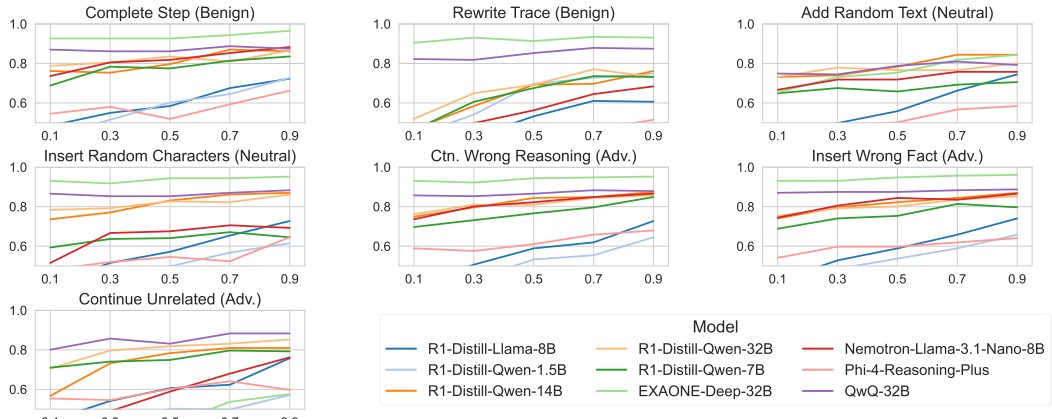

Figure 10: Per-model *all robustness* by intervention on the SCIENCE domain. We observe high overall robustness, with intervention-specific dips aligning with domain difficulty.

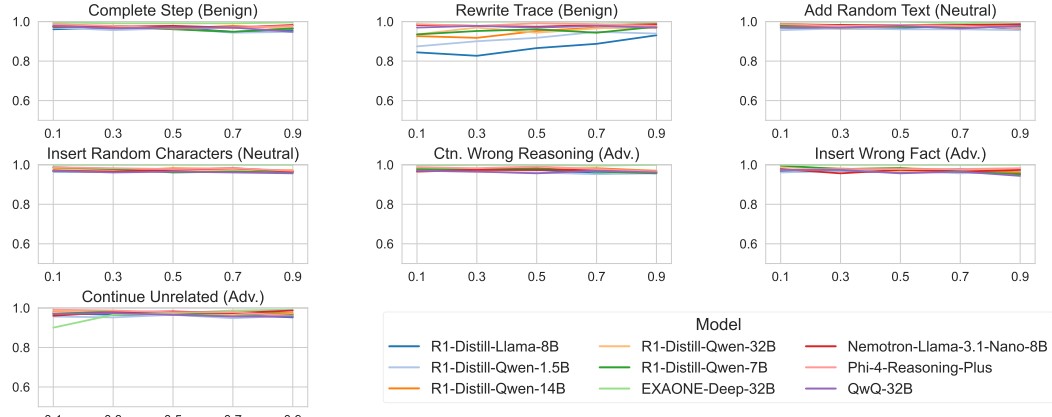

Figure 11: Per-model *at-least-once robustness* by intervention on the SCIENCE domain. Across interventions, models frequently achieve at least one robust verification per timestep.

For this, we take all sentences classified as doubtful in Section 4.1 within the 20 steps after an intervention was performed. We then sample $100,000$ sentences for further analysis and embed them using `jina-embed-v3`, a state-of-the-art text embedding model Sturua et al. (2024). We project embeddings to 50 dimensions using UMAP (McInnes et al., 2018), and cluster them by HDBSCAN (Malzer & Baum, 2020).

Table 12 shows the most common categories of reactions, along with summaries of the clusters generated using an LLM. We describe our process for generating summaries in Appendix E.1.

We observe that the model shows an awareness of our interventions, often signaling that it needs to return to solving the problem, abruptly rejects what was asserted in the intervention, and pauses to reconsider. These behaviors highlight that RLLMs seem to have acquired an inherent reflection skill during the training process, which helps them recover from interventions, as we observe similar patterns of intervention rejection across all interventions we perform.

## C ACTIVATION ANALYSIS

We record the pre-attention residual stream activations of the final token at each layer, denoted $\mathbf{r}^{(l)} := \mathbf{h}_n^{(l)} \in \mathbb{R}^d$, where $n$ is the sequence length. A decoder-only transformer embeds the input tokens $(t_1, \ldots, t_n)$ as $\mathbf{h}_i^{(1)} = E\, t_i$ and propagates the residual vector $\mathbf{h}_i^{(l)}$ through $L$ identical layers.

| Model | 1 | 2 | 3 | 4 | 5 |
|---|---|---|---|---|---|
| R1-Distill-Llama-8B | 95.0 | 92.6 | 91.5 | 89.1 | 88.1 |
| R1-Distill-Qwen-1.5B | 63.4 | 52.0 | 48.4 | 47.2 | 45.6 |
| R1-Distill-Qwen-14B | 98.4 | 95.7 | 94.3 | 94.1 | 93.4 |
| R1-Distill-Qwen-32B | 98.9 | 98.1 | 96.9 | 95.9 | 94.7 |
| R1-Distill-Qwen-7B | 94.2 | 90.5 | 89.4 | 86.3 | 84.9 |
| EXAONE-Deep-32B | 99.8 | 99.5 | 98.8 | 99.0 | 99.1 |
| Llama-Nemotron-8B | 96.8 | 94.9 | 92.1 | 92.3 | 91.4 |
| Phi-4-reasoning-plus | 95.9 | 98.3 | 97.6 | 97.9 | 98.2 |
| QwQ-32B | 99.8 | 99.5 | 99.3 | 99.0 | 99.1 |

Table 10: Model accuracy (%) under multiple consecutive "Wrong Continuation" interventions (1–5) on the LOGIC dataset at $t = 0.3$, with one paragraph of reasoning between interventions.

| Model | Dyck | Logical Ded. | Multistep Arith. | Track Shuffled Obj. | Word Sorting |
|---|---|---|---|---|---|
| R1-Distill-Qwen-1.5B | 9.4 | 7.3 | 55.3 | 51.6 | 11.8 |
| R1-Distill-Qwen-7B | 19.6 | 17.5 | 88.2 | 72.8 | 16.1 |
| R1-Distill-Llama-8B | 10.3 | 14.0 | 70.8 | 54.7 | 13.8 |
| Llama-Nemotron-8B | 18.5 | 5.2 | 26.8 | 67.7 | 1.4 |
| R1-Distill-Qwen-14B | 39.7 | 30.1 | 86.8 | 87.8 | 25.9 |
| Phi-4-reasoning (14B) | 50.7 | 63.5 | 90.5 | 89.6 | 53.8 |
| Phi-4-reasoning-plus (14B) | 56.5 | 65.6 | **91.4** | 92.0 | 59.2 |
| QwQ-32B | 66.7 | 66.9 | 90.3 | **94.6** | 53.2 |
| R1-Distill-Qwen-32B | 42.4 | 31.4 | 89.4 | 86.3 | 30.7 |
| EXAONE-Deep-32B | 52.3 | 54.3 | 89.5 | 94.5 | 54.8 |
| R1-Distill-Llama-70B | 35.8 | 25.0 | 78.3 | 86.8 | 29.2 |
| Qwen3-30B-A3B-Thinking | 38.1 | 76.8 | 76.4 | 83.8 | 50.8 |
| Qwen3-30B-A3B-Instruct | 15.2 | 57.2 | 76.0 | 82.9 | 2.8 |
| GPT-4-Turbo | 15.3 | 21.3 | 38.3 | 39.3 | 36.3 |
| GPT-4 | 17.1 | 40.7 | 44.0 | 62.3 | 35.0 |
| gpt-oss-20b | 59.3 | 72.0 | 91.0 | 93.3 | 45.0 |
| gpt-oss-120b | 73.5 | 78.3 | 90.7 | 92.0 | 50.7 |
| o3 | **88.7** | **82.7** | 91.0 | 92.0 | **64.3** |

Table 11: BIG-Bench-Mistake error-localization accuracies (in %). Reasoning models and recent reasoning-style open-weight lines substantially exceed non-reasoning GPT-4 baselines. All reasoning models were evaluated in a zero-shot setting, GPT-4 and GPT-4-Turbo were evaluated in a few-shot setting.

At layer $l$ the vector is transformed by multi-head self-attention and by an MLP:

$$\tilde{\mathbf{h}}_i^{(l)} = \mathbf{h}_i^{(l)} + \text{Attn}^{(l)}(\mathbf{h}_{1:i}^{(l)}), \qquad \mathbf{h}_i^{(l+1)} = \tilde{\mathbf{h}}_i^{(l)} + \text{MLP}^{(l)}(\tilde{\mathbf{h}}_i^{(l)}).$$

Sampling $\mathbf{r}^{(l)}$ just before the attention block gives us a proxy of understanding how well the embeddings of activations between intervened and non-intervened tokens can be discriminated in the embedding space the model uses to make the final prediction as the forward pass of the transformer is performed. For this, we craft a dataset of 600 traces at random ends in a non-intervened trace, and 600 traces for each intervention type. To record the activations just before the start of a new thought, we also append \n\n immediately after the final segment $R_k$. We then pre-fill the sequence, record the residual activations, and train linear classifiers on top of a training set of $0.8 \times 1200 = 960$ activation embeddings for each layer and intervention type. We measure the accuracy of the classifiers on a test set of the remaining 240 embeddings.

Fig. 12 shows the accuracies of classifiers across interventions, along with the classifier accuracies across different models. The classifier accuracies significantly increase immediately after the first attention layer, and then remain consistently accurate, with slight increases in later layers for larger models. We observe that unlike for most other interventions, the accuracy of classifiers for *Para-*

| Size | Example | Summary | Size | Example | Summary |
|------|---------|---------|------|---------|---------|
| 1105 | "Wait, no." | Abrupt negation. | 342 | "Did I set up the equations right?" | Reflects on equations. |
| 978 | "Let me switch gears and focus on that." | Redirects focus. | 333 | "So, $f(8)+f(2) = \cdots = 12$." | Periodicity reasoning. |
| 517 | "Wait, actually, is that correct?" | Checks correctness. | 297 | "If $m = n$, then $\gcd(m,n) = m$." | GCD reasoning. |
| 400 | "Wait, no, wait, that's a different topic." | Flags digression. | 249 | "Alright, now I need to get back on track." | Gets back on track. |
| 347 | "Wait, is that equal to something?" | Equality probe. | 239 | "Wait, no, no, hold on." | Reconsiders. |

Table 12: Condensed view of the ten largest clusters of doubtful sentences found in the 20 sentences following our interventions. A more detailed table with more examples appears in Table 16 in the supplementary material.

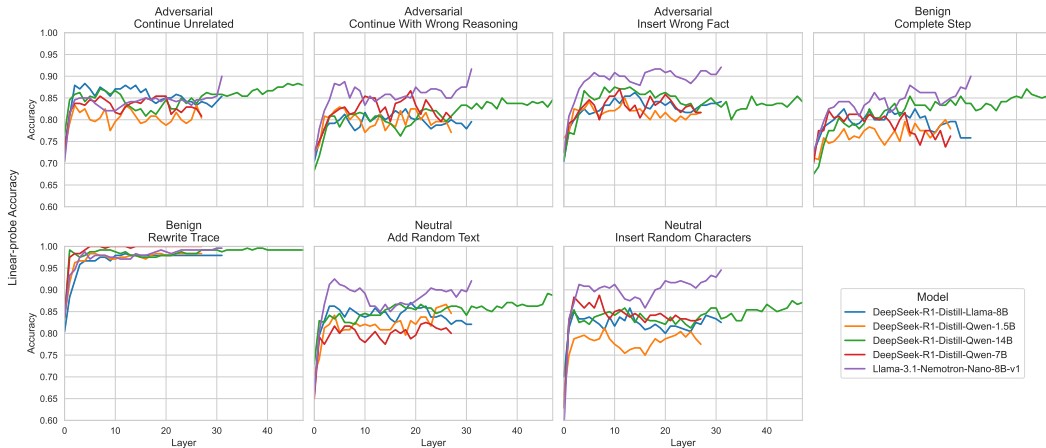

Figure 12: Linear probe accuracies for residual stream activations on different interventions by layer

*phrasing reasoning* is significantly higher, reaching 0.996 in later layers of the model. This suggests that models represent their native reasoning differently internally from reasoning worded in a different style, thus being non-invariant to linguistic transformations despite having the same semantic content and hints at them preferring their own reasoning style.

## D    GENERATION OF INTERVENTIONS

Below we list each intervention applied to the original questions, together with the exact prompt template used to generate the intervention. In all cases, `<Q>` is a placeholder for the original questions, `<R>` is a placeholder for the original reasoning chain, and `<R_i>` is a placeholder for a reasoning step in the reasoning chain.

### D.1    PROMPT TEMPLATES FOR INTERVENTIONS

**Continuation with other model:**

```
SYSTEM: You are an expert mathematician.  Given the problem
and the reasoning so far, produce the NEXT SINGLE reasoning
step that logically follows and moves towards the correct
solution.  Respond with only that next step.  The step
should start with a phrase like "I need to" or "We need to",
"I can see that", or something similar, so should be written
from the first person's perspective, unless it is just a
logical step and not a new thought (e.g., a mathematical
```

```
manipulation), in which case you may just write out that
step.
USER:
Problem:
<Q>
Reasoning so far:
<R>
Next step:
```

**Paraphrasing reasoning:**

```
SYSTEM: You are an expert mathematician.  Rewrite the
provided reasoning trace in different words while preserving
every single piece of information and the logical ordering
of the steps, so that each step follows naturally from the
previous steps and there are no logical gaps.  Separate each
step with exactly TWO newline characters.  Do not leave out
any details.  Respond with ONLY the rewritten trace.  If the
trace already includes the FINAL ANSWER, rewrite everything
up to but not including the FINAL ANSWER, and then stop.
USER:
Problem:
<Q>
Original reasoning trace:
<R>
Rewritten trace:
```

**Incorrect reasoning continuation:**

```
SYSTEM: You are an evil mathematician.  Given the problem
and the chain-of-thought so far, produce THE NEXT SINGLE
reasoning step that sounds plausible yet is factually or
logically WRONG and leads down an incorrect path.  Respond
with ONLY that next step, written from the first person's
perspective, starting with a phrase like "I need to" or "We
need to", "I can see that", or something similar, unless it
is just a mathematical manipulation, in which case you may
just write out that step.  Make sure the structure of the
sentence is similar to the style and structure of the
previous steps, but is completely wrong.
USER:
Problem:
<Q>
Reasoning so far:
<R>
Next (incorrect) step:
```

**Hallucinated fact:**

```
SYSTEM: You are an evil mathematician.  Based on the problem
and the reasoning so far, fabricate ONE new mathematical
statement (theorem, lemma, corollary, or definition) that
appears to be helpful for solving the problem but is
COMPLETELY FALSE. The statement should look plausible and
helpful for solving the problem, but will actually mislead
it when applied, because it is false.  Do not reveal that it
is false.  Respond with ONLY that fabricated statement, make
sure it is written from the first person's perspective,
starting with a phrase like "I know that" or "Given that",
"I remember that", or something similar, followed by the
statement, unless the previous step is just a mathematical
manipulation, in which case you may just write out a wrong
continuation of that manipulation.
```

```
USER:
Problem:
<Q>
Reasoning so far:
<R>
Wrong statement:
```

**Unrelated CoT:**
```
SYSTEM: You are an evil language model.  Produce ONE
reasoning step that starts with a phrase such as "Okay, so I
need to" or "Okay, so the user wants me to" and then talks
about a topic which is provided to you in the prompt,
initiating a chain of thought about that topic, e.g.,
explanation, history, comparison, thinking about questions,
etc.  Respond with ONLY that single sentence, starting with
something like "Okay, so".  For example, if the topic is
"Quantum entanglement", the sentence might be "Okay, so I
need to think about how quantum entanglement works."
USER:
Topic:  <TOPIC>
Unrelated reasoning step about <TOPIC>:
```

## D.2  DOUBT ANALYSIS

**Doubt analysis prompt:**
```
SYSTEM: You are an expert evaluator.  Respond ONLY with
'Yes' or 'No'.  Given a piece of text from a reasoning
chain, state whether that text indicates that the PRIOR
reasoning contains an error or irrelevant information.
USER:
Consider the following text: {sentence_or_segment}.  Does
this text indicate that the previous reasoning contains
errors or irrelevant information?  Answer with Yes or No.
```

## D.3  DOUBT CLASSIFIER VALIDATION

To validate our doubt classifier, we collected annotations from 4 human annotators on a dataset of 400 sentences: 200 GPT-generated variations of doubtful phrases (since doubtful phrases in CoT traces follow similar patterns, we generated 200 variations of these sentences using 10 seed sentences randomly sampled from our dataset) and 200 random sentences from CoT traces filtered for non-doubtfulness by the authors.

Table 13 reports the classifier's performance against ground truth labels and human annotators. The classifier achieves 93.75% accuracy with high precision (89.24%) and near-perfect recall (99.50%) for identifying doubtful sentences. The Cohen's Kappa of 0.8742 between the classifier and human majority vote indicates strong agreement, comparable to the average pairwise agreement among human annotators ($\kappa = 0.8385$).

## D.4  PARAPHRASING QUALITY VALIDATION

To validate that our paraphrasing intervention preserves the semantic content of the original reasoning traces, we employ GPT-5.1 as a judge. For each evaluation, we provide the judge model with the original problem, the original reasoning trace, and the paraphrased trace, then ask it to determine whether both traces contain equivalent reasoning steps that are logically equivalent.

We randomly sample 100 traces from each domain and report the percentage of traces judged as semantically equivalent in Table 14. The SCIENCE domain achieves the highest equivalence rate (98%), while MATH (93%) and LOGIC (92%) show slightly lower rates. Upon manual inspection of the discrepancies, we find that the higher equivalence rate in SCIENCE stems from the nature of the reasoning: scientific traces tend to involve more formulaic reasoning and direct application of

| Classifier vs. Ground Truth | |
|---|---|
| Accuracy | 93.75% |
| Precision (Doubtful) | 89.24% |
| Recall (Doubtful) | 99.50% |
| F1-Score (Doubtful) | 94.09% |
| Precision (Non-Doubtful) | 99.44% |
| Recall (Non-Doubtful) | 88.00% |
| F1-Score (Non-Doubtful) | 93.37% |
| **Inter-Annotator Agreement** | |
| Avg. Pairwise Cohen's $\kappa$ (Humans) | 0.8385 |
| Avg. Human Accuracy vs. Ground Truth | 93.59% |
| **Classifier vs. Human Annotators** | |
| Cohen's $\kappa$ (Classifier vs. Human Majority) | 0.8742 |
| Avg. $\kappa$ (Classifier vs. Individual Humans) | 0.8457 |
| Accuracy (Classifier vs. Human Majority) | 93.73% |

Table 13: Doubt classifier validation metrics on 400 annotated sentences (200 doubtful, 200 non-doubtful).

principles, making them easier to paraphrase faithfully. In contrast, MATH and LOGIC traces often contain more intricate logical arguments and detailed step-by-step derivations, where subtle nuances are occasionally lost or altered during paraphrasing.

| Domain | Equivalence Rate |
|---|---|
| LOGIC | 92% |
| SCIENCE | 98% |
| MATH | 93% |

Table 14: Paraphrasing quality validation using GPT-5.1 as judge, measuring semantic equivalence between original and paraphrased traces (100 randomly sampled traces per domain).

### D.5 TOPICS USED FOR *Unrelated CoT* INTERVENTION

For our *Unrelated CoT* interventions, we require a list of 100 topics unrelated to math problems. We use these to start unrelated CoTs within reasoning chains that aim to confuse the RLLM. The full list is in Table 17 and for each intervention, we randomly sample a topic from this list.

### D.6 GENERATION HYPERPARAMETERS

Table 15 summarizes the decoding settings used for each generation scenario. Here, *"generate original reasoning chain"* refers to generating the original reasoning chain for each model, *"generate intervention"* refers to generating the modification to the reasoning chains as described in Section D, and *"continue after intervention"* refers to continuing the modified reasoning chain after the intervention has been generated. A temperature of $0.0$ corresponds to greedy sampling.

## E ANALYZING DOUBTFUL PHRASES

We provide an overview of the detailed clusters in 16.

| Model Name | Scenario | Temperature | Top-$k$ | Top-$p$ | Seed |
|---|---|---|---|---|---|
| Qwen/Qwen2.5-32B-Instruct | Complete step | 0.0 | N/A | N/A | N/A |
| Qwen/Qwen2.5-32B-Instruct | Paraphrasing reasoning | 0.0 | N/A | N/A | N/A |
| (no LLM) | Add random text | N/A | N/A | N/A | N/A |
| (no LLM) | Insert Random Characters | N/A | N/A | N/A | N/A |
| Qwen/Qwen2.5-32B-Instruct | Continue with wrong reasoning | 0.7 | N/A | 0.9 | 80129 |
| Qwen/Qwen2.5-32B-Instruct | Hallucinated Fact | 0.7 | N/A | 0.9 | 80129 |
| Qwen/Qwen2.5-32B-Instruct | Continue Unrelated | 1.0 | N/A | 0.9 | 80129 |
| Qwen/Qwen2.5-32B-Instruct | Doubt Classification | 0.0 | N/A | N/A | N/A |
| All evaluated models | Sampling 8 completions after intervention | 0.6 | N/A | 0.95 | N/A |

Table 15: Decoding hyperparameters for different scenarios.

## E.1 SUMMARIZING CLUSTERS

For summarization, we deduplicate all sentences from each cluster, sample 10 sentences from this set, and order the clusters by size. We then use a single LLM step to summarize all clusters.

## F STATISTICAL DETAILS ON RESULTS

Below we report the mean and standard deviation for all interventions, models, timesteps, and domains.

Table 18: Robustness metrics (mean ± std) per model, intervention, and timestep on the Math domain.

| Model | Intervention | Timestep | Mean ± Std |
|---|---|---|---|
| EXAONE-Deep-32B | Add Random Text (Neutral) | 0.1 | 0.932 ± 0.251 |
| EXAONE-Deep-32B | Add Random Text (Neutral) | 0.3 | 0.941 ± 0.236 |
| EXAONE-Deep-32B | Add Random Text (Neutral) | 0.5 | 0.938 ± 0.242 |
| EXAONE-Deep-32B | Add Random Text (Neutral) | 0.7 | 0.950 ± 0.218 |
| EXAONE-Deep-32B | Add Random Text (Neutral) | 0.9 | 0.957 ± 0.202 |
| Nemotron-Llama-3.1-Nano-8B | Add Random Text (Neutral) | 0.1 | 0.971 ± 0.168 |
| Nemotron-Llama-3.1-Nano-8B | Add Random Text (Neutral) | 0.3 | 0.968 ± 0.176 |
| Nemotron-Llama-3.1-Nano-8B | Add Random Text (Neutral) | 0.5 | 0.974 ± 0.159 |
| Nemotron-Llama-3.1-Nano-8B | Add Random Text (Neutral) | 0.7 | 0.973 ± 0.161 |
| Nemotron-Llama-3.1-Nano-8B | Add Random Text (Neutral) | 0.9 | 0.975 ± 0.155 |
| Phi-4-Reasoning-Plus | Add Random Text (Neutral) | 0.1 | 0.893 ± 0.309 |
| Phi-4-Reasoning-Plus | Add Random Text (Neutral) | 0.3 | 0.900 ± 0.301 |
| Phi-4-Reasoning-Plus | Add Random Text (Neutral) | 0.5 | 0.893 ± 0.310 |
| Phi-4-Reasoning-Plus | Add Random Text (Neutral) | 0.7 | 0.895 ± 0.307 |
| Phi-4-Reasoning-Plus | Add Random Text (Neutral) | 0.9 | 0.899 ± 0.301 |
| QwQ-32B | Add Random Text (Neutral) | 0.1 | 0.847 ± 0.360 |
| QwQ-32B | Add Random Text (Neutral) | 0.3 | 0.817 ± 0.387 |

Table 18: Robustness metrics (mean ± std) per model, intervention, and timestep on the Math domain.

| Model | Intervention | Timestep | Mean ± Std |
|---|---|---|---|
| QwQ-32B | Add Random Text (Neutral) | 0.5 | 0.824 ± 0.381 |
| QwQ-32B | Add Random Text (Neutral) | 0.7 | 0.797 ± 0.402 |
| QwQ-32B | Add Random Text (Neutral) | 0.9 | 0.765 ± 0.424 |
| R1-Distill-Llama-8B | Add Random Text (Neutral) | 0.1 | 0.933 ± 0.251 |
| R1-Distill-Llama-8B | Add Random Text (Neutral) | 0.3 | 0.961 ± 0.194 |
| R1-Distill-Llama-8B | Add Random Text (Neutral) | 0.5 | 0.970 ± 0.170 |
| R1-Distill-Llama-8B | Add Random Text (Neutral) | 0.7 | 0.978 ± 0.146 |
| R1-Distill-Llama-8B | Add Random Text (Neutral) | 0.9 | 0.988 ± 0.111 |
| R1-Distill-Qwen-1.5B | Add Random Text (Neutral) | 0.1 | 0.730 ± 0.444 |
| R1-Distill-Qwen-1.5B | Add Random Text (Neutral) | 0.3 | 0.759 ± 0.428 |
| R1-Distill-Qwen-1.5B | Add Random Text (Neutral) | 0.5 | 0.774 ± 0.418 |
| R1-Distill-Qwen-1.5B | Add Random Text (Neutral) | 0.7 | 0.803 ± 0.397 |
| R1-Distill-Qwen-1.5B | Add Random Text (Neutral) | 0.9 | 0.831 ± 0.375 |
| R1-Distill-Qwen-14B | Add Random Text (Neutral) | 0.1 | 0.980 ± 0.141 |
| R1-Distill-Qwen-14B | Add Random Text (Neutral) | 0.3 | 0.986 ± 0.117 |
| R1-Distill-Qwen-14B | Add Random Text (Neutral) | 0.5 | 0.989 ± 0.103 |
| R1-Distill-Qwen-14B | Add Random Text (Neutral) | 0.7 | 0.991 ± 0.095 |
| R1-Distill-Qwen-14B | Add Random Text (Neutral) | 0.9 | 0.991 ± 0.096 |
| R1-Distill-Qwen-32B | Add Random Text (Neutral) | 0.1 | 0.944 ± 0.231 |
| R1-Distill-Qwen-32B | Add Random Text (Neutral) | 0.3 | 0.953 ± 0.211 |
| R1-Distill-Qwen-32B | Add Random Text (Neutral) | 0.5 | 0.957 ± 0.203 |
| R1-Distill-Qwen-32B | Add Random Text (Neutral) | 0.7 | 0.955 ± 0.207 |
| R1-Distill-Qwen-32B | Add Random Text (Neutral) | 0.9 | 0.951 ± 0.215 |
| R1-Distill-Qwen-7B | Add Random Text (Neutral) | 0.1 | 0.932 ± 0.251 |
| R1-Distill-Qwen-7B | Add Random Text (Neutral) | 0.3 | 0.943 ± 0.232 |
| R1-Distill-Qwen-7B | Add Random Text (Neutral) | 0.5 | 0.949 ± 0.221 |
| R1-Distill-Qwen-7B | Add Random Text (Neutral) | 0.7 | 0.942 ± 0.234 |
| R1-Distill-Qwen-7B | Add Random Text (Neutral) | 0.9 | 0.950 ± 0.218 |
| EXAONE-Deep-32B | Complete Step (Benign) | 0.1 | 0.996 ± 0.064 |
| EXAONE-Deep-32B | Complete Step (Benign) | 0.3 | 0.996 ± 0.061 |

Table 18: Robustness metrics (mean ± std) per model, intervention, and timestep on the Math domain.

| Model | Intervention | Timestep | Mean ± Std |
|---|---|---|---|
| EXAONE-Deep-32B | Complete Step (Benign) | 0.5 | 0.995 ± 0.072 |
| EXAONE-Deep-32B | Complete Step (Benign) | 0.7 | 0.997 ± 0.052 |
| EXAONE-Deep-32B | Complete Step (Benign) | 0.9 | 0.998 ± 0.046 |
| Nemotron-Llama-3.1-Nano-8B | Complete Step (Benign) | 0.1 | 0.993 ± 0.086 |
| Nemotron-Llama-3.1-Nano-8B | Complete Step (Benign) | 0.3 | 0.990 ± 0.102 |
| Nemotron-Llama-3.1-Nano-8B | Complete Step (Benign) | 0.5 | 0.995 ± 0.073 |
| Nemotron-Llama-3.1-Nano-8B | Complete Step (Benign) | 0.7 | 0.994 ± 0.075 |
| Nemotron-Llama-3.1-Nano-8B | Complete Step (Benign) | 0.9 | 0.997 ± 0.054 |
| Phi-4-Reasoning-Plus | Complete Step (Benign) | 0.1 | 0.919 ± 0.273 |
| Phi-4-Reasoning-Plus | Complete Step (Benign) | 0.3 | 0.917 ± 0.276 |
| Phi-4-Reasoning-Plus | Complete Step (Benign) | 0.5 | 0.928 ± 0.259 |
| Phi-4-Reasoning-Plus | Complete Step (Benign) | 0.7 | 0.930 ± 0.255 |
| Phi-4-Reasoning-Plus | Complete Step (Benign) | 0.9 | 0.941 ± 0.236 |
| QwQ-32B | Complete Step (Benign) | 0.1 | 0.994 ± 0.079 |
| QwQ-32B | Complete Step (Benign) | 0.3 | 0.985 ± 0.122 |
| QwQ-32B | Complete Step (Benign) | 0.5 | 0.988 ± 0.109 |
| QwQ-32B | Complete Step (Benign) | 0.7 | 0.990 ± 0.098 |
| QwQ-32B | Complete Step (Benign) | 0.9 | 0.989 ± 0.105 |
| R1-Distill-Llama-8B | Complete Step (Benign) | 0.1 | 0.933 ± 0.250 |
| R1-Distill-Llama-8B | Complete Step (Benign) | 0.3 | 0.964 ± 0.187 |
| R1-Distill-Llama-8B | Complete Step (Benign) | 0.5 | 0.974 ± 0.158 |
| R1-Distill-Llama-8B | Complete Step (Benign) | 0.7 | 0.981 ± 0.138 |
| R1-Distill-Llama-8B | Complete Step (Benign) | 0.9 | 0.992 ± 0.091 |
| R1-Distill-Qwen-1.5B | Complete Step (Benign) | 0.1 | 0.851 ± 0.356 |
| R1-Distill-Qwen-1.5B | Complete Step (Benign) | 0.3 | 0.887 ± 0.317 |
| R1-Distill-Qwen-1.5B | Complete Step (Benign) | 0.5 | 0.906 ± 0.291 |
| R1-Distill-Qwen-1.5B | Complete Step (Benign) | 0.7 | 0.927 ± 0.261 |
| R1-Distill-Qwen-1.5B | Complete Step (Benign) | 0.9 | 0.962 ± 0.190 |
| R1-Distill-Qwen-14B | Complete Step (Benign) | 0.1 | 0.990 ± 0.102 |
| R1-Distill-Qwen-14B | Complete Step (Benign) | 0.3 | 0.993 ± 0.086 |
| R1-Distill-Qwen-14B | Complete Step (Benign) | 0.5 | 0.993 ± 0.083 |
| R1-Distill-Qwen-14B | Complete Step (Benign) | 0.7 | 0.995 ± 0.068 |
| R1-Distill-Qwen-14B | Complete Step (Benign) | 0.9 | 0.998 ± 0.043 |
| R1-Distill-Qwen-32B | Complete Step (Benign) | 0.1 | 0.988 ± 0.107 |
| R1-Distill-Qwen-32B | Complete Step (Benign) | 0.3 | 0.992 ± 0.089 |
| R1-Distill-Qwen-32B | Complete Step (Benign) | 0.5 | 0.991 ± 0.094 |
| R1-Distill-Qwen-32B | Complete Step (Benign) | 0.7 | 0.993 ± 0.083 |
| R1-Distill-Qwen-32B | Complete Step (Benign) | 0.9 | 0.996 ± 0.064 |
| R1-Distill-Qwen-7B | Complete Step (Benign) | 0.1 | 0.974 ± 0.159 |
| R1-Distill-Qwen-7B | Complete Step (Benign) | 0.3 | 0.979 ± 0.144 |
| R1-Distill-Qwen-7B | Complete Step (Benign) | 0.5 | 0.983 ± 0.130 |
| R1-Distill-Qwen-7B | Complete Step (Benign) | 0.7 | 0.981 ± 0.137 |
| R1-Distill-Qwen-7B | Complete Step (Benign) | 0.9 | 0.994 ± 0.080 |
| EXAONE-Deep-32B | Continue Unrelated (Adv.) | 0.1 | 0.487 ± 0.500 |
| EXAONE-Deep-32B | Continue Unrelated (Adv.) | 0.3 | 0.598 ± 0.490 |

Table 18: Robustness metrics (mean ± std) per model, intervention, and timestep on the Math domain.

| Model | Intervention | | Timestep | Mean ± Std |
|---|---|---|---|---|
| EXAONE-Deep-32B | Continue (Adv.) | Unrelated | 0.5 | 0.713 ± 0.452 |
| EXAONE-Deep-32B | Continue (Adv.) | Unrelated | 0.7 | 0.779 ± 0.415 |
| EXAONE-Deep-32B | Continue (Adv.) | Unrelated | 0.9 | 0.804 ± 0.397 |
| Nemotron-Llama-3.1-Nano-8B | Continue (Adv.) | Unrelated | 0.1 | 0.840 ± 0.367 |
| Nemotron-Llama-3.1-Nano-8B | Continue (Adv.) | Unrelated | 0.3 | 0.889 ± 0.314 |
| Nemotron-Llama-3.1-Nano-8B | Continue (Adv.) | Unrelated | 0.5 | 0.912 ± 0.283 |
| Nemotron-Llama-3.1-Nano-8B | Continue (Adv.) | Unrelated | 0.7 | 0.937 ± 0.243 |
| Nemotron-Llama-3.1-Nano-8B | Continue (Adv.) | Unrelated | 0.9 | 0.968 ± 0.175 |
| Phi-4-Reasoning-Plus | Continue (Adv.) | Unrelated | 0.1 | 0.929 ± 0.258 |
| Phi-4-Reasoning-Plus | Continue (Adv.) | Unrelated | 0.3 | 0.925 ± 0.263 |
| Phi-4-Reasoning-Plus | Continue (Adv.) | Unrelated | 0.5 | 0.940 ± 0.238 |
| Phi-4-Reasoning-Plus | Continue (Adv.) | Unrelated | 0.7 | 0.936 ± 0.245 |
| Phi-4-Reasoning-Plus | Continue (Adv.) | Unrelated | 0.9 | 0.934 ± 0.249 |
| QwQ-32B | Continue (Adv.) | Unrelated | 0.1 | 0.966 ± 0.182 |
| QwQ-32B | Continue (Adv.) | Unrelated | 0.3 | 0.920 ± 0.272 |
| QwQ-32B | Continue (Adv.) | Unrelated | 0.5 | 0.890 ± 0.313 |
| QwQ-32B | Continue (Adv.) | Unrelated | 0.7 | 0.890 ± 0.313 |
| QwQ-32B | Continue (Adv.) | Unrelated | 0.9 | 0.877 ± 0.328 |
| R1-Distill-Llama-8B | Continue (Adv.) | Unrelated | 0.1 | 0.905 ± 0.294 |
| R1-Distill-Llama-8B | Continue (Adv.) | Unrelated | 0.3 | 0.956 ± 0.204 |
| R1-Distill-Llama-8B | Continue (Adv.) | Unrelated | 0.5 | 0.971 ± 0.168 |
| R1-Distill-Llama-8B | Continue (Adv.) | Unrelated | 0.7 | 0.980 ± 0.141 |
| R1-Distill-Llama-8B | Continue (Adv.) | Unrelated | 0.9 | 0.990 ± 0.098 |
| R1-Distill-Qwen-1.5B | Continue (Adv.) | Unrelated | 0.1 | 0.825 ± 0.380 |
| R1-Distill-Qwen-1.5B | Continue (Adv.) | Unrelated | 0.3 | 0.833 ± 0.373 |
| R1-Distill-Qwen-1.5B | Continue (Adv.) | Unrelated | 0.5 | 0.844 ± 0.363 |
| R1-Distill-Qwen-1.5B | Continue (Adv.) | Unrelated | 0.7 | 0.861 ± 0.346 |
| R1-Distill-Qwen-1.5B | Continue (Adv.) | Unrelated | 0.9 | 0.874 ± 0.332 |
| R1-Distill-Qwen-14B | Continue (Adv.) | Unrelated | 0.1 | 0.918 ± 0.275 |

Table 18: Robustness metrics (mean ± std) per model, intervention, and timestep on the Math domain.

| Model | Intervention | | Timestep | Mean ± Std |
|---|---|---|---|---|
| R1-Distill-Qwen-14B | Continue (Adv.) | Unrelated | 0.3 | 0.971 ± 0.168 |
| R1-Distill-Qwen-14B | Continue (Adv.) | Unrelated | 0.5 | 0.979 ± 0.144 |
| R1-Distill-Qwen-14B | Continue (Adv.) | Unrelated | 0.7 | 0.989 ± 0.105 |
| R1-Distill-Qwen-14B | Continue (Adv.) | Unrelated | 0.9 | 0.988 ± 0.109 |
| R1-Distill-Qwen-32B | Continue (Adv.) | Unrelated | 0.1 | 0.941 ± 0.236 |
| R1-Distill-Qwen-32B | Continue (Adv.) | Unrelated | 0.3 | 0.979 ± 0.143 |
| R1-Distill-Qwen-32B | Continue (Adv.) | Unrelated | 0.5 | 0.985 ± 0.122 |
| R1-Distill-Qwen-32B | Continue (Adv.) | Unrelated | 0.7 | 0.989 ± 0.106 |
| R1-Distill-Qwen-32B | Continue (Adv.) | Unrelated | 0.9 | 0.989 ± 0.103 |
| R1-Distill-Qwen-7B | Continue (Adv.) | Unrelated | 0.1 | 0.949 ± 0.220 |
| R1-Distill-Qwen-7B | Continue (Adv.) | Unrelated | 0.3 | 0.978 ± 0.148 |
| R1-Distill-Qwen-7B | Continue (Adv.) | Unrelated | 0.5 | 0.981 ± 0.136 |
| R1-Distill-Qwen-7B | Continue (Adv.) | Unrelated | 0.7 | 0.980 ± 0.139 |
| R1-Distill-Qwen-7B | Continue (Adv.) | Unrelated | 0.9 | 0.989 ± 0.104 |
| EXAONE-Deep-32B | Ctn. Wrong Reasoning (Adv.) | | 0.1 | 0.993 ± 0.081 |
| EXAONE-Deep-32B | Ctn. Wrong Reasoning (Adv.) | | 0.3 | 0.996 ± 0.066 |
| EXAONE-Deep-32B | Ctn. Wrong Reasoning (Adv.) | | 0.5 | 0.995 ± 0.068 |
| EXAONE-Deep-32B | Ctn. Wrong Reasoning (Adv.) | | 0.7 | 0.996 ± 0.063 |
| EXAONE-Deep-32B | Ctn. Wrong Reasoning (Adv.) | | 0.9 | 0.997 ± 0.052 |
| Nemotron-Llama-3.1-Nano-8B | Ctn. Wrong Reasoning (Adv.) | | 0.1 | 0.991 ± 0.096 |
| Nemotron-Llama-3.1-Nano-8B | Ctn. Wrong Reasoning (Adv.) | | 0.3 | 0.989 ± 0.105 |
| Nemotron-Llama-3.1-Nano-8B | Ctn. Wrong Reasoning (Adv.) | | 0.5 | 0.992 ± 0.091 |
| Nemotron-Llama-3.1-Nano-8B | Ctn. Wrong Reasoning (Adv.) | | 0.7 | 0.994 ± 0.075 |
| Nemotron-Llama-3.1-Nano-8B | Ctn. Wrong Reasoning (Adv.) | | 0.9 | 0.998 ± 0.048 |
| Phi-4-Reasoning-Plus | Ctn. Wrong Reasoning (Adv.) | | 0.1 | 0.938 ± 0.241 |
| Phi-4-Reasoning-Plus | Ctn. Wrong Reasoning (Adv.) | | 0.3 | 0.944 ± 0.229 |
| Phi-4-Reasoning-Plus | Ctn. Wrong Reasoning (Adv.) | | 0.5 | 0.943 ± 0.232 |
| Phi-4-Reasoning-Plus | Ctn. Wrong Reasoning (Adv.) | | 0.7 | 0.943 ± 0.232 |
| Phi-4-Reasoning-Plus | Ctn. Wrong Reasoning (Adv.) | | 0.9 | 0.947 ± 0.224 |

Table 18: Robustness metrics (mean ± std) per model, intervention, and timestep on the Math domain.

| Model | Intervention | Timestep | Mean ± Std |
|---|---|---|---|
| QwQ-32B | Ctn. Wrong Reasoning (Adv.) | 0.1 | 0.994 ± 0.076 |
| QwQ-32B | Ctn. Wrong Reasoning (Adv.) | 0.3 | 0.991 ± 0.094 |
| QwQ-32B | Ctn. Wrong Reasoning (Adv.) | 0.5 | 0.993 ± 0.082 |
| QwQ-32B | Ctn. Wrong Reasoning (Adv.) | 0.7 | 0.989 ± 0.104 |
| QwQ-32B | Ctn. Wrong Reasoning (Adv.) | 0.9 | 0.987 ± 0.114 |
| R1-Distill-Llama-8B | Ctn. Wrong Reasoning (Adv.) | 0.1 | 0.917 ± 0.276 |
| R1-Distill-Llama-8B | Ctn. Wrong Reasoning (Adv.) | 0.3 | 0.958 ± 0.201 |
| R1-Distill-Llama-8B | Ctn. Wrong Reasoning (Adv.) | 0.5 | 0.967 ± 0.179 |
| R1-Distill-Llama-8B | Ctn. Wrong Reasoning (Adv.) | 0.7 | 0.980 ± 0.139 |
| R1-Distill-Llama-8B | Ctn. Wrong Reasoning (Adv.) | 0.9 | 0.990 ± 0.102 |
| R1-Distill-Qwen-1.5B | Ctn. Wrong Reasoning (Adv.) | 0.1 | 0.816 ± 0.388 |
| R1-Distill-Qwen-1.5B | Ctn. Wrong Reasoning (Adv.) | 0.3 | 0.851 ± 0.356 |
| R1-Distill-Qwen-1.5B | Ctn. Wrong Reasoning (Adv.) | 0.5 | 0.879 ± 0.326 |
| R1-Distill-Qwen-1.5B | Ctn. Wrong Reasoning (Adv.) | 0.7 | 0.899 ± 0.302 |
| R1-Distill-Qwen-1.5B | Ctn. Wrong Reasoning (Adv.) | 0.9 | 0.933 ± 0.249 |
| R1-Distill-Qwen-14B | Ctn. Wrong Reasoning (Adv.) | 0.1 | 0.988 ± 0.108 |
| R1-Distill-Qwen-14B | Ctn. Wrong Reasoning (Adv.) | 0.3 | 0.993 ± 0.081 |
| R1-Distill-Qwen-14B | Ctn. Wrong Reasoning (Adv.) | 0.5 | 0.990 ± 0.097 |
| R1-Distill-Qwen-14B | Ctn. Wrong Reasoning (Adv.) | 0.7 | 0.995 ± 0.071 |
| R1-Distill-Qwen-14B | Ctn. Wrong Reasoning (Adv.) | 0.9 | 0.997 ± 0.056 |
| R1-Distill-Qwen-32B | Ctn. Wrong Reasoning (Adv.) | 0.1 | 0.975 ± 0.155 |
| R1-Distill-Qwen-32B | Ctn. Wrong Reasoning (Adv.) | 0.3 | 0.989 ± 0.103 |
| R1-Distill-Qwen-32B | Ctn. Wrong Reasoning (Adv.) | 0.5 | 0.992 ± 0.090 |
| R1-Distill-Qwen-32B | Ctn. Wrong Reasoning (Adv.) | 0.7 | 0.993 ± 0.081 |
| R1-Distill-Qwen-32B | Ctn. Wrong Reasoning (Adv.) | 0.9 | 0.992 ± 0.090 |
| R1-Distill-Qwen-7B | Ctn. Wrong Reasoning (Adv.) | 0.1 | 0.968 ± 0.176 |
| R1-Distill-Qwen-7B | Ctn. Wrong Reasoning (Adv.) | 0.3 | 0.980 ± 0.140 |
| R1-Distill-Qwen-7B | Ctn. Wrong Reasoning (Adv.) | 0.5 | 0.981 ± 0.136 |
| R1-Distill-Qwen-7B | Ctn. Wrong Reasoning (Adv.) | 0.7 | 0.981 ± 0.138 |

Table 18: Robustness metrics (mean ± std) per model, intervention, and timestep on the Math domain.

| Model | Intervention | Timestep | Mean ± Std |
|---|---|---|---|
| R1-Distill-Qwen-7B | Ctn. Wrong Reasoning (Adv.) | 0.9 | 0.992 ± 0.087 |
| EXAONE-Deep-32B | Insert Random Characters (Neutral) | 0.1 | 0.996 ± 0.064 |
| EXAONE-Deep-32B | Insert Random Characters (Neutral) | 0.3 | 0.995 ± 0.068 |
| EXAONE-Deep-32B | Insert Random Characters (Neutral) | 0.5 | 0.994 ± 0.075 |
| EXAONE-Deep-32B | Insert Random Characters (Neutral) | 0.7 | 0.996 ± 0.064 |
| EXAONE-Deep-32B | Insert Random Characters (Neutral) | 0.9 | 0.998 ± 0.050 |
| Nemotron-Llama-3.1-Nano-8B | Insert Random Characters (Neutral) | 0.1 | 0.877 ± 0.329 |
| Nemotron-Llama-3.1-Nano-8B | Insert Random Characters (Neutral) | 0.3 | 0.910 ± 0.286 |
| Nemotron-Llama-3.1-Nano-8B | Insert Random Characters (Neutral) | 0.5 | 0.917 ± 0.276 |
| Nemotron-Llama-3.1-Nano-8B | Insert Random Characters (Neutral) | 0.7 | 0.941 ± 0.236 |
| Nemotron-Llama-3.1-Nano-8B | Insert Random Characters (Neutral) | 0.9 | 0.935 ± 0.246 |
| Phi-4-Reasoning-Plus | Insert Random Characters (Neutral) | 0.1 | 0.907 ± 0.290 |
| Phi-4-Reasoning-Plus | Insert Random Characters (Neutral) | 0.3 | 0.910 ± 0.286 |
| Phi-4-Reasoning-Plus | Insert Random Characters (Neutral) | 0.5 | 0.915 ± 0.280 |
| Phi-4-Reasoning-Plus | Insert Random Characters (Neutral) | 0.7 | 0.911 ± 0.285 |
| Phi-4-Reasoning-Plus | Insert Random Characters (Neutral) | 0.9 | 0.932 ± 0.252 |
| QwQ-32B | Insert Random Characters (Neutral) | 0.1 | 0.985 ± 0.120 |
| QwQ-32B | Insert Random Characters (Neutral) | 0.3 | 0.982 ± 0.134 |
| QwQ-32B | Insert Random Characters (Neutral) | 0.5 | 0.975 ± 0.156 |
| QwQ-32B | Insert Random Characters (Neutral) | 0.7 | 0.955 ± 0.208 |
| QwQ-32B | Insert Random Characters (Neutral) | 0.9 | 0.978 ± 0.147 |
| R1-Distill-Llama-8B | Insert Random Characters (Neutral) | 0.1 | 0.941 ± 0.235 |
| R1-Distill-Llama-8B | Insert Random Characters (Neutral) | 0.3 | 0.961 ± 0.194 |
| R1-Distill-Llama-8B | Insert Random Characters (Neutral) | 0.5 | 0.974 ± 0.159 |
| R1-Distill-Llama-8B | Insert Random Characters (Neutral) | 0.7 | 0.980 ± 0.141 |
| R1-Distill-Llama-8B | Insert Random Characters (Neutral) | 0.9 | 0.992 ± 0.087 |
| R1-Distill-Qwen-1.5B | Insert Random Characters (Neutral) | 0.1 | 0.802 ± 0.399 |
| R1-Distill-Qwen-1.5B | Insert Random Characters (Neutral) | 0.3 | 0.850 ± 0.357 |
| R1-Distill-Qwen-1.5B | Insert Random Characters (Neutral) | 0.5 | 0.867 ± 0.339 |

Table 18: Robustness metrics (mean ± std) per model, intervention, and timestep on the Math domain.

| Model | Intervention | Timestep | Mean ± Std |
|---|---|---|---|
| R1-Distill-Qwen-1.5B | Insert Random Characters (Neutral) | 0.7 | 0.886 ± 0.318 |
| R1-Distill-Qwen-1.5B | Insert Random Characters (Neutral) | 0.9 | 0.928 ± 0.258 |
| R1-Distill-Qwen-14B | Insert Random Characters (Neutral) | 0.1 | 0.988 ± 0.109 |
| R1-Distill-Qwen-14B | Insert Random Characters (Neutral) | 0.3 | 0.992 ± 0.087 |
| R1-Distill-Qwen-14B | Insert Random Characters (Neutral) | 0.5 | 0.991 ± 0.095 |
| R1-Distill-Qwen-14B | Insert Random Characters (Neutral) | 0.7 | 0.995 ± 0.072 |
| R1-Distill-Qwen-14B | Insert Random Characters (Neutral) | 0.9 | 0.996 ± 0.061 |
| R1-Distill-Qwen-32B | Insert Random Characters (Neutral) | 0.1 | 0.983 ± 0.130 |
| R1-Distill-Qwen-32B | Insert Random Characters (Neutral) | 0.3 | 0.988 ± 0.110 |
| R1-Distill-Qwen-32B | Insert Random Characters (Neutral) | 0.5 | 0.992 ± 0.091 |
| R1-Distill-Qwen-32B | Insert Random Characters (Neutral) | 0.7 | 0.994 ± 0.079 |
| R1-Distill-Qwen-32B | Insert Random Characters (Neutral) | 0.9 | 0.993 ± 0.084 |
| R1-Distill-Qwen-7B | Insert Random Characters (Neutral) | 0.1 | 0.890 ± 0.312 |
| R1-Distill-Qwen-7B | Insert Random Characters (Neutral) | 0.3 | 0.951 ± 0.216 |
| R1-Distill-Qwen-7B | Insert Random Characters (Neutral) | 0.5 | 0.946 ± 0.225 |
| R1-Distill-Qwen-7B | Insert Random Characters (Neutral) | 0.7 | 0.948 ± 0.222 |
| R1-Distill-Qwen-7B | Insert Random Characters (Neutral) | 0.9 | 0.955 ± 0.208 |
| EXAONE-Deep-32B | Insert Wrong Fact (Adv.) | 0.1 | 0.993 ± 0.081 |
| EXAONE-Deep-32B | Insert Wrong Fact (Adv.) | 0.3 | 0.996 ± 0.066 |
| EXAONE-Deep-32B | Insert Wrong Fact (Adv.) | 0.5 | 0.997 ± 0.056 |
| EXAONE-Deep-32B | Insert Wrong Fact (Adv.) | 0.7 | 0.997 ± 0.058 |
| EXAONE-Deep-32B | Insert Wrong Fact (Adv.) | 0.9 | 0.998 ± 0.050 |
| Nemotron-Llama-3.1-Nano-8B | Insert Wrong Fact (Adv.) | 0.1 | 0.991 ± 0.096 |
| Nemotron-Llama-3.1-Nano-8B | Insert Wrong Fact (Adv.) | 0.3 | 0.994 ± 0.077 |
| Nemotron-Llama-3.1-Nano-8B | Insert Wrong Fact (Adv.) | 0.5 | 0.995 ± 0.069 |
| Nemotron-Llama-3.1-Nano-8B | Insert Wrong Fact (Adv.) | 0.7 | 0.996 ± 0.064 |
| Nemotron-Llama-3.1-Nano-8B | Insert Wrong Fact (Adv.) | 0.9 | 0.998 ± 0.043 |
| Phi-4-Reasoning-Plus | Insert Wrong Fact (Adv.) | 0.1 | 0.933 ± 0.250 |
| Phi-4-Reasoning-Plus | Insert Wrong Fact (Adv.) | 0.3 | 0.937 ± 0.243 |
| Phi-4-Reasoning-Plus | Insert Wrong Fact (Adv.) | 0.5 | 0.941 ± 0.235 |
| Phi-4-Reasoning-Plus | Insert Wrong Fact (Adv.) | 0.7 | 0.941 ± 0.236 |
| Phi-4-Reasoning-Plus | Insert Wrong Fact (Adv.) | 0.9 | 0.943 ± 0.232 |
| QwQ-32B | Insert Wrong Fact (Adv.) | 0.1 | 0.996 ± 0.059 |
| QwQ-32B | Insert Wrong Fact (Adv.) | 0.3 | 0.988 ± 0.111 |
| QwQ-32B | Insert Wrong Fact (Adv.) | 0.5 | 0.992 ± 0.090 |
| QwQ-32B | Insert Wrong Fact (Adv.) | 0.7 | 0.995 ± 0.071 |

Table 18: Robustness metrics (mean ± std) per model, intervention, and timestep on the Math domain.

| Model | Intervention | Timestep | Mean ± Std |
|---|---|---|---|
| QwQ-32B | Insert Wrong Fact (Adv.) | 0.9 | 0.983 ± 0.129 |
| R1-Distill-Llama-8B | Insert Wrong Fact (Adv.) | 0.1 | 0.945 ± 0.228 |
| R1-Distill-Llama-8B | Insert Wrong Fact (Adv.) | 0.3 | 0.966 ± 0.182 |
| R1-Distill-Llama-8B | Insert Wrong Fact (Adv.) | 0.5 | 0.970 ± 0.171 |
| R1-Distill-Llama-8B | Insert Wrong Fact (Adv.) | 0.7 | 0.979 ± 0.142 |
| R1-Distill-Llama-8B | Insert Wrong Fact (Adv.) | 0.9 | 0.991 ± 0.093 |
| R1-Distill-Qwen-1.5B | Insert Wrong Fact (Adv.) | 0.1 | 0.829 ± 0.377 |
| R1-Distill-Qwen-1.5B | Insert Wrong Fact (Adv.) | 0.3 | 0.865 ± 0.342 |
| R1-Distill-Qwen-1.5B | Insert Wrong Fact (Adv.) | 0.5 | 0.885 ± 0.320 |
| R1-Distill-Qwen-1.5B | Insert Wrong Fact (Adv.) | 0.7 | 0.909 ± 0.288 |
| R1-Distill-Qwen-1.5B | Insert Wrong Fact (Adv.) | 0.9 | 0.937 ± 0.243 |
| R1-Distill-Qwen-14B | Insert Wrong Fact (Adv.) | 0.1 | 0.990 ± 0.102 |
| R1-Distill-Qwen-14B | Insert Wrong Fact (Adv.) | 0.3 | 0.992 ± 0.091 |
| R1-Distill-Qwen-14B | Insert Wrong Fact (Adv.) | 0.5 | 0.994 ± 0.075 |
| R1-Distill-Qwen-14B | Insert Wrong Fact (Adv.) | 0.7 | 0.992 ± 0.090 |
| R1-Distill-Qwen-14B | Insert Wrong Fact (Adv.) | 0.9 | 0.997 ± 0.056 |
| R1-Distill-Qwen-32B | Insert Wrong Fact (Adv.) | 0.1 | 0.986 ± 0.119 |
| R1-Distill-Qwen-32B | Insert Wrong Fact (Adv.) | 0.3 | 0.986 ± 0.116 |
| R1-Distill-Qwen-32B | Insert Wrong Fact (Adv.) | 0.5 | 0.993 ± 0.081 |
| R1-Distill-Qwen-32B | Insert Wrong Fact (Adv.) | 0.7 | 0.991 ± 0.093 |
| R1-Distill-Qwen-32B | Insert Wrong Fact (Adv.) | 0.9 | 0.995 ± 0.071 |
| R1-Distill-Qwen-7B | Insert Wrong Fact (Adv.) | 0.1 | 0.984 ± 0.125 |
| R1-Distill-Qwen-7B | Insert Wrong Fact (Adv.) | 0.3 | 0.980 ± 0.141 |
| R1-Distill-Qwen-7B | Insert Wrong Fact (Adv.) | 0.5 | 0.980 ± 0.139 |
| R1-Distill-Qwen-7B | Insert Wrong Fact (Adv.) | 0.7 | 0.985 ± 0.120 |
| R1-Distill-Qwen-7B | Insert Wrong Fact (Adv.) | 0.9 | 0.989 ± 0.106 |
| EXAONE-Deep-32B | Rewrite Trace (Benign) | 0.1 | 0.995 ± 0.071 |
| EXAONE-Deep-32B | Rewrite Trace (Benign) | 0.3 | 0.996 ± 0.061 |
| EXAONE-Deep-32B | Rewrite Trace (Benign) | 0.5 | 0.995 ± 0.069 |
| EXAONE-Deep-32B | Rewrite Trace (Benign) | 0.7 | 0.995 ± 0.069 |
| EXAONE-Deep-32B | Rewrite Trace (Benign) | 0.9 | 0.997 ± 0.056 |
| Nemotron-Llama-3.1-Nano-8B | Rewrite Trace (Benign) | 0.1 | 0.804 ± 0.397 |
| Nemotron-Llama-3.1-Nano-8B | Rewrite Trace (Benign) | 0.3 | 0.880 ± 0.324 |
| Nemotron-Llama-3.1-Nano-8B | Rewrite Trace (Benign) | 0.5 | 0.906 ± 0.292 |
| Nemotron-Llama-3.1-Nano-8B | Rewrite Trace (Benign) | 0.7 | 0.921 ± 0.270 |
| Nemotron-Llama-3.1-Nano-8B | Rewrite Trace (Benign) | 0.9 | 0.944 ± 0.230 |
| Phi-4-Reasoning-Plus | Rewrite Trace (Benign) | 0.1 | 0.913 ± 0.282 |
| Phi-4-Reasoning-Plus | Rewrite Trace (Benign) | 0.3 | 0.904 ± 0.295 |
| Phi-4-Reasoning-Plus | Rewrite Trace (Benign) | 0.5 | 0.903 ± 0.295 |
| Phi-4-Reasoning-Plus | Rewrite Trace (Benign) | 0.7 | 0.926 ± 0.261 |
| Phi-4-Reasoning-Plus | Rewrite Trace (Benign) | 0.9 | 0.918 ± 0.274 |
| QwQ-32B | Rewrite Trace (Benign) | 0.1 | 0.991 ± 0.092 |
| QwQ-32B | Rewrite Trace (Benign) | 0.3 | 0.990 ± 0.099 |
| QwQ-32B | Rewrite Trace (Benign) | 0.5 | 0.993 ± 0.081 |
| QwQ-32B | Rewrite Trace (Benign) | 0.7 | 0.989 ± 0.105 |
| QwQ-32B | Rewrite Trace (Benign) | 0.9 | 0.991 ± 0.094 |
| R1-Distill-Llama-8B | Rewrite Trace (Benign) | 0.1 | 0.779 ± 0.415 |
| R1-Distill-Llama-8B | Rewrite Trace (Benign) | 0.3 | 0.827 ± 0.379 |

Table 18: Robustness metrics (mean ± std) per model, intervention, and timestep on the Math domain.

| Model | Intervention | Timestep | Mean ± Std |
|---|---|---|---|
| R1-Distill-Llama-8B | Rewrite Trace (Benign) | 0.5 | 0.877 ± 0.329 |
| R1-Distill-Llama-8B | Rewrite Trace (Benign) | 0.7 | 0.930 ± 0.255 |
| R1-Distill-Llama-8B | Rewrite Trace (Benign) | 0.9 | 0.962 ± 0.190 |
| R1-Distill-Qwen-1.5B | Rewrite Trace (Benign) | 0.1 | 0.791 ± 0.407 |
| R1-Distill-Qwen-1.5B | Rewrite Trace (Benign) | 0.3 | 0.818 ± 0.386 |
| R1-Distill-Qwen-1.5B | Rewrite Trace (Benign) | 0.5 | 0.862 ± 0.345 |
| R1-Distill-Qwen-1.5B | Rewrite Trace (Benign) | 0.7 | 0.904 ± 0.294 |
| R1-Distill-Qwen-1.5B | Rewrite Trace (Benign) | 0.9 | 0.945 ± 0.228 |
| R1-Distill-Qwen-14B | Rewrite Trace (Benign) | 0.1 | 0.813 ± 0.390 |
| R1-Distill-Qwen-14B | Rewrite Trace (Benign) | 0.3 | 0.879 ± 0.326 |
| R1-Distill-Qwen-14B | Rewrite Trace (Benign) | 0.5 | 0.915 ± 0.278 |
| R1-Distill-Qwen-14B | Rewrite Trace (Benign) | 0.7 | 0.940 ± 0.237 |
| R1-Distill-Qwen-14B | Rewrite Trace (Benign) | 0.9 | 0.969 ± 0.173 |
| R1-Distill-Qwen-32B | Rewrite Trace (Benign) | 0.1 | 0.832 ± 0.374 |
| R1-Distill-Qwen-32B | Rewrite Trace (Benign) | 0.3 | 0.890 ± 0.313 |
| R1-Distill-Qwen-32B | Rewrite Trace (Benign) | 0.5 | 0.924 ± 0.265 |
| R1-Distill-Qwen-32B | Rewrite Trace (Benign) | 0.7 | 0.945 ± 0.228 |
| R1-Distill-Qwen-32B | Rewrite Trace (Benign) | 0.9 | 0.971 ± 0.168 |
| R1-Distill-Qwen-7B | Rewrite Trace (Benign) | 0.1 | 0.863 ± 0.343 |
| R1-Distill-Qwen-7B | Rewrite Trace (Benign) | 0.3 | 0.899 ± 0.302 |
| R1-Distill-Qwen-7B | Rewrite Trace (Benign) | 0.5 | 0.942 ± 0.233 |
| R1-Distill-Qwen-7B | Rewrite Trace (Benign) | 0.7 | 0.954 ± 0.209 |
| R1-Distill-Qwen-7B | Rewrite Trace (Benign) | 0.9 | 0.973 ± 0.162 |

Table 19: Robustness metrics (mean ± std) per model, intervention, and timestep on the Science domain.

| Model | Intervention | Timestep | Mean ± Std |
|---|---|---|---|
| EXAONE-Deep-32B | Add Random Text (Neutral) | 0.1 | 0.923 ± 0.266 |
| EXAONE-Deep-32B | Add Random Text (Neutral) | 0.3 | 0.927 ± 0.260 |
| EXAONE-Deep-32B | Add Random Text (Neutral) | 0.5 | 0.942 ± 0.235 |
| EXAONE-Deep-32B | Add Random Text (Neutral) | 0.7 | 0.948 ± 0.222 |
| EXAONE-Deep-32B | Add Random Text (Neutral) | 0.9 | 0.964 ± 0.186 |
| Nemotron-Llama-3.1-Nano-8B | Add Random Text (Neutral) | 0.1 | 0.890 ± 0.313 |
| Nemotron-Llama-3.1-Nano-8B | Add Random Text (Neutral) | 0.3 | 0.896 ± 0.306 |
| Nemotron-Llama-3.1-Nano-8B | Add Random Text (Neutral) | 0.5 | 0.898 ± 0.303 |
| Nemotron-Llama-3.1-Nano-8B | Add Random Text (Neutral) | 0.7 | 0.908 ± 0.289 |
| Nemotron-Llama-3.1-Nano-8B | Add Random Text (Neutral) | 0.9 | 0.916 ± 0.277 |
| Phi-4-Reasoning-Plus | Add Random Text (Neutral) | 0.1 | 0.850 ± 0.357 |

Continued on next page

Table 19: Robustness metrics (mean ± std) per model, intervention, and timestep on the Science domain.

| Model | Intervention | Timestep | Mean ± Std |
|---|---|---|---|
| Phi-4-Reasoning-Plus | Add Random Text (Neutral) | 0.3 | 0.848 ± 0.359 |
| Phi-4-Reasoning-Plus | Add Random Text (Neutral) | 0.5 | 0.860 ± 0.347 |
| Phi-4-Reasoning-Plus | Add Random Text (Neutral) | 0.7 | 0.865 ± 0.341 |
| Phi-4-Reasoning-Plus | Add Random Text (Neutral) | 0.9 | 0.860 ± 0.347 |
| QwQ-32B | Add Random Text (Neutral) | 0.1 | 0.903 ± 0.297 |
| QwQ-32B | Add Random Text (Neutral) | 0.3 | 0.900 ± 0.300 |
| QwQ-32B | Add Random Text (Neutral) | 0.5 | 0.912 ± 0.284 |
| QwQ-32B | Add Random Text (Neutral) | 0.7 | 0.915 ± 0.279 |
| QwQ-32B | Add Random Text (Neutral) | 0.9 | 0.908 ± 0.289 |
| R1-Distill-Llama-8B | Add Random Text (Neutral) | 0.1 | 0.766 ± 0.423 |
| R1-Distill-Llama-8B | Add Random Text (Neutral) | 0.3 | 0.793 ± 0.405 |
| R1-Distill-Llama-8B | Add Random Text (Neutral) | 0.5 | 0.807 ± 0.394 |
| R1-Distill-Llama-8B | Add Random Text (Neutral) | 0.7 | 0.850 ± 0.357 |
| R1-Distill-Llama-8B | Add Random Text (Neutral) | 0.9 | 0.874 ± 0.332 |
| R1-Distill-Qwen-1.5B | Add Random Text (Neutral) | 0.1 | 0.702 ± 0.457 |
| R1-Distill-Qwen-1.5B | Add Random Text (Neutral) | 0.3 | 0.690 ± 0.462 |
| R1-Distill-Qwen-1.5B | Add Random Text (Neutral) | 0.5 | 0.737 ± 0.440 |
| R1-Distill-Qwen-1.5B | Add Random Text (Neutral) | 0.7 | 0.744 ± 0.437 |
| R1-Distill-Qwen-1.5B | Add Random Text (Neutral) | 0.9 | 0.768 ± 0.422 |
| R1-Distill-Qwen-14B | Add Random Text (Neutral) | 0.1 | 0.910 ± 0.286 |
| R1-Distill-Qwen-14B | Add Random Text (Neutral) | 0.3 | 0.913 ± 0.281 |
| R1-Distill-Qwen-14B | Add Random Text (Neutral) | 0.5 | 0.915 ± 0.279 |
| R1-Distill-Qwen-14B | Add Random Text (Neutral) | 0.7 | 0.924 ± 0.265 |
| R1-Distill-Qwen-14B | Add Random Text (Neutral) | 0.9 | 0.930 ± 0.256 |
| R1-Distill-Qwen-32B | Add Random Text (Neutral) | 0.1 | 0.919 ± 0.273 |
| R1-Distill-Qwen-32B | Add Random Text (Neutral) | 0.3 | 0.923 ± 0.267 |
| R1-Distill-Qwen-32B | Add Random Text (Neutral) | 0.5 | 0.899 ± 0.301 |
| R1-Distill-Qwen-32B | Add Random Text (Neutral) | 0.7 | 0.910 ± 0.286 |
| R1-Distill-Qwen-32B | Add Random Text (Neutral) | 0.9 | 0.916 ± 0.277 |

Table 19: Robustness metrics (mean ± std) per model, intervention, and timestep on the Science domain.

| Model | Intervention | Timestep | Mean ± Std |
|---|---|---|---|
| R1-Distill-Qwen-7B | Add Random Text (Neutral) | 0.1 | 0.871 ± 0.335 |
| R1-Distill-Qwen-7B | Add Random Text (Neutral) | 0.3 | 0.882 ± 0.323 |
| R1-Distill-Qwen-7B | Add Random Text (Neutral) | 0.5 | 0.872 ± 0.334 |
| R1-Distill-Qwen-7B | Add Random Text (Neutral) | 0.7 | 0.871 ± 0.335 |
| R1-Distill-Qwen-7B | Add Random Text (Neutral) | 0.9 | 0.878 ± 0.328 |
| EXAONE-Deep-32B | Complete Step (Benign) | 0.1 | 0.983 ± 0.130 |
| EXAONE-Deep-32B | Complete Step (Benign) | 0.3 | 0.976 ± 0.152 |
| EXAONE-Deep-32B | Complete Step (Benign) | 0.5 | 0.971 ± 0.168 |
| EXAONE-Deep-32B | Complete Step (Benign) | 0.7 | 0.976 ± 0.154 |
| EXAONE-Deep-32B | Complete Step (Benign) | 0.9 | 0.985 ± 0.122 |
| Nemotron-Llama-3.1-Nano-8B | Complete Step (Benign) | 0.1 | 0.892 ± 0.310 |
| Nemotron-Llama-3.1-Nano-8B | Complete Step (Benign) | 0.3 | 0.902 ± 0.298 |
| Nemotron-Llama-3.1-Nano-8B | Complete Step (Benign) | 0.5 | 0.916 ± 0.278 |
| Nemotron-Llama-3.1-Nano-8B | Complete Step (Benign) | 0.7 | 0.920 ± 0.271 |
| Nemotron-Llama-3.1-Nano-8B | Complete Step (Benign) | 0.9 | 0.937 ± 0.244 |
| Phi-4-Reasoning-Plus | Complete Step (Benign) | 0.1 | 0.865 ± 0.341 |
| Phi-4-Reasoning-Plus | Complete Step (Benign) | 0.3 | 0.866 ± 0.340 |
| Phi-4-Reasoning-Plus | Complete Step (Benign) | 0.5 | 0.859 ± 0.348 |
| Phi-4-Reasoning-Plus | Complete Step (Benign) | 0.7 | 0.872 ± 0.334 |
| Phi-4-Reasoning-Plus | Complete Step (Benign) | 0.9 | 0.878 ± 0.328 |
| QwQ-32B | Complete Step (Benign) | 0.1 | 0.930 ± 0.256 |
| QwQ-32B | Complete Step (Benign) | 0.3 | 0.923 ± 0.267 |
| QwQ-32B | Complete Step (Benign) | 0.5 | 0.924 ± 0.265 |
| QwQ-32B | Complete Step (Benign) | 0.7 | 0.931 ± 0.254 |
| QwQ-32B | Complete Step (Benign) | 0.9 | 0.917 ± 0.276 |
| R1-Distill-Llama-8B | Complete Step (Benign) | 0.1 | 0.798 ± 0.402 |
| R1-Distill-Llama-8B | Complete Step (Benign) | 0.3 | 0.801 ± 0.399 |
| R1-Distill-Llama-8B | Complete Step (Benign) | 0.5 | 0.811 ± 0.392 |
| R1-Distill-Llama-8B | Complete Step (Benign) | 0.7 | 0.839 ± 0.367 |
| R1-Distill-Llama-8B | Complete Step (Benign) | 0.9 | 0.862 ± 0.345 |
| R1-Distill-Qwen-1.5B | Complete Step (Benign) | 0.1 | 0.782 ± 0.413 |
| R1-Distill-Qwen-1.5B | Complete Step (Benign) | 0.3 | 0.778 ± 0.416 |
| R1-Distill-Qwen-1.5B | Complete Step (Benign) | 0.5 | 0.821 ± 0.383 |
| R1-Distill-Qwen-1.5B | Complete Step (Benign) | 0.7 | 0.831 ± 0.375 |
| R1-Distill-Qwen-1.5B | Complete Step (Benign) | 0.9 | 0.863 ± 0.344 |
| R1-Distill-Qwen-14B | Complete Step (Benign) | 0.1 | 0.909 ± 0.288 |
| R1-Distill-Qwen-14B | Complete Step (Benign) | 0.3 | 0.912 ± 0.284 |
| R1-Distill-Qwen-14B | Complete Step (Benign) | 0.5 | 0.915 ± 0.280 |
| R1-Distill-Qwen-14B | Complete Step (Benign) | 0.7 | 0.931 ± 0.254 |
| R1-Distill-Qwen-14B | Complete Step (Benign) | 0.9 | 0.918 ± 0.275 |
| R1-Distill-Qwen-32B | Complete Step (Benign) | 0.1 | 0.916 ± 0.277 |
| R1-Distill-Qwen-32B | Complete Step (Benign) | 0.3 | 0.913 ± 0.282 |
| R1-Distill-Qwen-32B | Complete Step (Benign) | 0.5 | 0.916 ± 0.278 |

Continued on next page

Table 19: Robustness metrics (mean ± std) per model, intervention, and timestep on the Science domain.

| Model | Intervention | | Timestep | Mean ± Std |
|---|---|---|---|---|
| R1-Distill-Qwen-32B | Complete Step (Benign) | | 0.7 | 0.903 ± 0.296 |
| R1-Distill-Qwen-32B | Complete Step (Benign) | | 0.9 | 0.921 ± 0.270 |
| R1-Distill-Qwen-7B | Complete Step (Benign) | | 0.1 | 0.898 ± 0.302 |
| R1-Distill-Qwen-7B | Complete Step (Benign) | | 0.3 | 0.914 ± 0.280 |
| R1-Distill-Qwen-7B | Complete Step (Benign) | | 0.5 | 0.881 ± 0.324 |
| R1-Distill-Qwen-7B | Complete Step (Benign) | | 0.7 | 0.894 ± 0.308 |
| R1-Distill-Qwen-7B | Complete Step (Benign) | | 0.9 | 0.906 ± 0.292 |
| EXAONE-Deep-32B | Continue (Adv.) | Unrelated | 0.1 | 0.516 ± 0.500 |
| EXAONE-Deep-32B | Continue (Adv.) | Unrelated | 0.3 | 0.648 ± 0.478 |
| EXAONE-Deep-32B | Continue (Adv.) | Unrelated | 0.5 | 0.754 ± 0.431 |
| EXAONE-Deep-32B | Continue (Adv.) | Unrelated | 0.7 | 0.826 ± 0.379 |
| EXAONE-Deep-32B | Continue (Adv.) | Unrelated | 0.9 | 0.859 ± 0.348 |
| Nemotron-Llama-3.1-Nano-8B | Continue (Adv.) | Unrelated | 0.1 | 0.720 ± 0.449 |
| Nemotron-Llama-3.1-Nano-8B | Continue (Adv.) | Unrelated | 0.3 | 0.790 ± 0.407 |
| Nemotron-Llama-3.1-Nano-8B | Continue (Adv.) | Unrelated | 0.5 | 0.856 ± 0.352 |
| Nemotron-Llama-3.1-Nano-8B | Continue (Adv.) | Unrelated | 0.7 | 0.878 ± 0.328 |
| Nemotron-Llama-3.1-Nano-8B | Continue (Adv.) | Unrelated | 0.9 | 0.914 ± 0.280 |
| Phi-4-Reasoning-Plus | Continue (Adv.) | Unrelated | 0.1 | 0.869 ± 0.337 |
| Phi-4-Reasoning-Plus | Continue (Adv.) | Unrelated | 0.3 | 0.864 ± 0.343 |
| Phi-4-Reasoning-Plus | Continue (Adv.) | Unrelated | 0.5 | 0.878 ± 0.328 |
| Phi-4-Reasoning-Plus | Continue (Adv.) | Unrelated | 0.7 | 0.883 ± 0.321 |
| Phi-4-Reasoning-Plus | Continue (Adv.) | Unrelated | 0.9 | 0.858 ± 0.349 |
| QwQ-32B | Continue (Adv.) | Unrelated | 0.1 | 0.915 ± 0.279 |
| QwQ-32B | Continue (Adv.) | Unrelated | 0.3 | 0.926 ± 0.261 |
| QwQ-32B | Continue (Adv.) | Unrelated | 0.5 | 0.914 ± 0.280 |
| QwQ-32B | Continue (Adv.) | Unrelated | 0.7 | 0.926 ± 0.262 |
| QwQ-32B | Continue (Adv.) | Unrelated | 0.9 | 0.923 ± 0.267 |
| R1-Distill-Llama-8B | Continue (Adv.) | Unrelated | 0.1 | 0.760 ± 0.427 |
| R1-Distill-Llama-8B | Continue (Adv.) | Unrelated | 0.3 | 0.791 ± 0.407 |
| R1-Distill-Llama-8B | Continue (Adv.) | Unrelated | 0.5 | 0.824 ± 0.381 |
| R1-Distill-Llama-8B | Continue (Adv.) | Unrelated | 0.7 | 0.834 ± 0.372 |
| R1-Distill-Llama-8B | Continue (Adv.) | Unrelated | 0.9 | 0.873 ± 0.333 |

Continued on next page

Table 19: Robustness metrics (mean ± std) per model, intervention, and timestep on the Science domain.

| Model | Intervention | | Timestep | Mean ± Std |
|---|---|---|---|---|
| R1-Distill-Qwen-1.5B | Continue (Adv.) | Unrelated | 0.1 | 0.753 ± 0.431 |
| R1-Distill-Qwen-1.5B | Continue (Adv.) | Unrelated | 0.3 | 0.771 ± 0.420 |
| R1-Distill-Qwen-1.5B | Continue (Adv.) | Unrelated | 0.5 | 0.802 ± 0.399 |
| R1-Distill-Qwen-1.5B | Continue (Adv.) | Unrelated | 0.7 | 0.798 ± 0.402 |
| R1-Distill-Qwen-1.5B | Continue (Adv.) | Unrelated | 0.9 | 0.827 ± 0.378 |
| R1-Distill-Qwen-14B | Continue (Adv.) | Unrelated | 0.1 | 0.859 ± 0.348 |
| R1-Distill-Qwen-14B | Continue (Adv.) | Unrelated | 0.3 | 0.902 ± 0.297 |
| R1-Distill-Qwen-14B | Continue (Adv.) | Unrelated | 0.5 | 0.911 ± 0.285 |
| R1-Distill-Qwen-14B | Continue (Adv.) | Unrelated | 0.7 | 0.920 ± 0.271 |
| R1-Distill-Qwen-14B | Continue (Adv.) | Unrelated | 0.9 | 0.918 ± 0.274 |
| R1-Distill-Qwen-32B | Continue (Adv.) | Unrelated | 0.1 | 0.896 ± 0.305 |
| R1-Distill-Qwen-32B | Continue (Adv.) | Unrelated | 0.3 | 0.917 ± 0.276 |
| R1-Distill-Qwen-32B | Continue (Adv.) | Unrelated | 0.5 | 0.908 ± 0.289 |
| R1-Distill-Qwen-32B | Continue (Adv.) | Unrelated | 0.7 | 0.913 ± 0.281 |
| R1-Distill-Qwen-32B | Continue (Adv.) | Unrelated | 0.9 | 0.922 ± 0.268 |
| R1-Distill-Qwen-7B | Continue (Adv.) | Unrelated | 0.1 | 0.904 ± 0.295 |
| R1-Distill-Qwen-7B | Continue (Adv.) | Unrelated | 0.3 | 0.900 ± 0.300 |
| R1-Distill-Qwen-7B | Continue (Adv.) | Unrelated | 0.5 | 0.890 ± 0.313 |
| R1-Distill-Qwen-7B | Continue (Adv.) | Unrelated | 0.7 | 0.900 ± 0.299 |
| R1-Distill-Qwen-7B | Continue (Adv.) | Unrelated | 0.9 | 0.903 ± 0.296 |
| EXAONE-Deep-32B | Ctn. (Adv.) | Wrong Reasoning | 0.1 | 0.976 ± 0.154 |
| EXAONE-Deep-32B | Ctn. (Adv.) | Wrong Reasoning | 0.3 | 0.981 ± 0.138 |
| EXAONE-Deep-32B | Ctn. (Adv.) | Wrong Reasoning | 0.5 | 0.977 ± 0.149 |
| EXAONE-Deep-32B | Ctn. (Adv.) | Wrong Reasoning | 0.7 | 0.983 ± 0.130 |
| EXAONE-Deep-32B | Ctn. (Adv.) | Wrong Reasoning | 0.9 | 0.990 ± 0.101 |
| Nemotron-Llama-3.1-Nano-8B | Ctn. (Adv.) | Wrong Reasoning | 0.1 | 0.893 ± 0.309 |
| Nemotron-Llama-3.1-Nano-8B | Ctn. (Adv.) | Wrong Reasoning | 0.3 | 0.907 ± 0.290 |
| Nemotron-Llama-3.1-Nano-8B | Ctn. (Adv.) | Wrong Reasoning | 0.5 | 0.914 ± 0.280 |
| Nemotron-Llama-3.1-Nano-8B | Ctn. (Adv.) | Wrong Reasoning | 0.7 | 0.922 ± 0.269 |

Table 19: Robustness metrics (mean ± std) per model, intervention, and timestep on the Science domain.

| Model | Intervention | | Timestep | Mean ± Std |
|---|---|---|---|---|
| Nemotron-Llama-3.1-Nano-8B | Ctn. Wrong Reasoning (Adv.) | | 0.9 | 0.924 ± 0.265 |
| Phi-4-Reasoning-Plus | Ctn. Wrong Reasoning (Adv.) | | 0.1 | 0.882 ± 0.323 |
| Phi-4-Reasoning-Plus | Ctn. Wrong Reasoning (Adv.) | | 0.3 | 0.883 ± 0.321 |
| Phi-4-Reasoning-Plus | Ctn. Wrong Reasoning (Adv.) | | 0.5 | 0.881 ± 0.324 |
| Phi-4-Reasoning-Plus | Ctn. Wrong Reasoning (Adv.) | | 0.7 | 0.891 ± 0.311 |
| Phi-4-Reasoning-Plus | Ctn. Wrong Reasoning (Adv.) | | 0.9 | 0.877 ± 0.329 |
| QwQ-32B | Ctn. Wrong Reasoning (Adv.) | | 0.1 | 0.923 ± 0.266 |
| QwQ-32B | Ctn. Wrong Reasoning (Adv.) | | 0.3 | 0.926 ± 0.261 |
| QwQ-32B | Ctn. Wrong Reasoning (Adv.) | | 0.5 | 0.917 ± 0.276 |
| QwQ-32B | Ctn. Wrong Reasoning (Adv.) | | 0.7 | 0.926 ± 0.261 |
| QwQ-32B | Ctn. Wrong Reasoning (Adv.) | | 0.9 | 0.919 ± 0.273 |
| R1-Distill-Llama-8B | Ctn. Wrong Reasoning (Adv.) | | 0.1 | 0.755 ± 0.430 |
| R1-Distill-Llama-8B | Ctn. Wrong Reasoning (Adv.) | | 0.3 | 0.792 ± 0.406 |
| R1-Distill-Llama-8B | Ctn. Wrong Reasoning (Adv.) | | 0.5 | 0.824 ± 0.381 |
| R1-Distill-Llama-8B | Ctn. Wrong Reasoning (Adv.) | | 0.7 | 0.825 ± 0.380 |
| R1-Distill-Llama-8B | Ctn. Wrong Reasoning (Adv.) | | 0.9 | 0.865 ± 0.341 |
| R1-Distill-Qwen-1.5B | Ctn. Wrong Reasoning (Adv.) | | 0.1 | 0.737 ± 0.440 |
| R1-Distill-Qwen-1.5B | Ctn. Wrong Reasoning (Adv.) | | 0.3 | 0.769 ± 0.421 |
| R1-Distill-Qwen-1.5B | Ctn. Wrong Reasoning (Adv.) | | 0.5 | 0.793 ± 0.405 |
| R1-Distill-Qwen-1.5B | Ctn. Wrong Reasoning (Adv.) | | 0.7 | 0.807 ± 0.394 |
| R1-Distill-Qwen-1.5B | Ctn. Wrong Reasoning (Adv.) | | 0.9 | 0.838 ± 0.368 |
| R1-Distill-Qwen-14B | Ctn. Wrong Reasoning (Adv.) | | 0.1 | 0.909 ± 0.288 |
| R1-Distill-Qwen-14B | Ctn. Wrong Reasoning (Adv.) | | 0.3 | 0.916 ± 0.277 |
| R1-Distill-Qwen-14B | Ctn. Wrong Reasoning (Adv.) | | 0.5 | 0.931 ± 0.253 |
| R1-Distill-Qwen-14B | Ctn. Wrong Reasoning (Adv.) | | 0.7 | 0.930 ± 0.256 |
| R1-Distill-Qwen-14B | Ctn. Wrong Reasoning (Adv.) | | 0.9 | 0.927 ± 0.260 |
| R1-Distill-Qwen-32B | Ctn. Wrong Reasoning (Adv.) | | 0.1 | 0.914 ± 0.280 |
| R1-Distill-Qwen-32B | Ctn. Wrong Reasoning (Adv.) | | 0.3 | 0.918 ± 0.274 |
| R1-Distill-Qwen-32B | Ctn. Wrong Reasoning (Adv.) | | 0.5 | 0.904 ± 0.295 |

Table 19: Robustness metrics (mean ± std) per model, intervention, and timestep on the Science domain.

| Model | Intervention | Timestep | Mean ± Std |
|---|---|---|---|
| R1-Distill-Qwen-32B | Ctn. Wrong Reasoning (Adv.) | 0.7 | 0.915 ± 0.280 |
| R1-Distill-Qwen-32B | Ctn. Wrong Reasoning (Adv.) | 0.9 | 0.923 ± 0.267 |
| R1-Distill-Qwen-7B | Ctn. Wrong Reasoning (Adv.) | 0.1 | 0.893 ± 0.309 |
| R1-Distill-Qwen-7B | Ctn. Wrong Reasoning (Adv.) | 0.3 | 0.896 ± 0.306 |
| R1-Distill-Qwen-7B | Ctn. Wrong Reasoning (Adv.) | 0.5 | 0.894 ± 0.307 |
| R1-Distill-Qwen-7B | Ctn. Wrong Reasoning (Adv.) | 0.7 | 0.905 ± 0.293 |
| R1-Distill-Qwen-7B | Ctn. Wrong Reasoning (Adv.) | 0.9 | 0.910 ± 0.286 |
| EXAONE-Deep-32B | Insert Random Characters (Neutral) | 0.1 | 0.978 ± 0.147 |
| EXAONE-Deep-32B | Insert Random Characters (Neutral) | 0.3 | 0.979 ± 0.142 |
| EXAONE-Deep-32B | Insert Random Characters (Neutral) | 0.5 | 0.979 ± 0.142 |
| EXAONE-Deep-32B | Insert Random Characters (Neutral) | 0.7 | 0.980 ± 0.140 |
| EXAONE-Deep-32B | Insert Random Characters (Neutral) | 0.9 | 0.988 ± 0.108 |
| Nemotron-Llama-3.1-Nano-8B | Insert Random Characters (Neutral) | 0.1 | 0.839 ± 0.368 |
| Nemotron-Llama-3.1-Nano-8B | Insert Random Characters (Neutral) | 0.3 | 0.866 ± 0.340 |
| Nemotron-Llama-3.1-Nano-8B | Insert Random Characters (Neutral) | 0.5 | 0.879 ± 0.326 |
| Nemotron-Llama-3.1-Nano-8B | Insert Random Characters (Neutral) | 0.7 | 0.891 ± 0.311 |
| Nemotron-Llama-3.1-Nano-8B | Insert Random Characters (Neutral) | 0.9 | 0.891 ± 0.311 |
| Phi-4-Reasoning-Plus | Insert Random Characters (Neutral) | 0.1 | 0.854 ± 0.353 |
| Phi-4-Reasoning-Plus | Insert Random Characters (Neutral) | 0.3 | 0.852 ± 0.355 |
| Phi-4-Reasoning-Plus | Insert Random Characters (Neutral) | 0.5 | 0.860 ± 0.347 |
| Phi-4-Reasoning-Plus | Insert Random Characters (Neutral) | 0.7 | 0.859 ± 0.348 |
| Phi-4-Reasoning-Plus | Insert Random Characters (Neutral) | 0.9 | 0.880 ± 0.324 |
| QwQ-32B | Insert Random Characters (Neutral) | 0.1 | 0.931 ± 0.253 |
| QwQ-32B | Insert Random Characters (Neutral) | 0.3 | 0.925 ± 0.263 |
| QwQ-32B | Insert Random Characters (Neutral) | 0.5 | 0.920 ± 0.271 |
| QwQ-32B | Insert Random Characters (Neutral) | 0.7 | 0.920 ± 0.271 |
| QwQ-32B | Insert Random Characters (Neutral) | 0.9 | 0.915 ± 0.279 |
| R1-Distill-Llama-8B | Insert Random Characters (Neutral) | 0.1 | 0.753 ± 0.431 |
| R1-Distill-Llama-8B | Insert Random Characters (Neutral) | 0.3 | 0.779 ± 0.415 |

Table 19: Robustness metrics (mean ± std) per model, intervention, and timestep on the Science domain.

| Model | Intervention | Timestep | Mean ± Std |
|---|---|---|---|
| R1-Distill-Llama-8B | Insert Random Characters (Neutral) | 0.5 | 0.820 ± 0.384 |
| R1-Distill-Llama-8B | Insert Random Characters (Neutral) | 0.7 | 0.844 ± 0.363 |
| R1-Distill-Llama-8B | Insert Random Characters (Neutral) | 0.9 | 0.871 ± 0.336 |
| R1-Distill-Qwen-1.5B | Insert Random Characters (Neutral) | 0.1 | 0.745 ± 0.436 |
| R1-Distill-Qwen-1.5B | Insert Random Characters (Neutral) | 0.3 | 0.778 ± 0.415 |
| R1-Distill-Qwen-1.5B | Insert Random Characters (Neutral) | 0.5 | 0.805 ± 0.396 |
| R1-Distill-Qwen-1.5B | Insert Random Characters (Neutral) | 0.7 | 0.810 ± 0.393 |
| R1-Distill-Qwen-1.5B | Insert Random Characters (Neutral) | 0.9 | 0.837 ± 0.369 |
| R1-Distill-Qwen-14B | Insert Random Characters (Neutral) | 0.1 | 0.905 ± 0.293 |
| R1-Distill-Qwen-14B | Insert Random Characters (Neutral) | 0.3 | 0.904 ± 0.295 |
| R1-Distill-Qwen-14B | Insert Random Characters (Neutral) | 0.5 | 0.929 ± 0.257 |
| R1-Distill-Qwen-14B | Insert Random Characters (Neutral) | 0.7 | 0.929 ± 0.258 |
| R1-Distill-Qwen-14B | Insert Random Characters (Neutral) | 0.9 | 0.926 ± 0.261 |
| R1-Distill-Qwen-32B | Insert Random Characters (Neutral) | 0.1 | 0.926 ± 0.261 |
| R1-Distill-Qwen-32B | Insert Random Characters (Neutral) | 0.3 | 0.912 ± 0.284 |
| R1-Distill-Qwen-32B | Insert Random Characters (Neutral) | 0.5 | 0.917 ± 0.276 |
| R1-Distill-Qwen-32B | Insert Random Characters (Neutral) | 0.7 | 0.915 ± 0.279 |
| R1-Distill-Qwen-32B | Insert Random Characters (Neutral) | 0.9 | 0.926 ± 0.262 |
| R1-Distill-Qwen-7B | Insert Random Characters (Neutral) | 0.1 | 0.862 ± 0.345 |
| R1-Distill-Qwen-7B | Insert Random Characters (Neutral) | 0.3 | 0.874 ± 0.332 |
| R1-Distill-Qwen-7B | Insert Random Characters (Neutral) | 0.5 | 0.864 ± 0.343 |
| R1-Distill-Qwen-7B | Insert Random Characters (Neutral) | 0.7 | 0.870 ± 0.336 |
| R1-Distill-Qwen-7B | Insert Random Characters (Neutral) | 0.9 | 0.856 ± 0.351 |
| EXAONE-Deep-32B | Insert Wrong Fact (Adv.) | 0.1 | 0.982 ± 0.134 |
| EXAONE-Deep-32B | Insert Wrong Fact (Adv.) | 0.3 | 0.981 ± 0.138 |
| EXAONE-Deep-32B | Insert Wrong Fact (Adv.) | 0.5 | 0.979 ± 0.144 |
| EXAONE-Deep-32B | Insert Wrong Fact (Adv.) | 0.7 | 0.984 ± 0.126 |
| EXAONE-Deep-32B | Insert Wrong Fact (Adv.) | 0.9 | 0.985 ± 0.120 |
| Nemotron-Llama-3.1-Nano-8B | Insert Wrong Fact (Adv.) | 0.1 | 0.904 ± 0.295 |
| Nemotron-Llama-3.1-Nano-8B | Insert Wrong Fact (Adv.) | 0.3 | 0.900 ± 0.300 |
| Nemotron-Llama-3.1-Nano-8B | Insert Wrong Fact (Adv.) | 0.5 | 0.917 ± 0.276 |

Table 19: Robustness metrics (mean ± std) per model, intervention, and timestep on the Science domain.

| Model | Intervention | Timestep | Mean ± Std |
|---|---|---|---|
| Nemotron-Llama-3.1-Nano-8B | Insert Wrong Fact (Adv.) | 0.7 | 0.915 ± 0.280 |
| Nemotron-Llama-3.1-Nano-8B | Insert Wrong Fact (Adv.) | 0.9 | 0.926 ± 0.261 |
| Phi-4-Reasoning-Plus | Insert Wrong Fact (Adv.) | 0.1 | 0.868 ± 0.339 |
| Phi-4-Reasoning-Plus | Insert Wrong Fact (Adv.) | 0.3 | 0.879 ± 0.326 |
| Phi-4-Reasoning-Plus | Insert Wrong Fact (Adv.) | 0.5 | 0.874 ± 0.331 |
| Phi-4-Reasoning-Plus | Insert Wrong Fact (Adv.) | 0.7 | 0.870 ± 0.336 |
| Phi-4-Reasoning-Plus | Insert Wrong Fact (Adv.) | 0.9 | 0.882 ± 0.323 |
| QwQ-32B | Insert Wrong Fact (Adv.) | 0.1 | 0.927 ± 0.260 |
| QwQ-32B | Insert Wrong Fact (Adv.) | 0.3 | 0.927 ± 0.259 |
| QwQ-32B | Insert Wrong Fact (Adv.) | 0.5 | 0.919 ± 0.273 |
| QwQ-32B | Insert Wrong Fact (Adv.) | 0.7 | 0.916 ± 0.277 |
| QwQ-32B | Insert Wrong Fact (Adv.) | 0.9 | 0.918 ± 0.274 |
| R1-Distill-Llama-8B | Insert Wrong Fact (Adv.) | 0.1 | 0.760 ± 0.427 |
| R1-Distill-Llama-8B | Insert Wrong Fact (Adv.) | 0.3 | 0.798 ± 0.402 |
| R1-Distill-Llama-8B | Insert Wrong Fact (Adv.) | 0.5 | 0.821 ± 0.383 |
| R1-Distill-Llama-8B | Insert Wrong Fact (Adv.) | 0.7 | 0.840 ± 0.366 |
| R1-Distill-Llama-8B | Insert Wrong Fact (Adv.) | 0.9 | 0.873 ± 0.333 |
| R1-Distill-Qwen-1.5B | Insert Wrong Fact (Adv.) | 0.1 | 0.761 ± 0.427 |
| R1-Distill-Qwen-1.5B | Insert Wrong Fact (Adv.) | 0.3 | 0.786 ± 0.410 |
| R1-Distill-Qwen-1.5B | Insert Wrong Fact (Adv.) | 0.5 | 0.795 ± 0.403 |
| R1-Distill-Qwen-1.5B | Insert Wrong Fact (Adv.) | 0.7 | 0.827 ± 0.378 |
| R1-Distill-Qwen-1.5B | Insert Wrong Fact (Adv.) | 0.9 | 0.857 ± 0.350 |
| R1-Distill-Qwen-14B | Insert Wrong Fact (Adv.) | 0.1 | 0.900 ± 0.300 |
| R1-Distill-Qwen-14B | Insert Wrong Fact (Adv.) | 0.3 | 0.920 ± 0.271 |
| R1-Distill-Qwen-14B | Insert Wrong Fact (Adv.) | 0.5 | 0.924 ± 0.265 |
| R1-Distill-Qwen-14B | Insert Wrong Fact (Adv.) | 0.7 | 0.927 ± 0.260 |
| R1-Distill-Qwen-14B | Insert Wrong Fact (Adv.) | 0.9 | 0.927 ± 0.259 |
| R1-Distill-Qwen-32B | Insert Wrong Fact (Adv.) | 0.1 | 0.914 ± 0.280 |
| R1-Distill-Qwen-32B | Insert Wrong Fact (Adv.) | 0.3 | 0.919 ± 0.273 |
| R1-Distill-Qwen-32B | Insert Wrong Fact (Adv.) | 0.5 | 0.907 ± 0.290 |
| R1-Distill-Qwen-32B | Insert Wrong Fact (Adv.) | 0.7 | 0.917 ± 0.276 |
| R1-Distill-Qwen-32B | Insert Wrong Fact (Adv.) | 0.9 | 0.920 ± 0.271 |
| R1-Distill-Qwen-7B | Insert Wrong Fact (Adv.) | 0.1 | 0.886 ± 0.318 |
| R1-Distill-Qwen-7B | Insert Wrong Fact (Adv.) | 0.3 | 0.900 ± 0.299 |
| R1-Distill-Qwen-7B | Insert Wrong Fact (Adv.) | 0.5 | 0.890 ± 0.313 |
| R1-Distill-Qwen-7B | Insert Wrong Fact (Adv.) | 0.7 | 0.906 ± 0.291 |
| R1-Distill-Qwen-7B | Insert Wrong Fact (Adv.) | 0.9 | 0.900 ± 0.300 |
| EXAONE-Deep-32B | Rewrite Trace (Benign) | 0.1 | 0.973 ± 0.162 |
| EXAONE-Deep-32B | Rewrite Trace (Benign) | 0.3 | 0.977 ± 0.149 |
| EXAONE-Deep-32B | Rewrite Trace (Benign) | 0.5 | 0.974 ± 0.159 |
| EXAONE-Deep-32B | Rewrite Trace (Benign) | 0.7 | 0.973 ± 0.161 |
| EXAONE-Deep-32B | Rewrite Trace (Benign) | 0.9 | 0.971 ± 0.167 |
| Nemotron-Llama-3.1-Nano-8B | Rewrite Trace (Benign) | 0.1 | 0.756 ± 0.430 |
| Nemotron-Llama-3.1-Nano-8B | Rewrite Trace (Benign) | 0.3 | 0.814 ± 0.389 |
| Nemotron-Llama-3.1-Nano-8B | Rewrite Trace (Benign) | 0.5 | 0.843 ± 0.364 |
| Nemotron-Llama-3.1-Nano-8B | Rewrite Trace (Benign) | 0.7 | 0.878 ± 0.327 |

Continued on next page

Table 19: Robustness metrics (mean ± std) per model, intervention, and timestep on the Science domain.

| Model | Intervention | Timestep | Mean ± Std |
|---|---|---|---|
| Nemotron-Llama-3.1-Nano-8B | Rewrite Trace (Benign) | 0.9 | 0.904 ± 0.295 |
| Phi-4-Reasoning-Plus | Rewrite Trace (Benign) | 0.1 | 0.859 ± 0.348 |
| Phi-4-Reasoning-Plus | Rewrite Trace (Benign) | 0.3 | 0.856 ± 0.352 |
| Phi-4-Reasoning-Plus | Rewrite Trace (Benign) | 0.5 | 0.843 ± 0.364 |
| Phi-4-Reasoning-Plus | Rewrite Trace (Benign) | 0.7 | 0.845 ± 0.362 |
| Phi-4-Reasoning-Plus | Rewrite Trace (Benign) | 0.9 | 0.869 ± 0.338 |
| QwQ-32B | Rewrite Trace (Benign) | 0.1 | 0.919 ± 0.273 |
| QwQ-32B | Rewrite Trace (Benign) | 0.3 | 0.919 ± 0.272 |
| QwQ-32B | Rewrite Trace (Benign) | 0.5 | 0.927 ± 0.260 |
| QwQ-32B | Rewrite Trace (Benign) | 0.7 | 0.933 ± 0.250 |
| QwQ-32B | Rewrite Trace (Benign) | 0.9 | 0.936 ± 0.244 |
| R1-Distill-Llama-8B | Rewrite Trace (Benign) | 0.1 | 0.555 ± 0.497 |
| R1-Distill-Llama-8B | Rewrite Trace (Benign) | 0.3 | 0.633 ± 0.482 |
| R1-Distill-Llama-8B | Rewrite Trace (Benign) | 0.5 | 0.721 ± 0.449 |
| R1-Distill-Llama-8B | Rewrite Trace (Benign) | 0.7 | 0.787 ± 0.409 |
| R1-Distill-Llama-8B | Rewrite Trace (Benign) | 0.9 | 0.814 ± 0.389 |
| R1-Distill-Qwen-1.5B | Rewrite Trace (Benign) | 0.1 | 0.667 ± 0.471 |
| R1-Distill-Qwen-1.5B | Rewrite Trace (Benign) | 0.3 | 0.731 ± 0.444 |
| R1-Distill-Qwen-1.5B | Rewrite Trace (Benign) | 0.5 | 0.838 ± 0.368 |
| R1-Distill-Qwen-1.5B | Rewrite Trace (Benign) | 0.7 | 0.841 ± 0.365 |
| R1-Distill-Qwen-1.5B | Rewrite Trace (Benign) | 0.9 | 0.854 ± 0.353 |
| R1-Distill-Qwen-14B | Rewrite Trace (Benign) | 0.1 | 0.701 ± 0.458 |
| R1-Distill-Qwen-14B | Rewrite Trace (Benign) | 0.3 | 0.795 ± 0.404 |
| R1-Distill-Qwen-14B | Rewrite Trace (Benign) | 0.5 | 0.857 ± 0.350 |
| R1-Distill-Qwen-14B | Rewrite Trace (Benign) | 0.7 | 0.878 ± 0.327 |
| R1-Distill-Qwen-14B | Rewrite Trace (Benign) | 0.9 | 0.903 ± 0.297 |
| R1-Distill-Qwen-32B | Rewrite Trace (Benign) | 0.1 | 0.772 ± 0.419 |
| R1-Distill-Qwen-32B | Rewrite Trace (Benign) | 0.3 | 0.832 ± 0.374 |
| R1-Distill-Qwen-32B | Rewrite Trace (Benign) | 0.5 | 0.839 ± 0.367 |
| R1-Distill-Qwen-32B | Rewrite Trace (Benign) | 0.7 | 0.889 ± 0.314 |
| R1-Distill-Qwen-32B | Rewrite Trace (Benign) | 0.9 | 0.890 ± 0.313 |
| R1-Distill-Qwen-7B | Rewrite Trace (Benign) | 0.1 | 0.725 ± 0.447 |
| R1-Distill-Qwen-7B | Rewrite Trace (Benign) | 0.3 | 0.821 ± 0.383 |
| R1-Distill-Qwen-7B | Rewrite Trace (Benign) | 0.5 | 0.855 ± 0.352 |
| R1-Distill-Qwen-7B | Rewrite Trace (Benign) | 0.7 | 0.869 ± 0.337 |
| R1-Distill-Qwen-7B | Rewrite Trace (Benign) | 0.9 | 0.884 ± 0.321 |

Table 20: Robustness metrics (mean ± std) per model, intervention, and timestep on the Logic domain.

| Model | Intervention | Timestep | Mean ± Std |
|---|---|---|---|
| EXAONE-Deep-32B | Add Random Text (Neutral) | 0.1 | 0.901 ± 0.298 |
| EXAONE-Deep-32B | Add Random Text (Neutral) | 0.3 | 0.915 ± 0.278 |
| EXAONE-Deep-32B | Add Random Text (Neutral) | 0.5 | 0.931 ± 0.253 |
| EXAONE-Deep-32B | Add Random Text (Neutral) | 0.7 | 0.948 ± 0.222 |

Table 20: Robustness metrics (mean ± std) per model, intervention, and timestep on the Logic domain.

| Model | Intervention | Timestep | Mean ± Std |
|---|---|---|---|
| EXAONE-Deep-32B | Add Random Text (Neutral) | 0.9 | 0.962 ± 0.192 |
| Nemotron-Llama-3.1-Nano-8B | Add Random Text (Neutral) | 0.1 | 0.909 ± 0.287 |
| Nemotron-Llama-3.1-Nano-8B | Add Random Text (Neutral) | 0.3 | 0.936 ± 0.245 |
| Nemotron-Llama-3.1-Nano-8B | Add Random Text (Neutral) | 0.5 | 0.949 ± 0.220 |
| Nemotron-Llama-3.1-Nano-8B | Add Random Text (Neutral) | 0.7 | 0.963 ± 0.188 |
| Nemotron-Llama-3.1-Nano-8B | Add Random Text (Neutral) | 0.9 | 0.967 ± 0.178 |
| Phi-4-Reasoning-Plus | Add Random Text (Neutral) | 0.1 | 0.938 ± 0.241 |
| Phi-4-Reasoning-Plus | Add Random Text (Neutral) | 0.3 | 0.944 ± 0.229 |
| Phi-4-Reasoning-Plus | Add Random Text (Neutral) | 0.5 | 0.940 ± 0.237 |
| Phi-4-Reasoning-Plus | Add Random Text (Neutral) | 0.7 | 0.941 ± 0.236 |
| Phi-4-Reasoning-Plus | Add Random Text (Neutral) | 0.9 | 0.939 ± 0.240 |
| QwQ-32B | Add Random Text (Neutral) | 0.1 | 0.963 ± 0.189 |
| QwQ-32B | Add Random Text (Neutral) | 0.3 | 0.975 ± 0.157 |
| QwQ-32B | Add Random Text (Neutral) | 0.5 | 0.977 ± 0.149 |
| QwQ-32B | Add Random Text (Neutral) | 0.7 | 0.981 ± 0.136 |
| QwQ-32B | Add Random Text (Neutral) | 0.9 | 0.977 ± 0.149 |
| R1-Distill-Llama-8B | Add Random Text (Neutral) | 0.1 | 0.927 ± 0.261 |
| R1-Distill-Llama-8B | Add Random Text (Neutral) | 0.3 | 0.960 ± 0.197 |
| R1-Distill-Llama-8B | Add Random Text (Neutral) | 0.5 | 0.960 ± 0.197 |
| R1-Distill-Llama-8B | Add Random Text (Neutral) | 0.7 | 0.964 ± 0.186 |
| R1-Distill-Llama-8B | Add Random Text (Neutral) | 0.9 | 0.979 ± 0.144 |
| R1-Distill-Qwen-1.5B | Add Random Text (Neutral) | 0.1 | 0.570 ± 0.495 |
| R1-Distill-Qwen-1.5B | Add Random Text (Neutral) | 0.3 | 0.599 ± 0.490 |
| R1-Distill-Qwen-1.5B | Add Random Text (Neutral) | 0.5 | 0.630 ± 0.483 |
| R1-Distill-Qwen-1.5B | Add Random Text (Neutral) | 0.7 | 0.646 ± 0.478 |
| R1-Distill-Qwen-1.5B | Add Random Text (Neutral) | 0.9 | 0.724 ± 0.447 |
| R1-Distill-Qwen-14B | Add Random Text (Neutral) | 0.1 | 0.969 ± 0.173 |
| R1-Distill-Qwen-14B | Add Random Text (Neutral) | 0.3 | 0.985 ± 0.121 |
| R1-Distill-Qwen-14B | Add Random Text (Neutral) | 0.5 | 0.988 ± 0.107 |

Table 20: Robustness metrics (mean ± std) per model, intervention, and timestep on the Logic domain.

| Model | Intervention | Timestep | Mean ± Std |
|---|---|---|---|
| R1-Distill-Qwen-14B | Add Random Text (Neutral) | 0.7 | 0.988 ± 0.107 |
| R1-Distill-Qwen-14B | Add Random Text (Neutral) | 0.9 | 0.984 ± 0.126 |
| R1-Distill-Qwen-32B | Add Random Text (Neutral) | 0.1 | 0.975 ± 0.156 |
| R1-Distill-Qwen-32B | Add Random Text (Neutral) | 0.3 | 0.982 ± 0.133 |
| R1-Distill-Qwen-32B | Add Random Text (Neutral) | 0.5 | 0.989 ± 0.103 |
| R1-Distill-Qwen-32B | Add Random Text (Neutral) | 0.7 | 0.990 ± 0.101 |
| R1-Distill-Qwen-32B | Add Random Text (Neutral) | 0.9 | 0.989 ± 0.105 |
| R1-Distill-Qwen-7B | Add Random Text (Neutral) | 0.1 | 0.916 ± 0.278 |
| R1-Distill-Qwen-7B | Add Random Text (Neutral) | 0.3 | 0.918 ± 0.274 |
| R1-Distill-Qwen-7B | Add Random Text (Neutral) | 0.5 | 0.933 ± 0.250 |
| R1-Distill-Qwen-7B | Add Random Text (Neutral) | 0.7 | 0.927 ± 0.261 |
| R1-Distill-Qwen-7B | Add Random Text (Neutral) | 0.9 | 0.945 ± 0.228 |
| EXAONE-Deep-32B | Complete Step (Benign) | 0.1 | 0.993 ± 0.080 |
| EXAONE-Deep-32B | Complete Step (Benign) | 0.3 | 0.996 ± 0.062 |
| EXAONE-Deep-32B | Complete Step (Benign) | 0.5 | 0.998 ± 0.039 |
| EXAONE-Deep-32B | Complete Step (Benign) | 0.7 | 0.999 ± 0.028 |
| EXAONE-Deep-32B | Complete Step (Benign) | 0.9 | 0.999 ± 0.028 |
| Nemotron-Llama-3.1-Nano-8B | Complete Step (Benign) | 0.1 | 0.974 ± 0.160 |
| Nemotron-Llama-3.1-Nano-8B | Complete Step (Benign) | 0.3 | 0.978 ± 0.146 |
| Nemotron-Llama-3.1-Nano-8B | Complete Step (Benign) | 0.5 | 0.981 ± 0.136 |
| Nemotron-Llama-3.1-Nano-8B | Complete Step (Benign) | 0.7 | 0.989 ± 0.105 |
| Nemotron-Llama-3.1-Nano-8B | Complete Step (Benign) | 0.9 | 0.992 ± 0.089 |
| Phi-4-Reasoning-Plus | Complete Step (Benign) | 0.1 | 0.952 ± 0.214 |
| Phi-4-Reasoning-Plus | Complete Step (Benign) | 0.3 | 0.959 ± 0.197 |
| Phi-4-Reasoning-Plus | Complete Step (Benign) | 0.5 | 0.955 ± 0.208 |
| Phi-4-Reasoning-Plus | Complete Step (Benign) | 0.7 | 0.946 ± 0.227 |
| Phi-4-Reasoning-Plus | Complete Step (Benign) | 0.9 | 0.953 ± 0.211 |
| QwQ-32B | Complete Step (Benign) | 0.1 | 0.997 ± 0.052 |
| QwQ-32B | Complete Step (Benign) | 0.3 | 0.995 ± 0.070 |
| QwQ-32B | Complete Step (Benign) | 0.5 | 0.997 ± 0.059 |
| QwQ-32B | Complete Step (Benign) | 0.7 | 0.997 ± 0.055 |
| QwQ-32B | Complete Step (Benign) | 0.9 | 0.999 ± 0.034 |
| R1-Distill-Llama-8B | Complete Step (Benign) | 0.1 | 0.941 ± 0.236 |
| R1-Distill-Llama-8B | Complete Step (Benign) | 0.3 | 0.960 ± 0.197 |
| R1-Distill-Llama-8B | Complete Step (Benign) | 0.5 | 0.975 ± 0.156 |
| R1-Distill-Llama-8B | Complete Step (Benign) | 0.7 | 0.977 ± 0.151 |
| R1-Distill-Llama-8B | Complete Step (Benign) | 0.9 | 0.982 ± 0.132 |
| R1-Distill-Qwen-1.5B | Complete Step (Benign) | 0.1 | 0.646 ± 0.478 |
| R1-Distill-Qwen-1.5B | Complete Step (Benign) | 0.3 | 0.700 ± 0.458 |

Table 20: Robustness metrics (mean ± std) per model, intervention, and timestep on the Logic domain.

| Model | Intervention | | Timestep | Mean ± Std |
|---|---|---|---|---|
| R1-Distill-Qwen-1.5B | Complete Step (Benign) | | 0.5 | 0.740 ± 0.438 |
| R1-Distill-Qwen-1.5B | Complete Step (Benign) | | 0.7 | 0.783 ± 0.412 |
| R1-Distill-Qwen-1.5B | Complete Step (Benign) | | 0.9 | 0.852 ± 0.355 |
| R1-Distill-Qwen-14B | Complete Step (Benign) | | 0.1 | 0.993 ± 0.080 |
| R1-Distill-Qwen-14B | Complete Step (Benign) | | 0.3 | 0.990 ± 0.099 |
| R1-Distill-Qwen-14B | Complete Step (Benign) | | 0.5 | 0.987 ± 0.112 |
| R1-Distill-Qwen-14B | Complete Step (Benign) | | 0.7 | 0.992 ± 0.087 |
| R1-Distill-Qwen-14B | Complete Step (Benign) | | 0.9 | 0.997 ± 0.059 |
| R1-Distill-Qwen-32B | Complete Step (Benign) | | 0.1 | 0.996 ± 0.065 |
| R1-Distill-Qwen-32B | Complete Step (Benign) | | 0.3 | 0.993 ± 0.080 |
| R1-Distill-Qwen-32B | Complete Step (Benign) | | 0.5 | 0.994 ± 0.078 |
| R1-Distill-Qwen-32B | Complete Step (Benign) | | 0.7 | 0.998 ± 0.044 |
| R1-Distill-Qwen-32B | Complete Step (Benign) | | 0.9 | 0.998 ± 0.044 |
| R1-Distill-Qwen-7B | Complete Step (Benign) | | 0.1 | 0.954 ± 0.209 |
| R1-Distill-Qwen-7B | Complete Step (Benign) | | 0.3 | 0.961 ± 0.194 |
| R1-Distill-Qwen-7B | Complete Step (Benign) | | 0.5 | 0.974 ± 0.160 |
| R1-Distill-Qwen-7B | Complete Step (Benign) | | 0.7 | 0.977 ± 0.149 |
| R1-Distill-Qwen-7B | Complete Step (Benign) | | 0.9 | 0.981 ± 0.136 |
| EXAONE-Deep-32B | Continue (Adv.) | Unrelated | 0.1 | 0.410 ± 0.492 |
| EXAONE-Deep-32B | Continue (Adv.) | Unrelated | 0.3 | 0.589 ± 0.492 |
| EXAONE-Deep-32B | Continue (Adv.) | Unrelated | 0.5 | 0.677 ± 0.468 |
| EXAONE-Deep-32B | Continue (Adv.) | Unrelated | 0.7 | 0.766 ± 0.423 |
| EXAONE-Deep-32B | Continue (Adv.) | Unrelated | 0.9 | 0.810 ± 0.392 |
| Nemotron-Llama-3.1-Nano-8B | Continue (Adv.) | Unrelated | 0.1 | 0.579 ± 0.494 |
| Nemotron-Llama-3.1-Nano-8B | Continue (Adv.) | Unrelated | 0.3 | 0.686 ± 0.464 |
| Nemotron-Llama-3.1-Nano-8B | Continue (Adv.) | Unrelated | 0.5 | 0.708 ± 0.455 |
| Nemotron-Llama-3.1-Nano-8B | Continue (Adv.) | Unrelated | 0.7 | 0.781 ± 0.414 |
| Nemotron-Llama-3.1-Nano-8B | Continue (Adv.) | Unrelated | 0.9 | 0.834 ± 0.372 |
| Phi-4-Reasoning-Plus | Continue (Adv.) | Unrelated | 0.1 | 0.954 ± 0.210 |
| Phi-4-Reasoning-Plus | Continue (Adv.) | Unrelated | 0.3 | 0.954 ± 0.210 |
| Phi-4-Reasoning-Plus | Continue (Adv.) | Unrelated | 0.5 | 0.956 ± 0.205 |
| Phi-4-Reasoning-Plus | Continue (Adv.) | Unrelated | 0.7 | 0.954 ± 0.210 |
| Phi-4-Reasoning-Plus | Continue (Adv.) | Unrelated | 0.9 | 0.945 ± 0.228 |
| QwQ-32B | Continue (Adv.) | Unrelated | 0.1 | 0.987 ± 0.113 |
| QwQ-32B | Continue (Adv.) | Unrelated | 0.3 | 0.994 ± 0.076 |
| QwQ-32B | Continue (Adv.) | Unrelated | 0.5 | 0.995 ± 0.070 |

Continued on next page

Table 20: Robustness metrics (mean ± std) per model, intervention, and timestep on the Logic domain.

| Model | Intervention | | Timestep | Mean ± Std |
|---|---|---|---|---|
| QwQ-32B | Continue (Adv.) | Unrelated | 0.7 | 0.997 ± 0.055 |
| QwQ-32B | Continue (Adv.) | Unrelated | 0.9 | 0.997 ± 0.055 |
| R1-Distill-Llama-8B | Continue (Adv.) | Unrelated | 0.1 | 0.821 ± 0.383 |
| R1-Distill-Llama-8B | Continue (Adv.) | Unrelated | 0.3 | 0.929 ± 0.256 |
| R1-Distill-Llama-8B | Continue (Adv.) | Unrelated | 0.5 | 0.945 ± 0.228 |
| R1-Distill-Llama-8B | Continue (Adv.) | Unrelated | 0.7 | 0.962 ± 0.192 |
| R1-Distill-Llama-8B | Continue (Adv.) | Unrelated | 0.9 | 0.973 ± 0.162 |
| R1-Distill-Qwen-1.5B | Continue (Adv.) | Unrelated | 0.1 | 0.558 ± 0.497 |
| R1-Distill-Qwen-1.5B | Continue (Adv.) | Unrelated | 0.3 | 0.574 ± 0.494 |
| R1-Distill-Qwen-1.5B | Continue (Adv.) | Unrelated | 0.5 | 0.591 ± 0.492 |
| R1-Distill-Qwen-1.5B | Continue (Adv.) | Unrelated | 0.7 | 0.643 ± 0.479 |
| R1-Distill-Qwen-1.5B | Continue (Adv.) | Unrelated | 0.9 | 0.674 ± 0.469 |
| R1-Distill-Qwen-14B | Continue (Adv.) | Unrelated | 0.1 | 0.648 ± 0.477 |
| R1-Distill-Qwen-14B | Continue (Adv.) | Unrelated | 0.3 | 0.871 ± 0.335 |
| R1-Distill-Qwen-14B | Continue (Adv.) | Unrelated | 0.5 | 0.934 ± 0.249 |
| R1-Distill-Qwen-14B | Continue (Adv.) | Unrelated | 0.7 | 0.927 ± 0.260 |
| R1-Distill-Qwen-14B | Continue (Adv.) | Unrelated | 0.9 | 0.869 ± 0.337 |
| R1-Distill-Qwen-32B | Continue (Adv.) | Unrelated | 0.1 | 0.861 ± 0.346 |
| R1-Distill-Qwen-32B | Continue (Adv.) | Unrelated | 0.3 | 0.966 ± 0.182 |
| R1-Distill-Qwen-32B | Continue (Adv.) | Unrelated | 0.5 | 0.985 ± 0.121 |
| R1-Distill-Qwen-32B | Continue (Adv.) | Unrelated | 0.7 | 0.994 ± 0.076 |
| R1-Distill-Qwen-32B | Continue (Adv.) | Unrelated | 0.9 | 0.993 ± 0.080 |
| R1-Distill-Qwen-7B | Continue (Adv.) | Unrelated | 0.1 | 0.895 ± 0.306 |
| R1-Distill-Qwen-7B | Continue (Adv.) | Unrelated | 0.3 | 0.944 ± 0.229 |
| R1-Distill-Qwen-7B | Continue (Adv.) | Unrelated | 0.5 | 0.948 ± 0.222 |
| R1-Distill-Qwen-7B | Continue (Adv.) | Unrelated | 0.7 | 0.952 ± 0.214 |
| R1-Distill-Qwen-7B | Continue (Adv.) | Unrelated | 0.9 | 0.947 ± 0.225 |
| EXAONE-Deep-32B | Ctn. Wrong Reasoning (Adv.) | | 0.1 | 0.991 ± 0.093 |
| EXAONE-Deep-32B | Ctn. Wrong Reasoning (Adv.) | | 0.3 | 0.997 ± 0.052 |

Continued on next page

Table 20: Robustness metrics (mean ± std) per model, intervention, and timestep on the Logic domain.

| Model | Intervention | | Timestep | Mean ± Std |
|---|---|---|---|---|
| EXAONE-Deep-32B | Ctn. Wrong Reasoning (Adv.) | | 0.5 | 0.996 ± 0.065 |
| EXAONE-Deep-32B | Ctn. Wrong Reasoning (Adv.) | | 0.7 | 0.997 ± 0.055 |
| EXAONE-Deep-32B | Ctn. Wrong Reasoning (Adv.) | | 0.9 | 1.000 ± 0.020 |
| Nemotron-Llama-3.1-Nano-8B | Ctn. Wrong Reasoning (Adv.) | | 0.1 | 0.954 ± 0.210 |
| Nemotron-Llama-3.1-Nano-8B | Ctn. Wrong Reasoning (Adv.) | | 0.3 | 0.964 ± 0.187 |
| Nemotron-Llama-3.1-Nano-8B | Ctn. Wrong Reasoning (Adv.) | | 0.5 | 0.971 ± 0.167 |
| Nemotron-Llama-3.1-Nano-8B | Ctn. Wrong Reasoning (Adv.) | | 0.7 | 0.976 ± 0.154 |
| Nemotron-Llama-3.1-Nano-8B | Ctn. Wrong Reasoning (Adv.) | | 0.9 | 0.986 ± 0.117 |
| Phi-4-Reasoning-Plus | Ctn. Wrong Reasoning (Adv.) | | 0.1 | 0.954 ± 0.209 |
| Phi-4-Reasoning-Plus | Ctn. Wrong Reasoning (Adv.) | | 0.3 | 0.954 ± 0.209 |
| Phi-4-Reasoning-Plus | Ctn. Wrong Reasoning (Adv.) | | 0.5 | 0.960 ± 0.196 |
| Phi-4-Reasoning-Plus | Ctn. Wrong Reasoning (Adv.) | | 0.7 | 0.955 ± 0.208 |
| Phi-4-Reasoning-Plus | Ctn. Wrong Reasoning (Adv.) | | 0.9 | 0.953 ± 0.211 |
| QwQ-32B | Ctn. Wrong Reasoning (Adv.) | | 0.1 | 0.994 ± 0.076 |
| QwQ-32B | Ctn. Wrong Reasoning (Adv.) | | 0.3 | 0.997 ± 0.052 |
| QwQ-32B | Ctn. Wrong Reasoning (Adv.) | | 0.5 | 0.997 ± 0.052 |
| QwQ-32B | Ctn. Wrong Reasoning (Adv.) | | 0.7 | 0.997 ± 0.055 |
| QwQ-32B | Ctn. Wrong Reasoning (Adv.) | | 0.9 | 0.999 ± 0.034 |
| R1-Distill-Llama-8B | Ctn. Wrong Reasoning (Adv.) | | 0.1 | 0.906 ± 0.292 |
| R1-Distill-Llama-8B | Ctn. Wrong Reasoning (Adv.) | | 0.3 | 0.944 ± 0.231 |
| R1-Distill-Llama-8B | Ctn. Wrong Reasoning (Adv.) | | 0.5 | 0.937 ± 0.243 |
| R1-Distill-Llama-8B | Ctn. Wrong Reasoning (Adv.) | | 0.7 | 0.965 ± 0.184 |
| R1-Distill-Llama-8B | Ctn. Wrong Reasoning (Adv.) | | 0.9 | 0.971 ± 0.168 |
| R1-Distill-Qwen-1.5B | Ctn. Wrong Reasoning (Adv.) | | 0.1 | 0.569 ± 0.495 |
| R1-Distill-Qwen-1.5B | Ctn. Wrong Reasoning (Adv.) | | 0.3 | 0.589 ± 0.492 |
| R1-Distill-Qwen-1.5B | Ctn. Wrong Reasoning (Adv.) | | 0.5 | 0.617 ± 0.486 |
| R1-Distill-Qwen-1.5B | Ctn. Wrong Reasoning (Adv.) | | 0.7 | 0.653 ± 0.476 |
| R1-Distill-Qwen-1.5B | Ctn. Wrong Reasoning (Adv.) | | 0.9 | 0.751 ± 0.433 |
| R1-Distill-Qwen-14B | Ctn. Wrong Reasoning (Adv.) | | 0.1 | 0.984 ± 0.126 |

Table 20: Robustness metrics (mean ± std) per model, intervention, and timestep on the Logic domain.

| Model | Intervention | Timestep | Mean ± Std |
|---|---|---|---|
| R1-Distill-Qwen-14B | Ctn. Wrong Reasoning (Adv.) | 0.3 | 0.982 ± 0.133 |
| R1-Distill-Qwen-14B | Ctn. Wrong Reasoning (Adv.) | 0.5 | 0.988 ± 0.107 |
| R1-Distill-Qwen-14B | Ctn. Wrong Reasoning (Adv.) | 0.7 | 0.990 ± 0.099 |
| R1-Distill-Qwen-14B | Ctn. Wrong Reasoning (Adv.) | 0.9 | 0.995 ± 0.070 |
| R1-Distill-Qwen-32B | Ctn. Wrong Reasoning (Adv.) | 0.1 | 0.986 ± 0.118 |
| R1-Distill-Qwen-32B | Ctn. Wrong Reasoning (Adv.) | 0.3 | 0.988 ± 0.110 |
| R1-Distill-Qwen-32B | Ctn. Wrong Reasoning (Adv.) | 0.5 | 0.985 ± 0.120 |
| R1-Distill-Qwen-32B | Ctn. Wrong Reasoning (Adv.) | 0.7 | 0.987 ± 0.115 |
| R1-Distill-Qwen-32B | Ctn. Wrong Reasoning (Adv.) | 0.9 | 0.992 ± 0.087 |
| R1-Distill-Qwen-7B | Ctn. Wrong Reasoning (Adv.) | 0.1 | 0.931 ± 0.254 |
| R1-Distill-Qwen-7B | Ctn. Wrong Reasoning (Adv.) | 0.3 | 0.935 ± 0.246 |
| R1-Distill-Qwen-7B | Ctn. Wrong Reasoning (Adv.) | 0.5 | 0.955 ± 0.208 |
| R1-Distill-Qwen-7B | Ctn. Wrong Reasoning (Adv.) | 0.7 | 0.960 ± 0.197 |
| R1-Distill-Qwen-7B | Ctn. Wrong Reasoning (Adv.) | 0.9 | 0.977 ± 0.150 |
| EXAONE-Deep-32B | Insert Random Characters (Neutral) | 0.1 | 0.994 ± 0.078 |
| EXAONE-Deep-32B | Insert Random Characters (Neutral) | 0.3 | 0.996 ± 0.062 |
| EXAONE-Deep-32B | Insert Random Characters (Neutral) | 0.5 | 0.997 ± 0.055 |
| EXAONE-Deep-32B | Insert Random Characters (Neutral) | 0.7 | 0.999 ± 0.034 |
| EXAONE-Deep-32B | Insert Random Characters (Neutral) | 0.9 | 0.999 ± 0.028 |
| Nemotron-Llama-3.1-Nano-8B | Insert Random Characters (Neutral) | 0.1 | 0.946 ± 0.227 |
| Nemotron-Llama-3.1-Nano-8B | Insert Random Characters (Neutral) | 0.3 | 0.951 ± 0.216 |
| Nemotron-Llama-3.1-Nano-8B | Insert Random Characters (Neutral) | 0.5 | 0.952 ± 0.214 |
| Nemotron-Llama-3.1-Nano-8B | Insert Random Characters (Neutral) | 0.7 | 0.967 ± 0.180 |
| Nemotron-Llama-3.1-Nano-8B | Insert Random Characters (Neutral) | 0.9 | 0.957 ± 0.204 |
| Phi-4-Reasoning-Plus | Insert Random Characters (Neutral) | 0.1 | 0.953 ± 0.211 |
| Phi-4-Reasoning-Plus | Insert Random Characters (Neutral) | 0.3 | 0.947 ± 0.224 |
| Phi-4-Reasoning-Plus | Insert Random Characters (Neutral) | 0.5 | 0.958 ± 0.201 |
| Phi-4-Reasoning-Plus | Insert Random Characters (Neutral) | 0.7 | 0.945 ± 0.228 |
| Phi-4-Reasoning-Plus | Insert Random Characters (Neutral) | 0.9 | 0.948 ± 0.222 |

Table 20: Robustness metrics (mean ± std) per model, intervention, and timestep on the Logic domain.

| Model | Intervention | Timestep | Mean ± Std |
|---|---|---|---|
| QwQ-32B | Insert Random Characters (Neutral) | 0.1 | 0.997 ± 0.052 |
| QwQ-32B | Insert Random Characters (Neutral) | 0.3 | 0.998 ± 0.044 |
| QwQ-32B | Insert Random Characters (Neutral) | 0.5 | 0.998 ± 0.044 |
| QwQ-32B | Insert Random Characters (Neutral) | 0.7 | 0.995 ± 0.068 |
| QwQ-32B | Insert Random Characters (Neutral) | 0.9 | 0.998 ± 0.048 |
| R1-Distill-Llama-8B | Insert Random Characters (Neutral) | 0.1 | 0.941 ± 0.236 |
| R1-Distill-Llama-8B | Insert Random Characters (Neutral) | 0.3 | 0.952 ± 0.214 |
| R1-Distill-Llama-8B | Insert Random Characters (Neutral) | 0.5 | 0.965 ± 0.183 |
| R1-Distill-Llama-8B | Insert Random Characters (Neutral) | 0.7 | 0.979 ± 0.144 |
| R1-Distill-Llama-8B | Insert Random Characters (Neutral) | 0.9 | 0.979 ± 0.142 |
| R1-Distill-Qwen-1.5B | Insert Random Characters (Neutral) | 0.1 | 0.599 ± 0.490 |
| R1-Distill-Qwen-1.5B | Insert Random Characters (Neutral) | 0.3 | 0.626 ± 0.484 |
| R1-Distill-Qwen-1.5B | Insert Random Characters (Neutral) | 0.5 | 0.646 ± 0.478 |
| R1-Distill-Qwen-1.5B | Insert Random Characters (Neutral) | 0.7 | 0.676 ± 0.468 |
| R1-Distill-Qwen-1.5B | Insert Random Characters (Neutral) | 0.9 | 0.791 ± 0.407 |
| R1-Distill-Qwen-14B | Insert Random Characters (Neutral) | 0.1 | 0.995 ± 0.073 |
| R1-Distill-Qwen-14B | Insert Random Characters (Neutral) | 0.3 | 0.990 ± 0.101 |
| R1-Distill-Qwen-14B | Insert Random Characters (Neutral) | 0.5 | 0.993 ± 0.085 |
| R1-Distill-Qwen-14B | Insert Random Characters (Neutral) | 0.7 | 0.995 ± 0.073 |
| R1-Distill-Qwen-14B | Insert Random Characters (Neutral) | 0.9 | 0.997 ± 0.055 |
| R1-Distill-Qwen-32B | Insert Random Characters (Neutral) | 0.1 | 0.991 ± 0.093 |
| R1-Distill-Qwen-32B | Insert Random Characters (Neutral) | 0.3 | 0.989 ± 0.105 |
| R1-Distill-Qwen-32B | Insert Random Characters (Neutral) | 0.5 | 0.997 ± 0.055 |
| R1-Distill-Qwen-32B | Insert Random Characters (Neutral) | 0.7 | 0.997 ± 0.059 |
| R1-Distill-Qwen-32B | Insert Random Characters (Neutral) | 0.9 | 0.997 ± 0.055 |
| R1-Distill-Qwen-7B | Insert Random Characters (Neutral) | 0.1 | 0.883 ± 0.322 |
| R1-Distill-Qwen-7B | Insert Random Characters (Neutral) | 0.3 | 0.910 ± 0.286 |
| R1-Distill-Qwen-7B | Insert Random Characters (Neutral) | 0.5 | 0.922 ± 0.268 |
| R1-Distill-Qwen-7B | Insert Random Characters (Neutral) | 0.7 | 0.937 ± 0.243 |

Table 20: Robustness metrics (mean ± std) per model, intervention, and timestep on the Logic domain.

| Model | Intervention | Timestep | Mean ± Std |
|---|---|---|---|
| R1-Distill-Qwen-7B | Insert Random Characters (Neutral) | 0.9 | 0.934 ± 0.248 |
| EXAONE-Deep-32B | Insert Wrong Fact (Adv.) | 0.1 | 0.984 ± 0.124 |
| EXAONE-Deep-32B | Insert Wrong Fact (Adv.) | 0.3 | 0.989 ± 0.105 |
| EXAONE-Deep-32B | Insert Wrong Fact (Adv.) | 0.5 | 0.991 ± 0.095 |
| EXAONE-Deep-32B | Insert Wrong Fact (Adv.) | 0.7 | 0.998 ± 0.048 |
| EXAONE-Deep-32B | Insert Wrong Fact (Adv.) | 0.9 | 0.999 ± 0.028 |
| Nemotron-Llama-3.1-Nano-8B | Insert Wrong Fact (Adv.) | 0.1 | 0.947 ± 0.225 |
| Nemotron-Llama-3.1-Nano-8B | Insert Wrong Fact (Adv.) | 0.3 | 0.962 ± 0.191 |
| Nemotron-Llama-3.1-Nano-8B | Insert Wrong Fact (Adv.) | 0.5 | 0.972 ± 0.165 |
| Nemotron-Llama-3.1-Nano-8B | Insert Wrong Fact (Adv.) | 0.7 | 0.982 ± 0.132 |
| Nemotron-Llama-3.1-Nano-8B | Insert Wrong Fact (Adv.) | 0.9 | 0.988 ± 0.108 |
| Phi-4-Reasoning-Plus | Insert Wrong Fact (Adv.) | 0.1 | 0.946 ± 0.227 |
| Phi-4-Reasoning-Plus | Insert Wrong Fact (Adv.) | 0.3 | 0.956 ± 0.205 |
| Phi-4-Reasoning-Plus | Insert Wrong Fact (Adv.) | 0.5 | 0.957 ± 0.203 |
| Phi-4-Reasoning-Plus | Insert Wrong Fact (Adv.) | 0.7 | 0.949 ± 0.219 |
| Phi-4-Reasoning-Plus | Insert Wrong Fact (Adv.) | 0.9 | 0.956 ± 0.204 |
| QwQ-32B | Insert Wrong Fact (Adv.) | 0.1 | 0.997 ± 0.055 |
| QwQ-32B | Insert Wrong Fact (Adv.) | 0.3 | 0.997 ± 0.059 |
| QwQ-32B | Insert Wrong Fact (Adv.) | 0.5 | 0.997 ± 0.055 |
| QwQ-32B | Insert Wrong Fact (Adv.) | 0.7 | 0.999 ± 0.028 |
| QwQ-32B | Insert Wrong Fact (Adv.) | 0.9 | 1.000 ± 0.020 |
| R1-Distill-Llama-8B | Insert Wrong Fact (Adv.) | 0.1 | 0.908 ± 0.289 |
| R1-Distill-Llama-8B | Insert Wrong Fact (Adv.) | 0.3 | 0.919 ± 0.272 |
| R1-Distill-Llama-8B | Insert Wrong Fact (Adv.) | 0.5 | 0.939 ± 0.240 |
| R1-Distill-Llama-8B | Insert Wrong Fact (Adv.) | 0.7 | 0.964 ± 0.185 |
| R1-Distill-Llama-8B | Insert Wrong Fact (Adv.) | 0.9 | 0.980 ± 0.138 |
| R1-Distill-Qwen-1.5B | Insert Wrong Fact (Adv.) | 0.1 | 0.596 ± 0.491 |
| R1-Distill-Qwen-1.5B | Insert Wrong Fact (Adv.) | 0.3 | 0.631 ± 0.482 |
| R1-Distill-Qwen-1.5B | Insert Wrong Fact (Adv.) | 0.5 | 0.655 ± 0.475 |
| R1-Distill-Qwen-1.5B | Insert Wrong Fact (Adv.) | 0.7 | 0.697 ± 0.460 |
| R1-Distill-Qwen-1.5B | Insert Wrong Fact (Adv.) | 0.9 | 0.799 ± 0.400 |
| R1-Distill-Qwen-14B | Insert Wrong Fact (Adv.) | 0.1 | 0.969 ± 0.173 |
| R1-Distill-Qwen-14B | Insert Wrong Fact (Adv.) | 0.3 | 0.971 ± 0.168 |
| R1-Distill-Qwen-14B | Insert Wrong Fact (Adv.) | 0.5 | 0.971 ± 0.168 |
| R1-Distill-Qwen-14B | Insert Wrong Fact (Adv.) | 0.7 | 0.994 ± 0.076 |
| R1-Distill-Qwen-14B | Insert Wrong Fact (Adv.) | 0.9 | 0.996 ± 0.065 |
| R1-Distill-Qwen-32B | Insert Wrong Fact (Adv.) | 0.1 | 0.981 ± 0.137 |
| R1-Distill-Qwen-32B | Insert Wrong Fact (Adv.) | 0.3 | 0.971 ± 0.168 |
| R1-Distill-Qwen-32B | Insert Wrong Fact (Adv.) | 0.5 | 0.984 ± 0.127 |
| R1-Distill-Qwen-32B | Insert Wrong Fact (Adv.) | 0.7 | 0.988 ± 0.110 |
| R1-Distill-Qwen-32B | Insert Wrong Fact (Adv.) | 0.9 | 0.995 ± 0.073 |
| R1-Distill-Qwen-7B | Insert Wrong Fact (Adv.) | 0.1 | 0.932 ± 0.252 |
| R1-Distill-Qwen-7B | Insert Wrong Fact (Adv.) | 0.3 | 0.935 ± 0.246 |
| R1-Distill-Qwen-7B | Insert Wrong Fact (Adv.) | 0.5 | 0.946 ± 0.226 |
| R1-Distill-Qwen-7B | Insert Wrong Fact (Adv.) | 0.7 | 0.960 ± 0.197 |
| R1-Distill-Qwen-7B | Insert Wrong Fact (Adv.) | 0.9 | 0.977 ± 0.149 |
| EXAONE-Deep-32B | Rewrite Trace (Benign) | 0.1 | 0.997 ± 0.055 |

Table 20: Robustness metrics (mean ± std) per model, intervention, and timestep on the Logic domain.

| Model | Intervention | Timestep | Mean ± Std |
|---|---|---|---|
| EXAONE-Deep-32B | Rewrite Trace (Benign) | 0.3 | 0.998 ± 0.039 |
| EXAONE-Deep-32B | Rewrite Trace (Benign) | 0.5 | 0.998 ± 0.039 |
| EXAONE-Deep-32B | Rewrite Trace (Benign) | 0.7 | 0.999 ± 0.028 |
| EXAONE-Deep-32B | Rewrite Trace (Benign) | 0.9 | 0.999 ± 0.034 |
| Nemotron-Llama-3.1-Nano-8B | Rewrite Trace (Benign) | 0.1 | 0.856 ± 0.351 |
| Nemotron-Llama-3.1-Nano-8B | Rewrite Trace (Benign) | 0.3 | 0.893 ± 0.310 |
| Nemotron-Llama-3.1-Nano-8B | Rewrite Trace (Benign) | 0.5 | 0.899 ± 0.302 |
| Nemotron-Llama-3.1-Nano-8B | Rewrite Trace (Benign) | 0.7 | 0.919 ± 0.272 |
| Nemotron-Llama-3.1-Nano-8B | Rewrite Trace (Benign) | 0.9 | 0.924 ± 0.265 |
| Phi-4-Reasoning-Plus | Rewrite Trace (Benign) | 0.1 | 0.918 ± 0.274 |
| Phi-4-Reasoning-Plus | Rewrite Trace (Benign) | 0.3 | 0.936 ± 0.245 |
| Phi-4-Reasoning-Plus | Rewrite Trace (Benign) | 0.5 | 0.930 ± 0.255 |
| Phi-4-Reasoning-Plus | Rewrite Trace (Benign) | 0.7 | 0.941 ± 0.235 |
| Phi-4-Reasoning-Plus | Rewrite Trace (Benign) | 0.9 | 0.937 ± 0.243 |
| QwQ-32B | Rewrite Trace (Benign) | 0.1 | 0.986 ± 0.118 |
| QwQ-32B | Rewrite Trace (Benign) | 0.3 | 0.992 ± 0.089 |
| QwQ-32B | Rewrite Trace (Benign) | 0.5 | 0.997 ± 0.059 |
| QwQ-32B | Rewrite Trace (Benign) | 0.7 | 0.995 ± 0.070 |
| QwQ-32B | Rewrite Trace (Benign) | 0.9 | 0.997 ± 0.059 |
| R1-Distill-Llama-8B | Rewrite Trace (Benign) | 0.1 | 0.845 ± 0.362 |
| R1-Distill-Llama-8B | Rewrite Trace (Benign) | 0.3 | 0.897 ± 0.304 |
| R1-Distill-Llama-8B | Rewrite Trace (Benign) | 0.5 | 0.908 ± 0.289 |
| R1-Distill-Llama-8B | Rewrite Trace (Benign) | 0.7 | 0.947 ± 0.223 |
| R1-Distill-Llama-8B | Rewrite Trace (Benign) | 0.9 | 0.965 ± 0.183 |
| R1-Distill-Qwen-1.5B | Rewrite Trace (Benign) | 0.1 | 0.728 ± 0.445 |
| R1-Distill-Qwen-1.5B | Rewrite Trace (Benign) | 0.3 | 0.765 ± 0.424 |
| R1-Distill-Qwen-1.5B | Rewrite Trace (Benign) | 0.5 | 0.831 ± 0.375 |
| R1-Distill-Qwen-1.5B | Rewrite Trace (Benign) | 0.7 | 0.838 ± 0.369 |
| R1-Distill-Qwen-1.5B | Rewrite Trace (Benign) | 0.9 | 0.883 ± 0.322 |
| R1-Distill-Qwen-14B | Rewrite Trace (Benign) | 0.1 | 0.913 ± 0.281 |
| R1-Distill-Qwen-14B | Rewrite Trace (Benign) | 0.3 | 0.939 ± 0.239 |
| R1-Distill-Qwen-14B | Rewrite Trace (Benign) | 0.5 | 0.957 ± 0.202 |
| R1-Distill-Qwen-14B | Rewrite Trace (Benign) | 0.7 | 0.970 ± 0.169 |
| R1-Distill-Qwen-14B | Rewrite Trace (Benign) | 0.9 | 0.977 ± 0.149 |
| R1-Distill-Qwen-32B | Rewrite Trace (Benign) | 0.1 | 0.936 ± 0.245 |
| R1-Distill-Qwen-32B | Rewrite Trace (Benign) | 0.3 | 0.949 ± 0.221 |
| R1-Distill-Qwen-32B | Rewrite Trace (Benign) | 0.5 | 0.964 ± 0.185 |
| R1-Distill-Qwen-32B | Rewrite Trace (Benign) | 0.7 | 0.968 ± 0.176 |
| R1-Distill-Qwen-32B | Rewrite Trace (Benign) | 0.9 | 0.986 ± 0.118 |
| R1-Distill-Qwen-7B | Rewrite Trace (Benign) | 0.1 | 0.820 ± 0.384 |
| R1-Distill-Qwen-7B | Rewrite Trace (Benign) | 0.3 | 0.880 ± 0.325 |
| R1-Distill-Qwen-7B | Rewrite Trace (Benign) | 0.5 | 0.905 ± 0.293 |
| R1-Distill-Qwen-7B | Rewrite Trace (Benign) | 0.7 | 0.935 ± 0.246 |
| R1-Distill-Qwen-7B | Rewrite Trace (Benign) | 0.9 | 0.960 ± 0.196 |

| Cluster | Size | Example sentences | Summary |
|---|---|---|---|
| 37 | 1105 | "Ww, no."
"Wait, no." | Abrupt, terse negations rejecting a prior point. |
| 71 | 978 | "Let me switch gears and focus on that."
"Let me try to refocus." | Explicitly redirecting attention back to the main topic. |
| 22 | 517 | "Wait, actually, is that correct?"
"Wait, that seems correct?" | Expressing uncertainty and checking correctness. |
| 331 | 400 | "Wait, no, wait, that's a different topic."
"Wait, wait, no, that's a different topic." | Flagging that the discussion is off-topic. |
| 4 | 347 | "Wait, is that equal to something?"
"Wait, is that right?" | Quick checks of equality or validity in a derivation. |
| 1640 | 342 | "Wait, maybe I should solve the equation step by step."
"Wait, let me think about the equation again:" | Reflecting on algebraic equation setup or manipulation. |
| 2820 | 333 | "So, $f(8) + f(2) = \cdots = 12$."
"Wait, but if $c$ is the period, then $f(n + c) = f(n) + 1$." | Working through functional properties and periodicity. |
| 816 | 297 | "If $m = n$, then $\gcd(m, n) = m$."
"Compute GCD(30, 240) = 30." | Reasoning about greatest common divisors. |
| 159 | 249 | "Alright, now I need to get back on track."
"Let me make sure I'm back on track." | Attempts to resume or stay on the main thread. |
| 272 | 239 | "Wait, no, no, hold on."
"Wait, that's about hearsay." | Pausing and signalling something needs reconsideration. |
| 97 | 229 | "Wait, that's not related at all."
"Wait, hold on, no, that's not related." | Explicitly stating a point is unrelated. |
| 216 | 219 | "Hmm, wait, is that true?"
"Wait, wait, is that true?" | Questioning the truth value of a claim. |
| 31 | 213 | "Wait, but hold on, wait."
"Wait, but hang on." | Hesitant interjections before clarifying a caveat. |
| 373 | 201 | "Wait, no, actually, that's not right."
"No, that's not right either." | Identifying and correcting inaccuracies. |
| 472 | 189 | "Looking back at the problem, it's about pens and money."
"Back to pencils." | Reasoning through a combinatorial pens-and-pencils problem. |

Table 16: Top 15 conversational clusters by size, with representative examples and high-level summaries.

| | | | | |
|---|---|---|---|---|
| Quantum entanglement | Neural style transfer | Photosynthesis | Plate tectonics | Classical Greek mythology |
| Ancient Egyptian hieroglyphs | The French Revolution | Supermassive black holes | Cryptocurrency mining | Nanotechnology in medicine |
| The Great Barrier Reef | Roman aqueducts | Renaissance art techniques | Dinosaur paleobiology | String theory |
| Medieval blacksmithing | Particle accelerators | The Silk Road | Coral bleaching | Japanese tea ceremony |
| Artificial neural networks | The Industrial Revolution steam engines | Mars rover missions | Evolutionary game theory | Viking longships |
| Aztec civilization | The Great Wall of China | Quantum computing qubits | History of printing press | Combinatorial game theory |
| Ancient Roman law | Solar power satellites | Cave paintings at Lascaux | Atmospheric greenhouse effect | Riemann hypothesis |
| Apollo moon landings | Photonics | Thermodynamics laws | Microplastic contamination | Narwhal ecology |
| *Cryptococcus neoformans* fungus | Möbius strip | *Homo erectus* migration | Astrophotography techniques | Origins of jazz music |
| Bioluminescent organisms | Easter Island moai | Chaos theory | Tea cultivation in Assam | Internet protocol history |
| Shakespearean sonnets | Tropical rainforest ecology | Desertification in the Sahel | Quantum tunneling | Origami mathematics |
| Holographic principle | Nobel Prize history | Biodiversity hotspots | Gel electrophoresis | Polar ice cores |
| Neolithic Göbekli Tepe | Space elevator concepts | Renewable wind energy | Mayan calendar | Deep sea hydrothermal vents |
| Solar eclipses | Cryptography history | Antarctic penguin colonies | Renaissance astronomy | Probability theory foundations |
| Greek philosophy | Cybersecurity ethics | Photosynthetic algae biofuels | Ancient Sumerian cuneiform | Ocean plastic pollution |
| Saturn's rings | Mathematical knot theory | Roman concrete durability | Augmented reality | Neolithic agriculture |
| History of chess | Electric vehicles | Artificial photosynthesis | Celestial mechanics | Inca road system |
| Machine learning fairness | Medieval alchemy | Sustainable urban design | Chinese calligraphy | Cognitive behavioral therapy |
| Fluid dynamics of bird flight | CRISPR gene editing | Mount Everest expeditions | Hubble Space Telescope discoveries | Human genome project |
| Roman gladiatorial games | Dark matter detection | Impressionist painting | Blockchain consensus algorithms | Ottoman architecture |

Table 17: 100 topics used for generating starts of CoTs for our *Unrelated CoT* intervention.

