# OpenReview forum: "Are Reasoning LLMs Robust to Interventions on their Chain-of-Thought?"
_ICLR.cc/2026/Conference — ICLR 2026 Poster_

### Official Review · Reviewer_6rea · 2025-10-27

**Soundness:** 2
**Presentation:** 3
**Contribution:** 3
**Rating:** 6
**Confidence:** 4

**Summary:**

This paper studies how robust are reasoning models to disruptions of their own reasoning traces. To study this question, authors introduce a comprehensive intervention framework that contains benign, neutral, and adversarial interventions and analyze models' behaviors including 1) performance on different data domain, different intervening timesteps, different models, 2) doubtfulness rate, and 3) CoT length. The results indicate that reasoning models are robust in general, but the robustness is not style-invariant.

**Strengths:**

1. The paper is well-written, easy to follow, and the findings are clear. The findings by the authors could give a big contribution to the community.
2. The motivation of this work (a new benchmark for reasoning robustness) is valid and sound. The early parts of Section 3 well describes this point.
3. Authors comprehensively analyze the observed phenomenon in Section 4. This helps readers to understand the limitation of the robustness of the reasoning models more deeply. -- The finding in line 416-418 is very interesting.

**Weaknesses:**

1. In Section 3 line 135, "RLLMs outperform non-reasoning models of comparable and even larger size, ...". In order to support this sentence, I think it is more fair and reliable to compare reasoning models and their non-reasoning counterparts (e.g., Qwen3 think vs no-think, DeepSeek V3 vs DeepSeek R1). Could you report results for one of these?
2. In line 187, "To validate semantic preservation, we manually ..." So what is the result of this manual comparison? Since the paraphrasing reasoning leads a key finding, ensuring that the paraphrasing is performed well by the models is very important to support authors' claims.
3. Why is "Unrelated CoT" adversarial? isn't it neutral? -- inserting "unrelated" factual content is neutral now. Other two types of adversarial reasoning contain incorrect information, whereas this type does not. By definition by authors, neutral interventions mean "irrelevant information into the CoT", which also fits to this type as well. I understand authors' intention, so to improve soundness of this work, clarification is needed.
4. The findings in Section 4.2 The role of doubt in reasoning is not surprising.
5. In Table 5, Llama-nemotron-8B shows the highest increase of CoT length in paraphrasing reasoning scenario. Why does this happen? This result could indicate that the effect of paraphrasing is only applied to R1-distill-qwen/llama models. Please justify this result.

**Questions:**

1. What if the intervened reasoning is given to another model that is different from the model that originally generates the reasoning trace?
2. Is there any reason that authors try to fix the timestep instead of reasoning step? If authors try this, what's the reason not to choose this strategy?

---

> ### Author Response · Authors · 2025-12-03
> **Author Response to Reviewer 6rea (1/2)**
>
> Thank you very much for the detailed and insightful review. We are pleased the reviewer finds the motivation of our work clear and finds our analysis comprehensive, interesting, and easy to follow, giving an important contribution to the community. Below, we respond to the individual points mentioned in the review.
>
> ---
>
> > **(W1)** In Section 3 line 135, "RLLMs outperform non-reasoning models of comparable and even larger size, ...". In order to support this sentence, I think it is more fair and reliable to compare reasoning models and their non-reasoning counterparts (e.g., Qwen3 think vs no-think, DeepSeek V3 vs DeepSeek R1). Could you report results for one of these?
>
> Thank you for the great suggestion. We conducted the requested experiment using Qwen3-30B-A3B in both its Thinking and Instruct configurations, and we included the results in Tables 1 and 11 in the revision.
>
> The results support our initial claim as the Thinking model outperforms the Non-Thinking baseline across all evaluated tasks. This advantage is most evident in symbolic manipulation domains. In the word sorting task the Thinking model achieved 50.8% accuracy compared to 2.8% for the Non-Thinking model. We observed a similar trend in Dyck languages where the Thinking model scored 38.1% against 15.2%. Our error analysis indicates that the Non-Thinking model produces chain-of-thought responses for arithmetic and logic prompts even without explicit prompting. We attribute this behavior to the high frequency of mathematical reasoning data seen during pretraining. However the Non-Thinking model failed to generalize this reasoning behavior to other domains and provided ineffective short answers for symbolic tasks.
>
> ---
>
> > **(W2)** In line 187, "To validate semantic preservation, we manually ..." So what is the result of this manual comparison? Since the paraphrasing reasoning leads a key finding, ensuring that the paraphrasing is performed well by the models is very important to support authors' claims.
>
> In the revision, we include a quantitative evaluation using GPT-5.1 as a judge to determine whether the paraphrases faithfully retain the structure and arguments of the originals. We sample 300 paraphrases, with 100 each from the Logic, Science, and Math benchmarks. GPT-5.1 finds the paraphrases faithful in 92 samples from the Logic benchmark, 98 from the Science benchmark, and 93 from the Math benchmark, yielding an overall accuracy of 94.3%. This demonstrates that the paraphrases are faithful in almost all instances. Manual inspection of the failure cases shows that the Logic and Math benchmarks contain more intricate reasoning, with the models examining more cases and constructing linguistically more complex logical arguments compared to Science, where many of the reasoning steps simply rely on applying formulas, which yield themselves more to summarization. Another failure mode is that the summary states something as a fact which was only considered hypothetical by the original model. This accounts for most of the few cases labeled as non-faithful by GPT-5.1.
>
> ---
>
> > **(W3)** Why is "Unrelated CoT" adversarial? isn't it neutral? -- inserting "unrelated" factual content is neutral now. Other two types of adversarial reasoning contain incorrect information, whereas this type does not. By definition by authors, neutral interventions mean "irrelevant information into the CoT", which also fits to this type as well. I understand authors' intention, so to improve the soundness of this work, clarification is needed.
>
> Thank you for pointing this out. We agree that the “Unrelated CoT” intervention is a borderline case, and the reviewer’s arguments make a good case to include it in the neutral category. We still opt for the adversarial category, because we want to differentiate it from the two neutral interventions that insert text entirely unrelated to reasoning. Contrarily “Unrelated CoT” could be more misleading, because it still follows the CoT format, but with unrelated content. This has a more adversarial quality than adding unrelated text. We have clarified this argument in Section 3.1.
>
> ---
>
> > **(W4)** The findings in Section 4.2 The role of doubt in reasoning is not surprising.
>
> Although some readers may anticipate the results, we maintain that a rigorous scientific evaluation is necessary. We provide this analysis in our paper. Moreover, the role of doubt is an interesting and important finding, but not the only contribution of our paper, as we primarily evaluate the robustness of RLLMs to interventions on their CoT traces and the effects on reasoning length, specifically efficiency. Therefore, our paper provides significant value to the community, even if this particular aspect appears predictable.

---

> > ### Author Response · Authors · 2025-12-03
> > **Author Response to Reviewer 6rea (2/2)**
> >
> > ---
> >
> > > **(W5)** In Table 5, Llama-nemotron-8B shows the highest increase of CoT length in paraphrasing reasoning scenario. Why does this happen? This result could indicate that the effect of paraphrasing is only applied to R1-distill-qwen/llama models. Please justify this result.
> >
> > Thank you for pointing this out. We have included a list of CoT length increases for all 9 models to the revision in Section 4.2. The decrease in CoT length for paraphrasing is present in 7 out of 9 models, however, 2 models generate significantly longer CoTs, especially Phi-4-Reasoning. After conducting a qualitative analysis, we observed that this happens because often after an intervention, these two models repeat past reasoning steps unusually often, and way more so than all other 7 models, indicating a much more pronounced lack of intervention robustness for these two models compared to the other 7 models.
> >
> > ---
> >
> > > **(Q1)** What if the intervened reasoning is given to another model that is different from the model that originally generates the reasoning trace?
> >
> > This is a great suggestion. We have conducted the experiment swapping reasoning traces between DeepSeek-R1-Distill-Llama-8B, DeepSeek-R1-Distill-Qwen-1.5B, and QwQ-32B. The results are added in Section 5 in the revision.
> >
> > Our results indicate that intervened reasoning traces are highly transferable across models of different sizes and architectures. The final performance depends primarily on the capability of the model receiving the trace rather than the model that generated it, confirming our result that robustness is primarily dependent on overall model capabilities. As shown in the table below, QwQ-32B maintains a success rate exceeding 0.97 regardless of the trace source. Furthermore, the smaller DeepSeek-R1-Distill-Qwen-1.5B achieves higher accuracy when utilizing traces from larger models than when using its own self-generated reasoning.
> >
> > ---
> >
> > > **(Q2)** Is there any reason that authors try to fix the timestep instead of reasoning step? If authors try this, what's the reason not to choose this strategy?
> >
> > The main reason is to maintain comparability between different models. The length of reasoning traces varies considerably across models (e.g., as discussed above, Llama-Nemotron-8B generates CoTs longer than other models of its size, especially after an intervention), so using absolute step indices complicates the analysis. We consider that using relative positions (timesteps in the reasoning chain) yields results that are easier to understand and fairer. For example, this allows us to compare interventions at the beginning or end of the reasoning trace, irrespective of the total trace length.

---

### Official Review · Reviewer_FNsV · 2025-11-01

**Soundness:** 3
**Presentation:** 3
**Contribution:** 3
**Rating:** 4
**Confidence:** 4

**Summary:**

This paper investigates the robustness of Reasoning Large Language Models to interventions within their own generated CoT reasoning processes. The authors introduce a controlled evaluation framework where they interrupt a model's correct CoT at various timesteps and inject one of seven different perturbations. These interventions are categorized as benign (e.g., paraphrasing), neutral (e.g., inserting random text), and adversarial (e.g., adding a wrong reasoning step). The same model is then prompted to continue its reasoning, allowing for a direct measurement of its recovery capabilities. The study evaluates multiple RLLMs across MATH, SCIENCE, and LOGIC domains. Key findings indicate that RLLMs are generally robust to such disruptions, with robustness increasing with model scale. However, this recovery comes at a computational cost, often significantly inflating the CoT length. Finally, the paper identifies the expression of "doubt" as a key recovery mechanism and reveals that models are not style-invariant, as paraphrasing can suppress doubt and degrade accuracy despite preserving semantics.

**Strengths:**

1.  By intervening directly in a model's own reasoning trace, the authors create a clean and realistic testbed for self-correction, which is a significant step forward in robustness evaluation.

2.  The study is impressively broad, covering 9 models, 3 domains, 7 intervention types, and 5 timesteps. This thoroughness lends high credibility to the conclusions and suggests that the findings are generalizable.

3.  The paper is well-written. The research question, methods, and results are communicated with great clarity, making the experiments easy to understand.

**Weaknesses:**

1.  While identifying "doubt" is a major strength, the analysis relies on an LLM-based classifier to label sentences. This approach, while pragmatic, is somewhat superficial. The paper would be significantly strengthened by a deeper investigation into how "doubt" is represented internally. The activation analysis in the appendix is a good first step, but it feels underdeveloped. A more detailed analysis connecting specific internal states to the expression of doubt would benifit the contribution of the work.

2. The interventions are all single-step disruptions. In many real-world scenarios (e.g., interacting with a faulty tool), errors can be more subtle or cascading. For instance, a model might accept a slightly incorrect tool output and then build several subsequent reasoning steps upon it. The current framework does not evaluate robustness to such persistent or multi-step correlated errors, which may be harder for the model to detect and recover from.

3.  The paper excels at diagnosing the problem but is light on proposing solutions. The conclusion suggests that future training methods should address these weaknesses (e.g., style invariance, recovery efficiency), but it offers no preliminary experiments on how this could be achieved. For example, could data augmentation with paraphrased CoTs during finetuning improve style robustness?

**Questions:**

1.  Regarding the LLM-based "doubt" classifier: How was its performance validated? What is its estimated accuracy, and what are some examples of sentences it correctly or incorrectly classifies as expressing doubt? Could you elaborate more on the findings from the activation analysis?

2.  The interventions are currently single-step. Have you considered experiments with multi-step interventions or cascading errors (e.g., where a wrong fact is introduced and then used logically in the next step)? Do you hypothesize that the "doubt" mechanism would still be effective in such scenarios?

3.  The finding that paraphrasing hurts performance is intresting. It suggests models overfit to a particular reasoning "style." Based on this, what are your thoughts on potential training strategies to mitigate this? Would training on a more stylistically diverse set of CoTs (e.g., via paraphrasing augmentation) be a promising direction?

4.  Could you clarify the process for the "Continuation with other model" intervention? Is the other model a weaker, non-reasoning-specialized model?

---

> ### Author Response · Authors · 2025-12-03
> **Author Response to Reviewer FNsV (1/2)**
>
> We thank the reviewer for the constructive feedback and valuable suggestions. We are particularly happy that the reviewer finds our study “impressively broad” with “high credibility to the conclusions”, creating a “clean and realistic testbed” which “is a significant step forward in robustness evaluation”.
>
> ---
>
> > **(W1)** The paper would be significantly strengthened by a deeper investigation into how "doubt" is represented internally. A more detailed analysis connecting specific internal states to the expression of doubt would benefit the contribution of the work.
>
> We thank the reviewer for the great suggestion. We agree that this could strengthen the analysis. Future work, therefore, could look into clusterings of doubt expressions, whether they are predictable from preceding tokens, or designing steering vectors to control different doubt expressions. Further, methods such as Direct Logit Attribution and looking into attention heads related to doubting behaviour could also be interesting directions. We have added a discussion of these future directions in Section 6.
>
> ---
>
> > **(Q1)** Regarding the LLM-based "doubt" classifier: How was its performance validated? How was its performance validated? What is its estimated accuracy, and what are some examples of sentences it correctly or incorrectly classifies as expressing doubt? Could you elaborate more on the findings from the activation analysis?
>
> Thank you for asking this! We validated the doubt classifier's performance against human annotations. We asked four annotators to label 200 GPT-generated doubtful phrases and 200 randomly selected non-doubtful sentences, establishing a ground truth dataset. Our classifier achieved 93.75% accuracy with 89.24% precision and 99.50% recall for detecting doubt. Inter-annotator agreement among humans yielded an average Cohen's kappa of 0.8385, while the classifier achieved a kappa of 0.8742 against human majority labels and 0.8457 when averaged against individual annotators. These results demonstrate that the classifier performs comparably to human raters, whose average accuracy against ground truth was 93.59%. The high recall indicates the classifier successfully identifies nearly all doubtful expressions, while maintaining strong precision to minimize false positives.
>
> In our revision, we have added these results in Appendix D.3.
>
> ---
>
> > **(W2/Q2)** The interventions are all single-step disruptions. In many real-world scenarios (e.g., interacting with a faulty tool), errors can be more subtle or cascading. For instance, a model might accept a slightly incorrect tool output and then build several subsequent reasoning steps upon it. The current framework does not evaluate robustness to such persistent or multi-step correlated errors, which may be harder for the model to detect and recover from.
>
> We have added an experiment where we injected between one and five distinct interventions separated by multiple reasoning steps to simulate recurring mistakes. Results are in Appendix A.2 in the revision.
>
> Our findings indicate that model performance remains robust also under repeated interventions. While accuracy generally follows a downward trend as the number of interventions increases, this decline is gradual for most models. For instance, the DeepSeek-R1-Distill-Llama-8B model moves from 95.0% accuracy with a single intervention to 88.1% with five. Larger models display even greater resilience as EXAONE-Deep-32B and QwQ-32B maintained accuracy above 99% across all five steps. These results suggest that the reasoning capabilities of these models allow them to effectively detect and recover from cascading errors as well.

---

> > ### Author Response · Authors · 2025-12-03
> > **Author Response to Reviewer FNsV (2/2)**
> >
> > ---
> >
> > > **(W3/Q3)** For example, could data augmentation with paraphrased CoTs during finetuning improve style robustness? Based on this, what are your thoughts on potential training strategies to mitigate this? Would training on a more stylistically diverse set of CoTs (e.g., via paraphrasing augmentation) be a promising direction?
> >
> > Thank you for these great questions. We are happy to elaborate. We also agree that fine tuning with more diverse data will help RLLMs become more robust. Here, we should distinguish between supervised fine-tuning (SFT and reinforcement learning (RL) as post-training strategies. In SFT, we have direct control over the reasoning traces and can apply strategies mentioned by the reviewer, such as paraphrasing as augmentation, or curating more diverse datasets. For RL, we could reward models for more diverse reasoning traces so the model naturally learns to reasoning in different formats and styles. We have included this discussion in Section 5.
> >
> > ---
> >
> > > **(Q4)** Could you clarify the process for the "Continuation with other model" intervention? Is the other model a weaker, non-reasoning-specialized model?
> >
> > Sure, we are happy to clarify this. We give details on the used model and parameters in Appendix D (Tab. 9). For the “continuation with other model” intervention, we used Qwen/Qwen2.5-32B-Instruct, i.e. a non-reasoning model of comparable size or larger with respect to the evaluated reasoning models. The exact prompt we used is in Appendix D.1. We have clarified this in Section 3.1 and added pointers to the information in the supplementary material.

---

### Official Review · Reviewer_pGbL · 2025-11-01

**Soundness:** 4
**Presentation:** 4
**Contribution:** 3
**Rating:** 6
**Confidence:** 5

**Summary:**

The paper studies how robust reasoning models (RMs) are to perturbations in their reasoning trace. They find that accuracy does not suffer as much, but there is an impact on efficiency.

**Strengths:**

- Efficiency of RMs is an important consideration. While it is not the explicit goal, I think the paper's results on efficiency are interesting and the dataset can help study efficiency further.
- Reasonable set of interventions and results across models.
- Ablations on why RMs are able to recover (using "wait" -like tokens) and how the interventions increase inference time per answer. I think this is the strongest part of the paper.

**Weaknesses:**

- Some of the results need more examination. For example, table 5 shows that benign rewrites lead to drops up to 60%, but then Table 6 shows that there is no drop in CoT length across all intervention timesteps (except 0.9).
- There are interesting observations, but the insight is weak. For example, introducing wrong reasoning increases accuracy robustness and paraphrasing reduces accuracy. Surely, there are some confounding factors here that can be explored?
- Many of the results are "observations", but no solution is motivated or discussed. For example, if certain kinds of interventions lead to higher use of "doubt tokens", then can it be used to improve accuracy (without introducing adversarial interventions)?

**Questions:**

I think there are some confounding factors at play.

1. It seems counter-intuitive that introducing benign rewrites reduces accuracy, whereas introducing wrong reasoning improves accuracy. Can the authors unpack this observation? Is it the writing style? What if the benign rewrite is written in a more tentative style or a question? It will be good to do some qualitative study/ablation to see what the cause of robustness is.

2. Same question for efficiency. How can benign rewrites reduce CoT length and accuracy, while other (stronger) interventions don't lead to accuracy drop? Something else may be at play, beyond the type of intervention defined by the authors.

---

> ### Author Response · Authors · 2025-12-03
> **Author Response to Reviewer pGbL**
>
> Thank you for the detailed review and the positive evaluation of our work, highlighting our set of interventions is reasonable and that we include important ablations to strengthen our analysis. We are also happy the reviewer finds our insights on reasoning model efficiency interesting. We answer the questions and doubts in the review individually below.
>
> ---
>
> > **(W1)** Some of the results need more examination. For example, table 5 shows that benign rewrites lead to drops up to 60%, but then Table 6 shows that there is no drop in CoT length across all intervention timesteps (except 0.9).
>
> We apologize for the confusion, and thank the reviewer for the good catch. Tab. 5 shows the changes in reasoning length for different models, and averaged across intervention timesteps. Tab. 6 shows the changes for different intervention timesteps, but averaged across models. This means that outliers can skew the numbers when averaging, and this is what happened here. Specifically, in Tab. 6, two models lead to a biased average: Llama-Nemotron-8B and Phi-4-Reasoning-Plus, which means the “Benign Rewrite” length increase can only be observed in Table 5.
>
> ---
>
> > **(W2/Q1)** There are interesting observations, but the insight is weak. For example, introducing wrong reasoning increases accuracy robustness and paraphrasing reduces accuracy. Surely, there are some confounding factors here that can be explored? It seems counter-intuitive that introducing benign rewrites reduces accuracy, whereas introducing wrong reasoning improves accuracy. Can the authors unpack this observation? Is it the writing style? What if the benign rewrite is written in a more tentative style or a question? It will be good to do some qualitative study/ablation to see what the cause of robustness is.
>
> Thank you for pointing this out. To clarify, inserting wrong reasoning does also lead to a drop in robustness, however, this drop is less pronounced than the one due to the “Rewrite Trace” intervention. We have clarified this in the description of Figure 2.
>
> ---
>
> > **(Q2)** Same question for efficiency. How can benign rewrites reduce CoT length and accuracy, while other (stronger) interventions don't lead to accuracy drop? Something else may be at play, beyond the type of intervention defined by the authors.
>
> To clear this up, benign rewrites do lead to reduced CoT lengths in most models, which impact the accuracy of the model, whereas other interventions increase the CoT length, because the model needs additional tokens to recover from the interventions we introduce.
>
> ---
>
> > **(W3)** Many of the results are "observations", but no solution is motivated or discussed. For example, if certain kinds of interventions lead to higher use of "doubt tokens", then can it be used to improve accuracy (without introducing adversarial interventions)?
>
> This is a great suggestion. We conducted an additional experiment where we appended "Wait" after the interventions to simulate doubt expression and repeated the evaluations on 1 benchmark (Logic) on 326 traces and for all 9 models, with 91280 traces generated per model. Results are included in Section 5 in the revision.
>
> The results confirm that doubt consistently improves recovery from interventions across various models. This effect is particularly pronounced for “Unrelated CoT” interventions where the accuracy increased by over 20.5% at the same timestep. These results clearly indicate that doubt expression contributes significantly to recovering from adversarial injections.
>
> We conclude from this that doubt indeed plays an important role in recovery from errors, as we already analyzed in the original submission, and can be leveraged to improve accuracy of reasoning models.

---

### Official Review · Reviewer_5HWi · 2025-11-01

**Soundness:** 3
**Presentation:** 4
**Contribution:** 3
**Rating:** 2
**Confidence:** 4

**Summary:**

This paper investigates the robustness of reasoning process when interruptions are injected in the middle of reasoning trace.
They introduce unconventional setting where correct pre-generated reasoning traces from RLLMs are produced beforehand, and at each reasoning step in the CoT trace, they inject three types of noises. The benign injection continues the generation from other reasoning model or simply introduce a paraphrasing which do not change the original semantics of the reasoning trace. Neutral injection is adding random characters or documents from wikipedia, whereas adversarial perturbation is to inject unrelated cot trace or incorrect facts.

They test the reasoning robustness in three domains, math, science and logic. And for five difference RLLMs, they analyze the recovery behavior of each model. Starting from the observation that RL trained reasoning models are more capable of finding the reasoning errors compared to conventional large language models, based on the evaluation on BigBench-mistake, this paper builds a novel benchmark that systematically evaluates self-correction and self-reflection abilities. They define the robustness metric by checking whether the language model is able to produce final correct answer, despite the intervention at the intermediate reasoning steps.

**Strengths:**

The dataset builds over large number of existing benchmark dataset, extending over various science, math, and logic domains.
Moreover, this paper experimented over wide variety of reasoning models including, R1 distill, exaone, nemotron, phi, and QwQ.
The discovery that LRMs are generally robust to various types of intervention, regardless of the benign, neural, or adversarial .types.
Also, it is notable that LRM generation length inflates the most when interrupted with random texts for recovery.

**Weaknesses:**

The largest concern is that the reasoning trace interruption scenario is extremely far from realistic usage cases, which assumes that the model reasoning will be abruptly interrupted by external signals. Moreover the interruptions are hardly meaningful in logical or semantic sense, since it introduces noises that are completely irrelevant with previous context and model generations. Rather than injecting random text from external sources, what if the model is injected with reasoning trace that leads to wrong answer? Does doubt expression contribute to successfully recovering from this adversarial injection?
Also, it would be helpful if the authors could augment their explanation on realistic cases where random interruptions might happen during the model generation.

**Questions:**

Please answer to the weakness above.

---

> ### Author Response · Authors · 2025-12-03
> **Author Response to Reviewer 5HWi**
>
> We thank the reviewer for the clear and constructive feedback. We are happy that the reviewer acknowledges the “wide variety of reasoning models” and domains that we evaluate, and we are also happy that the reviewer finds our discoveries regarding robustness and length inflation interesting. Below, we answer in detail to the concerns mentioned in the review.
>
> ---
>
> > **(W1)** The largest concern is that the reasoning trace interruption scenario is extremely far from realistic usage cases, which assumes that the model reasoning will be abruptly interrupted by external signals.
>
> Thank you for pointing this out. We acknowledge that some of the evaluated CoT interventions may be unlikely in practical settings, but we respectfully disagree that this diminishes or invalidates the value of our findings. First, we would like to note that our interventions include “Wrong Continuation”, “Hallucinated Fact”, or “Unrelated CoT” as adversarial interventions, which resemble realistic problems. Mistakes and hallucinations are common problems of LLMs. Moreover, our aim is to systematically study CoT robustness, which necessarily requires a benchmark setting that may not be entirely faithful to deployment settings, although we believe our intervention covers a range of realistic problems. Finally, we would like to note that it is typical for work on CoT analysis to assume similar setups, for example [1, 2, 3].
>
> In light of these reasons, we strongly believe our work provides interesting, novel and relevant insights regarding CoT robustness that will inform improvements in real-world settings, for example increasing accuracy through doubting, more robust reasoning through invariance to style, and training to keep reasoning traces short even when making errors. We have clarified the nature of our interventions in Section 3.1 to clearly state their purpose and extent.
>
> **References**\
> [1] Yang et al.: How Well Can Reasoning Models Identify and Recover from Unhelpful Thoughts? In EMNLP, 2025\
> [2] Zhou et al.: Can Language Models Perform Robust Reasoning in Chain‑of‑thought Prompting with Noisy Rationales? In NeurIPS, 2024\
> [3] Hong et al.: Context Rot: How Increasing Input Tokens Impacts LLM Performance. In Chroma Technical Report, July 2025
>
> ---
>
> > **(W2)** Moreover the interruptions are hardly meaningful in logical or semantic sense, since it introduces noises that are completely irrelevant with previous context and model generations.
>
> We are happy to clarify this. We evaluate seven different interventions, of which only three may be inconsistent with the context (“Wikipedia Text”, “Random Character”, “Unrelated CoT”). The other four interventions are explicitly designed to take the context into account. Thus, our benchmark evaluates a range of interventions, some more akin to random noise (like we evaluate OOD robustness in vision models by applying random corruptions to images), and others targeting realistic error modes. We have clarified this in Section 3.1 in the revision.
>
> ---
>
> > **(W3)** Rather than injecting random text from external sources, what if the model is injected with reasoning trace that leads to wrong answer? Does doubt expression contribute to successfully recovering from this adversarial injection?
>
> This is a very interesting experiment, thank you for suggesting it. We appended "wait" after the interventions to simulate doubt expression and repeated the evaluations on 1 benchmark (Logic), on 326 x 8 x 5 x 7 traces and for all models. Results are included in Section 5 in the revision.
>
> The results confirm that doubt consistently improves recovery from interventions across various models. This effect is particularly pronounced for “Unrelated CoT” interventions where the accuracy increased by over 20.5% at the same timestep. These results clearly indicate that doubt expression contributes significantly to recovering from adversarial injections.
>
> We conclude from this that doubt indeed plays an important role in recovery from errors, as we already analyzed in the original submission, and can be leveraged to improve accuracy of reasoning models.
>
> ---
>
> > **(W4)** Also, it would be helpful if the authors could augment their explanation on realistic cases where random interruptions might happen during the model generation.
>
> One example would be tool calls performed in the CoT, where seemingly random content could be introduced through a call to a web search tool, a RAG system, etc. However, most of the interventions we test specifically test the robustness of the model to non-random interventions, as pointed out in our response to **(W1)**.

---

### Comment · Area_Chair_7s2s · 2025-11-27
**Reminder: discussion period is about to end**

Dear authors,

This is a friendly reminder that you have until Dec 03 to respond to the initial reviews. While a response is optional, I encourage you to discuss the feedback with the reviewers to improve your submission. If you have any concerns regarding the reviews, please let me know.

Best,

AC

---

> ### Author Response · Authors · 2025-11-27
>
> Dear AC,
>
> Thank you very much for the reminder and for your efforts in managing the review process for ICLR 2026.
> We apologize for the delay in our response, which was due to an illness in the author team. We are finalizing our revisions now and plan to post our responses and the updated version by tomorrow.
>
> Sincerely,
> the authors

---

### Author Response · Authors · 2025-12-03
**General Response**

Dear AC and Reviewers,

We thank you for your efforts in reviewing for ICLR 2026. We appreciate the positive and helpful reviews. We have uploaded our revised version and responses to all reviewers.

Our paper receives ratings of 6, 6, 4, and 2, which demonstrates that our research provides important contributions of interest to the community. Reviewers find our paper clear and well-written (6rea, FNsV) and highlight the comprehensive evaluation (6rea, 5HWi) and “impressively broad” (FNsV) analysis. They find our benchmark “reasonable” (pGbL), yielding a “significant step forward in robustness evaluation” (FNsV) and a “big contribution to the community" (6rea). Finally, reviewers express strong interest in our analysis of doubt tokens (pGbL) and reasoning efficiency (5HWi, pGbL).

We highlight our main contributions:
1. A comprehensive benchmark to test the robustness of reasoning LLMs via benign, neutral, and adversarial interventions to their reasoning traces;
2. A broad analysis of 9 reasoning models on benchmarks from 3 domains;
3. Insights on the role of doubt tokens in error recovery, the insufficient robustness of reasoning models to stylistic changes in their reasoning traces, and the effects of errors on reasoning length, which directly translates into higher deployment costs.

Taken together, we provide a highly relevant analysis that directly benefits and informs model improvements. Our insights suggest improving reasoning models by using more stylistically diverse chain-of-thought traces during training. Specifically, we motivate training models to create shorter traces even in the presence of noise or errors, and improve performance by increasing doubt in models.

Reviewers also suggest important points for clarification and interesting ablations, which we include in our revision. In our uploaded revision PDF, we mark changes in blue. In individual responses to all reviewers, we address their remaining concerns in detail.

---

### Meta-Review · Area_Chair_smh7 · 2026-01-03

**Summary:**

This paper investigate the robustness of reasoning LLMs to interventions of their chain of thoughts. The ratings are borderline, with the main concerns listed as follows:

- while there are some observations that different types of perturbations lead to different behaviors, the paper lacks deep insights and analysis for the reason behind those behavior differences.
- the interventions and perturbations are very unnatural and unlikely to happen in practice, therefore making the observations less relevant to real world robustness
- lack of discussions and experiments on mitigation methods

**Reviewer Concerns:**

The authors clarified specific questions from reviewer pGbL's concern on lack of insights, but they did not provide further development or deeper analysis.

The authors provided detailed clarification on why their settings follow "standard" frameworks in the literature and the results could be relevant. This should partially address the 2nd concern.

The authors provided additional experiments by adding "wait" token, which provide some discussions on potential mitigation methods.

**Reviewer Scores:**

I expect the reviewers originally voting acceptance to mostly maintain their original score.

I expect reviewer 5HWi to slightly increase their score, but likely keep it below the acceptance threshold, depending on how convinced they would agree that the settings are realistic enough to provide relevant results for real world application scenarios.

I expect reviewer FNsV to slightly increase their score, as their concerns were mostly addressed in the rebuttal.

---

### Decision · Program_Chairs · 2026-01-26

Accept (Poster)